

# Symplectic reduction of Yang-Mills theory with boundaries: from superselection sectors to edge modes, and back

Aldo Riello⋆

Physique Théorique et Mathématique, Université libre de Bruxelles,
Campus Plaine C.P. 231, B-1050 Bruxelles, Belgium

⋆ aldo.riello@ulb.be

## Abstract

I develop a theory of symplectic reduction that applies to bounded regions in electromagnetism and Yang–Mills theories. In this theory gauge-covariant superselection sectors for the electric flux through the boundary of the region play a central role: within such sectors, there exists a natural, canonically defined, symplectic structure for the reduced Yang–Mills theory. This symplectic structure does not require the inclusion of any new degrees of freedom. In the non-Abelian case, it also supports a family of Hamiltonian vector fields, which I call "flux rotations," generated by smeared, Poisson-non-commutative, electric fluxes. Since the action of flux rotations affects the total energy of the system, I argue that flux rotations fail to be dynamical symmetries of Yang–Mills theory restricted to a region. I also consider the possibility of defining a symplectic structure on the union of all superselection sectors. This in turn requires including additional boundary degrees of freedom aka "edge modes." However, I argue that a commonly used phase space extension by edge modes is inherently ambiguous and gauge-breaking.



# 1   Introduction

**1.1   Context and motivations**   Building on [1–5], in this article I elaborate and present—in a self-contained manner—a theory of symplectic reduction for Yang–Mills gauge theories over finite and bounded regions. Physically, this article answers the following question: what are the quasilocal degrees of freedom (dof) in electromagnetism and non-Abelian Yang-Mills (YM) theories? By "quasilocal" I mean "confined in a finite and bounded region," with possibly a degree of nonlocality allowed *within* the region.

Gauge theoretical dof cannot be completely localized, since gauge invariant quantities are somewhat nonlocal, the prototypical example being a Wilson line. In electromagnetism, or any Abelian YM theory, although the field strength $F_{\mu\nu} = 2\partial_{[\mu}A_{\nu]}$ provides a basis of local gauge invariant observables, its components do not provide gauge invariant *canonical* coordinates on field space: in 3 space dimensions, $\{E^i(x), B^j(y)\} = \epsilon^{jik}\partial_k\delta(x,y)$ is not a canonical Poisson bracket and the presence of the derivative on the right-hand-side is the signature of a nonlocal behavior.

From a canonical perspective, what is responsible for this nonlocality is the Gauss constraint, $\mathsf{G} = \partial^i E_i + [A^i, E_i] - \rho$, whose Poisson bracket generates gauge transformations.[1] The Gauss constraint is an *elliptic* equation that initial data on a Cauchy surface $\Sigma$ must satisfy. In other words, the initial values of the gauge potential $A_i$ and its momentum $E^i$ cannot be freely specified throughout a Cauchy surface $\Sigma$. Ultimately, this is the source of both the nonlocality and the difficulty of identifying freely specifiable initial data—the "true" *degrees of freedom*.

To summarize, the identification of the quasilocal dof requires dealing with (*1*) the Gauss constraint and with (*2*) the separation of pure-gauge and physical dof. The two tasks are related, but distinct. I will start with the second task by focusing on the foliation of the YM phase space by the gauge orbits.

**1.2   Sketch of the reduction procedure**   The goal of the quasilocal symplectic reduction is to construct a closed and non-degenerate, i.e. symplectic, 2-form on the reduced phase space of YM and matter fields over a spacelike region $R \subset \Sigma$. The reduced phase space is the space of gauge orbits of the YM configurations—comprising the gauge potential, the electric field, and the matter fields—which are on shell of the Gauss constraint.

Since the reduced phase space is de facto inaccessible to an intrinsic description, it is most convenient to concentrate one's efforts on the space of gauge-variant fields. Viewing this space as a foliated space with the structure of a fiducial infinite-dimensional fibre bundle, I will focus on the construction of a horizontal and gauge-invariant (i.e. basic) presymplectic 2-form. This pre-symplectic 2-form can then be restricted to the on-shell[2] gauge-variant configurations and thus projected down to the reduced phase space. Here, "horizontal" means "transverse to the

---

[1]See [6,7] for a thorough discussion of the central and pervasive role the Gauss constraint plays in quantum electrodynamics and quantum Yang–Mills theory.

[2]In this article, "on-shell" refers to the Gauss constraint only, and not to the equations of motion.

gauge orbits." Note that, since there is no canonical notion of horizontality,[3] the *construction* of the basic 2-form will a priori depend on the choice of such a notion.[4]

A basic 2-form projects down to a *non-degenerate* 2-form on the reduced space only if its kernel coincides with the space of gauge transformations, i.e. with the space of vertical vectors. I will find that the "naive" presymplectic 2-form satisfies this condition automatically only in Abelian theories—or in the absence of boundaries. In non-Abelian theories with boundaries, however, a canonical completion of the naive presymplectic 2-form exists which leads to an actual non-degenerate, and therefore symplectic, 2-form on the reduced phase space.

This completion also erases all dependence on the chosen notion of horizontality, so that the final result is independent from it.

Crucially, throughout this construction, I will let gauge transformations be *un*con-strained at the boundary $\partial R$.[5] I will also associate separate phase spaces to different gauge-classes of the electric flux through $\partial R$, which I will call *covariant superselection sectors* of the flux. As I will explain later, these two ingredients are closely related to each other.

The fixing of a (covariant) superselction sector has two crucial consequences. First, within a covariant superselection sector the construction of the symplectic form on the reduced phase space does *not* require the addition of new dof, i.e. it does not require a modification of the phase space manifold; even in the non-Abelian case this extension is avoided, and the projected 2-form is rather made non-degenerate through the addition of a *canonical* term to the presymplectic 2-form which eliminates its kernel. And second, (only) within these sectors it is possible to define a symplectic form on the reduced phase space which is independent of the chosen notion of horizontality—think of this as a form of gauge-fixing independence.

### 1.3 Flux superselection sectors

Given their central role in the construction of the reduced symplectic structure, I will now spend a few words discussing flux superselection sectors.

Flux superselection sectors find their origin in the structure of the Gauss constraint in bounded regions. To streamline this introductory discussion, let me introduce the Coulombic potential $\varphi$, so that the Gauss constraint can be written as an elliptic (Poisson) equation that fixes the Laplacian of $\varphi$ in terms of the YM charge density,[6] $D^2\varphi = \rho$. However, in a finite and bounded region, this Poisson equation is insufficient to fully fix the Coulombic potential: a boundary condition is needed.

A preferred choice of boundary condition (in YM theory) is dictated by the interplay between the symplectic and fibre-bundle geometry of the YM phase space. This choice of boundary condition corresponds to fixing the electric flux $f$ through the boundary $\partial R$. I.e. denoting by $s^i$ the unit outgoing normal at $\partial R$, the missing boundary condition is $D_s\varphi = f$.

From the global perspective of the entire Cauchy surface $\Sigma$, $f$ at $\partial R$ depends on the geometry[7] and the field content of both $R$ and its complement. But these data are inaccessible from

---

[3]This statement is analogous, albeit more encompassing, to the statement that there is no canonical gauge fixing. Note that despite the non-existence of a canonical choice, I will discuss a particularly natural (i.e. convenient) one.

[4]Note that an explicit notion of horizontality (as encoded in a connection form, see below) is not needed to decide *whether* a given form is basic, but it is needed to *build* a horizontal form out of a given one.

[5]This distinguishes the current approach from the standard lore, by which gauge symmetry is usually frozen or broken at the boundary. See e.g. [8–11] (cf. also [12–15] on the related but different topic of asymptotic boundaries and infra-red sectors of gauge theories).

[6]$D = d + A$ is the gauge-covariant differential, and $D^2$ is the gauge covariant Laplace-Beltrami operator.

[7]Indeed, although the "creation" of a point charge in $R$ will in general affect the flux $f$, to be able to *compute* the induced change in the flux one needs the Green's function of the Laplacian over $\Sigma$, which in turn depends on the geometry of both $R$ *and its complement*. Therefore, from *within R*, there is no way to compute the induced change in flux. Maybe more strikingly, at generic background configurations of the non-Abelian theory (which are irreducible), the charge and flux data are completely independent. At reducible configurations (and in Abelian theories), at most a finite number of surface integrals of $f$ is fixed—through an integral Gauss law—by the charge content within $R$. This number is bound by the dimension of the charge group of the theory. See [5] for a more

the quasilocal perspective intrinsic to $R$: hence the only meaningful manner to understand the quasilocal Gauss constraint is within a *superselection sector* of fixed electric flux. However, fixing the flux completely goes against my pledge of not treating gauge transformations differently at the boundary. Therefore, in the non-Abelian theory where the flux is gauge variant, it becomes necessary to introduce *covariant superselection sectors* (CSSS) labelled not by a flux, but by a conjugacy class of fluxes, $[f]$.

In quantum mechanics, a superselection sector is invoked when a certain physical observable $O$ with eigenvalues $o_\alpha$ commutes with *all* other available observables in the theory; then, the theory's states decompose into statistical mixtures of states in sectors labeled by the eigenvalues $o_\alpha$. Such an $O$ is said "superselected." How is this characterization of the notion of superselection consistent with the one used above for the electric flux $f$? In Abelian theories $f$ does not appear in the reduced symplectic structure associated to a superselection sector; therefore, $f$ is treated as a mere parameter which commutes with all other quantities, and the two notion of superselection are perfectly compatible. In the non-Abelian theory, however, the situation is more subtle: the different components of $f$ turn out to be conjugate to each other as a consequence of the above-mentioned completion procedure. This fact, whose origin I will explain in the next paragraph, means in turn that the bulk "Coulombic" electric field—which depends functionally on $f$ as a consequence of the Gauss constraint—*fails* to commute with itself; however, $f$ *does* commute with all the unconstrained, i.e. "freely specifiable," bulk fields and observables, which are provided by the matter fields as well as the "radiative" part of the YM field.

**1.4 Flux rotations** In non-Abelian theories, the enlargement of the notion of superselection sector to its covariant counterpart introduces the possibility of "rotating" $f$ within its conjugacy class $[f]$ *without* altering neither $\rho$ nor $A$. These transformations—that I name *flux rotations*—produce genuinely new field configurations, for they alter, via the Gauss constraint, the Coulombic potential and thus the energy content of $R$. Flux rotations are therefore *physical* transformations that survive the reduction.

This is the problem with the naive reduction in the non-Abelian theory: flux rotations end up being in the kernel of the candidate symplectic 2-form. Heuristically, this could have been expected because the more naive reduction procedure gets rid of the pure-gauge dof as well as of the Coulombic dof which are conjugate to them; but since $f$ is imprinted precisely in the Coulombic part of the electric field, getting rid of it means that $f$ drops from the candidate symplectic 2-form.

This is not a problem in Abelian theories, where $f$ is fixed once and for all in a given superselection sector. But in non-Abelian theories, the covariant superselection sector only fixes the conjugacy class of $f$: within $[f]$, variations of $f$ *relative to the bulk fields* become thus possible and physically relevant—but remain "invisible" to the candidate symplectic 2-form.

**1.5 Completion of the reduced symplectic structure** This problem can be overcome by completing the candidate symplectic structure in a canonical manner. Indeed, within a CSSS, the space of allowed fluxes $[f]$ is equipped with a *canonical* symplectic structure: the Kirillov–Konstant–Souriau (KKS) symplectic structure for the (co)adjoint orbits. Using the techniques developed for the bulk fields, also the KKS symplectic structure can be modified into a basic 2-form and thus fed into the reduction procedure.

Remarkably, the inclusion of this basic 2-form on the flux space, makes the ensuing *total* symplectic structure over a covariant superselection sector $[f]$ independent from the chosen notion of horizontality!

---

thorough discussion.

The ensuing "completed" symplectic structure over a covariant superselection sector $[f]$ can be characterized in a simple way: it is given by the sum of (the pullback to the given superselection sector of) the naive symplectic structure $\Omega_{\mathrm{YM}} = \int \mathrm{Tr}(\mathbb{d}E \wedge \mathbb{d}A)$ and a boundary KKS 2-form over the space of covariantly superselected fluxes $[f]$, $\omega_{\mathrm{KKS}}^{[f]}$; inclusion of charged matter fields does not alter this structure. Schematically:[8]

$$\Omega_{\mathrm{total}}^{[f]} \approx \Omega_{\mathrm{YM}} + \Omega_{\mathrm{matter}} + \omega_{\mathrm{KKS}}^{[f]}. \tag{1}$$

As sketched in Appendix A.1, this construction closely parallels the "shifting trick" [16, Sect.26] (also [17]) used to define the Marsden–Weinstein–Meyer symplectic reduction [18, 19] at (regular) values of the moment map different from zero.[9] Heuristically, this trick allows one to reduce at non-zero values of the moment map, and thus to deal in our case with non-vanishing electric fluxes through the boundary. However, in order to make this statement precise, a careful separation is necessary between bulk gauge transformations—whose moment map is the usual Gauss constraint and must therefore be set to zero value—and boundary gauge transformations—whose moment map is, loosely speaking, given by the non-vanishing flux. Elaboration of this equivalent viewpoint is left to future work.[10]

### 1.6 Superselection sectors vs. edge modes

The physical viability of the notion of super-selection sector for the electric fluxes—which implies that of the superselection of the electric charge charge [21, 22][11]—has been criticized in the past [28, 29] on the basis of a theory of relational reference frames [30, Chapter 6].[12]

This tension can be reconciled through a careful analysis of the gluing of the reduced phase spaces associated to adjacent bounded regions [5, Sect.6]. Indeed, this analysis shows that the origin of the superselection is precisely to be found in the dismissal of the dof in one of the two regions which would otherwise serve as a relational reference frame (see [5, Sect.7] and [20, Rmk.5.5] for details)—a novel viewpoint which is consistent with the general stance on superselection elaborated e.g. in [28, 30, 31].[13]

It is nonetheless instructive to attempt the reduction procedure in the union of all superse-lection sectors to see what precisely fails in this context. But, on the union of all superselection sectors, no symplectic structure can be defined in a completely canonical fashion: therefore to go beyond the superselection framework, it is necessary to resort to an *extended* phase space which includes additional new fields symplectically conjugate to the electric flux—the most natural choice for these new dof gives precisely DF's "edge modes."[14]

Curiously, the total symplectic structure on a covariant superselection sector, once written in an over-complete set of coordinates over $[f]$, is formally similar to the "edge mode" proposal of Donnelly and Freidel (DF) [32–35] (see also [8, 9, 36–40] among many others). But the two *are* very different in substance: contrary to what happens in a superselection sector, in the

---

[8]The symbol $\approx$ here indicates equality upon pullback to the covariant superselection sector of $[f]$.

[9]I would like to thank Michele Schiavina for pointing out this trick to me.

[10]In [20], which summarizes the content of this and other papers by myself and collaborators in more elementary but less general terms, the presentation follows much more closely the Marsden–Weinstein paradigm, even though it makes no *explicit* mention of the shifting trick.

[11]See also: [23,24] as well as [25]. Moreover, for recent results on a residual gauge-fixing dependence of QED in the presence of (asymptotic) boundaries and flux superselection, see [26] (and also [27]). At present it is unclear how these recent results square with the classical treatment presented here.

[12]See also the review [31] and references therein.

[13]See also [1] and especially the introduction of [3] for a discussion of the relationship between a choice of relational reference frame for the gauge system and the choice of a functional connection $\varpi$ over field space (the geometrtical role of $\varpi$ is the aforementioned definition of a notion of horizontality in field-space).

[14]From the perspective of [30, Chapter 6], edge modes can be understood as (models for) phase, or gauge, reference frames. In [5, Sect.6-7] and [20, Sect.5] I argued that the most natural such reference frame is constituted by the (gauge invariant) dof present in the complementary region.

DF edge mode framework *no* superslection sector is fixed and new dof *are* added to the phase space. The relationship between the two constructions will be made explicit in Sect. 5.

Beyond the addition of new dof, there is also another price to pay for the DF construction: a new type of ambiguity emerges which is related to the *non*-canonical nature of the DF extension of phase space. This ambiguity is rooted in the fact that the DF extended phase space can as well be obtained from breaking the boundary gauge symmetry, and different ways to do so lead to different (although isomorphic) DF phase spaces.

In sum, it appears that any attempt to go beyond the superselection framework must not only introduce new dof but also introduce ambiguities which would not be present otherwise. In my opinion, this strongly supports the viability and necessity of the concept of covariant superselection sectors. In the conclusions I will come back to this point.

After this overview, it is time to delve into the details.

## 2 Mathematical setup

**2.1 The YM configuration space as a foliated space**    To start, let me introduce some notation and recall some simple facts. Let $G = \mathrm{SU}(N)$ be the *charge group* of the YM theory under investigation; in the Abelian case, I will have electromagnetism in mind, with $G = \mathrm{U}(1)$ or $(\mathbb{R}^+, \times)$. The quasilocal configuration space of the gauge potential[15] over $R \subset \Sigma$, with $\mathring{R} \cong \mathbb{R}^D$, is the space of Lie-algebra valued one forms $A \in \mathcal{A} := \Omega^1(R, \mathrm{Lie}(G))$ that transform under gauge transformations $g : R \to G$ as $A \mapsto A^g = g^{-1}Ag + g^{-1}\mathrm{d}g$, with d the spatial exterior derivative. The space of gauge transformations $\mathcal{G} := \mathcal{C}^\infty(R, G) \ni g$ inherits a group structure from $G$ via pointwise multiplication. Call $\mathcal{G}$ the *gauge group*. The action of gauge transformations $g$ on the gauge potential $A$, provides an action of $\mathcal{G}$ on $\mathcal{A}$. The orbits of this action, $\mathcal{O}_A$, are called gauge orbits and their space $\mathcal{A}/\mathcal{G} = \bigcup_{A \in \mathcal{A}} \mathcal{O}_A$ is the space of physical configurations. This is the "true" configuration space of the theory, but it is de facto inaccessible.

The orbits of $\mathcal{G}$ on $\mathcal{A}$ induces an infinite-dimensional foliation of configuration space, $\mathcal{A} \to \mathcal{A}/\mathcal{G}$.[16] An infinitesimal gauge transformation, $\xi \in \mathrm{Lie}(\mathcal{G})$, defines a vector field tangent to the gauge orbits. I will denote this vector field[17] by $\xi^\sharp \in \Gamma(\mathrm{T}\mathcal{A})$,

$$\xi^\sharp = \int (\mathrm{D}_i \xi)^\alpha(x) \frac{\delta}{\delta A_i^\alpha(x)}, \tag{2}$$

where $\int := \int_R \mathrm{d}^D x$. At each $A \in \mathcal{A}$, the span of the $\xi^\sharp_{|A}$ defines the *vertical* subspace of $\mathrm{T}_A \mathcal{A}$, i.e. $V_A := \mathrm{Span}_{\xi \in \mathrm{Lie}(\mathcal{G})}(\xi^\sharp_{|A}) \subset \mathrm{T}_A \mathcal{A}$. The ensemble of these vertical subspace gives the vertical distribution $V = \mathrm{T}(\bigcup_{A \in \mathcal{A}} \mathcal{O}_A) \subset \mathrm{T}\mathcal{A}$.

---

[15]This kinematical (or off-shell) setup can be understood as stemming e.g. from a canonical approach in temporal gauge ($A_0 = 0$). Consider electromagnetism. Since $A$ and $E$ will be treated as independent coordinates on $\mathrm{T}^*\mathcal{A}$, the spatial components of the gauge potential can be further gauge-fixed to e.g. Coulomb gauge, $\nabla^i A_i = 0$, without losing the Coulombic potential which is the pure-gradient part of $E$. Analogous statements hold in the non-Abelian case. This will become clear below. Even *without* fixing temporal gauge, one is led to the same kinematical phase space $\mathrm{T}^*\mathcal{A}$, by focusing on the ghost-number-zero part of the BV-BFV boundary structure of YM theory, e.g. [41] (here "boundary" is understood in relation to the spacetime bulk).

[16]Strictly speaking the above choice of gauge group does not lead to a bona-fide (albeit infinite dimensional) foliation of $\mathcal{A}$. This is due to the presence of reducible configurations [42–44], i.e. configurations that are left invariant by a (necessarily finite) subgroup of $\mathcal{G}$. For what concerns the present article, this issue can be solved by replacing $\mathcal{G}$ with $\mathcal{G}_* \subset \mathcal{G}$, the subgroup of gauge transformations which are trivial at one given point $x_* \in R$, $g(x_*) \equiv 1$. For a more thorough discussion of this subtlety and its physical consequences in relation to global charges, see [5].

[17]The covariant derivative is $\mathrm{D}\xi = \mathrm{d}\xi + [A, \xi]$.

Physically, vertical directions are "pure gauge" and variations of the fields in these directions are physically irrelevant. Therefore, the "physical directions" in $T\mathcal{A}$ must be those transverse to $V$, i.e. the *horizontal directions* $H \subset T\mathcal{A}$. However, the decomposition $T\mathcal{A} = V \oplus H$ is not canonically defined; in loose terms, there is no canonical "gauge fixing" for infinitesimal variations of the fields. Rather, the choice of an equivariant horizontal distribution is equivalent to the choice of a functional Ehresmann connection on $\mathcal{A} \to \mathcal{A}/\mathcal{G}$. This is a functional 1-form valued in the Lie algebra of the gauge group,[18]

$$\varpi \in \Omega^1(\mathcal{A}, \mathrm{Lie}(\mathcal{G})), \tag{3}$$

characterized by the following two properties:

$$\begin{cases} \mathbb{i}_{\xi^\sharp} \varpi = \xi, \\ \mathbb{L}_{\xi^\sharp} \varpi = [\varpi, \xi] + \mathbb{d}\xi. \end{cases} \tag{4}$$

Hereafter, double-struck symbols refer to geometrical objects and operations in field space: $\mathbb{d}$ is the (formal) field-space exterior differential,[19] with $\mathbb{d}^2 \equiv 0$; $\mathbb{i}$ is the inclusion, or contraction, operator of field-space vectors into field-space forms; and $\mathbb{L}_{\mathbb{X}}$ is the field-space Lie derivative of field-space vectors and forms along the field-space vector field $\mathbb{X} \in \mathfrak{X}^1(\mathcal{A})$. When acting on forms, the field-space Lie derivative can be computed through Cartan's formula, $\mathbb{L}_{\mathbb{X}} = \mathbb{i}_{\mathbb{X}} \mathbb{d} + \mathbb{d} \mathbb{i}_{\mathbb{X}}$. When acting on vectors, I will also use the Lie-bracket notation, $\mathbb{L}_{\mathbb{X}} \mathbb{Y} \equiv [\![\mathbb{X}, \mathbb{Y}]\!]$. Finally, I will denote the wedge product between field space forms by $\lambda$.

The first of the above properties, the projection property, is what grants one to define $H$ as the complement to $V$ through

$$H := \ker \varpi. \tag{5}$$

This means that the vertical and horizontal projections in $T\mathcal{A}$ can be written respectively as $\widehat{V}(\mathbb{X}) = (\mathbb{i}_{\mathbb{X}} \varpi)^\sharp$ and $\widehat{H}(\mathbb{X}) = \mathbb{X} - (\mathbb{i}_{\mathbb{X}} \varpi)^\sharp$. The second property ensures that the above definition is compatible with the group action of $\mathcal{G}$ on $\mathcal{A}$, i.e. transforms "covariantly" under the action of gauge transformations. The term $\mathbb{d}\xi$ is only present if $\xi$ is chosen differently at different points of $\mathcal{A}$, i.e. if $\xi$ is an infinitesimal *field-dependent* gauge transformation. Gauge fixings, and changes between gauge fixings, bring about typical examples of field-dependent gauge transformations. Note that the generalization to field-dependent gauge transformations comes "for free:" the equivariance property of $\varpi$ as it appears in (4) can be deduced from the standard transformation property $\mathbb{L}_{\xi^\sharp} \varpi = [\varpi, \xi]$ for field-independent $\xi$'s (i.e. $\xi$'s constant throughout $\mathcal{A}$), together with the projection property of $\varpi$.

From a mathematical perspective, generalizing to field-dependent gauge transformations means promoting $\xi$ to be a general, i.e. not-necessarily constant, section of the trivial bundle $\mathcal{A} \times \mathrm{Lie}(\mathcal{G}) \to \mathcal{A}$, i.e. $\xi \in \Gamma(\mathcal{A}, \mathcal{A} \times \mathrm{Lie}(\mathcal{G}))$. In this way, *any* vertical vector field $\mathbb{V} \in \Gamma(\mathcal{A}, V)$ can be written as $\mathbb{V} = \xi^\sharp$ for the field-dependent $\xi = \varpi(\mathbb{V})$—here $\xi^\sharp$ is defined pointwise over $\mathcal{A}$ as in (2). This setup is best expressed in terms of the *action Lie algebroid*[20] associated to the action of $\mathcal{G}$ on $\mathcal{A}$, with anchor map $\cdot^\sharp$ (see [5, Sect.2]). In the following, equations that hold only for field-*in*dependent $\xi$'s will be accompanied by the notation ($\mathbb{d}\xi = 0$) which indicates the choice of a *constant* section of $\mathcal{A} \times \mathrm{Lie}(\mathcal{G}) \to \mathcal{A}$. (Later we will replace $\mathcal{A}$ with a larger space $\Phi$ which includes the electric and matter fields.)

One last property of the horizontal distribution $H = \ker \varpi$ is its anholonomicity, i.e. a measure of its failure of being integrable in the sense of Frobenius's theorem:

$$\mathbb{F} := \varpi([\![\widehat{H}(\cdot), \widehat{H}(\cdot)]\!]) \in \Omega^2(\mathcal{A}, \mathrm{Lie}(\mathcal{G})), \tag{6}$$

---

[18] I usually pronounce $\varpi$ by its typographical code, VAR-PIE.

[19] I prefer this notation to the more common $\delta$, because the latter is often used to indicate vectors as well as forms, hence creating possible confusions.

[20] For generalities on action Lie algebroids, see e.g. [45].

so that $\mathbb{F}^{\sharp}$ captures the vertical part of the Lie bracket of any two horizontal vector fields. As standard from the theory of principal fibre bundles, one can prove that

$$\mathbb{F} = \mathbb{d}\varpi + \tfrac{1}{2}[\varpi \overset{\wedge}{,} \varpi], \tag{7}$$

for $[\,\cdot\,,\,\cdot\,]$ the Lie-bracket in $\mathrm{Lie}(G)$ pointwise extended to $\mathrm{Lie}(\mathcal{G})$. The functional curvature $\mathbb{F}$ satisfies an algebraic Bianchi identity $\mathbb{d}_H \mathbb{F} = \mathbb{d}\mathbb{F} + [\varpi \overset{\wedge}{,} \mathbb{F}] = 0$.

If $\mathbb{F} = 0$, $\varpi$ is said to be flat. In this case, the horizontal distribution is integrable and defines a horizontal foliation of $\mathcal{A}$. A leaf in such a foliation provides a global section $\mathcal{A}/\mathcal{G} \to \mathcal{A}$, i.e. a gauge fixing proper. In this sense functional connections are "infinitesimal" gauge fixings which generalize the usual global concept.

Finally, the choice of a horizontal distribution allows to introduce a new differential: the horizontal differential $\mathbb{d}_H$. This is by definition transverse to the vertical, pure gauge, directions: that is, for any $\mathbb{X}$, $\mathbb{i}_{\mathbb{X}}\mathbb{d}_H A := \mathbb{i}_{\widehat{H}(\mathbb{X})}\mathbb{d}A$. Thus, $\mathbb{i}_{\xi^{\sharp}}\mathbb{d}_H A \equiv 0$. Later, I will show that (in most circumstances of interest) $\mathbb{d}_H$, expressed in terms of $\varpi$, takes the form of a "covariant differential" $\mathbb{d}_H = \mathbb{d} - \varpi$, whose failure to be nilpotent is captured by $\mathbb{F}$.

**2.2  An example: the SdW connection**  An important example of functional connection is provided by the Singer–DeWitt connection (SdW), $\varpi_{\mathrm{SdW}}$. Whenever $\mathcal{A}$ is equipped with a positive-definite gauge-invariant supermetric[21] $\mathbb{G}$—i.e. a supermetric such that $\mathbb{L}_{\xi^{\sharp}}\mathbb{G} = 0$ if $\mathbb{d}\xi = 0$,—then an appropriately covariant functional connection I call SdW can be defined by orthogonality to the fibres, $\mathrm{T}\mathcal{A} = H_{\mathbb{G}} \perp V$.

In the case of YM theory there is a natural such candidate for $\mathbb{G}$. This is the kinetic supermetric[22]

$$\mathbb{G}(\mathbb{X}^1, \mathbb{X}^2) := \int \sqrt{g}\, g^{ij}\mathrm{Tr}(X_i^1 X_j^2) \qquad \forall \mathbb{X}^{1,2} = \int X_i^{1,2} \frac{\delta}{\delta A_i} \in \mathrm{T}\mathcal{A}. \tag{8}$$

From this definition, it is easy to see that a vector $\mathbb{h} = \int h_i \frac{\delta}{\delta A_i}$ is SdW-horizontal, i.e. $\mathbb{G}$-orthogonal to all $\xi^{\sharp} = \int \mathrm{D}_i \xi \frac{\delta}{\delta A_i}$, if and only if

$$\begin{cases} \mathrm{D}^i h_i = 0 & \text{in } R, \\ s^i h_i = 0 & \text{at } \partial R. \end{cases} \tag{9}$$

Therefore, SdW-horizontality is a generalization, to the non-Abelian and bounded case, of Coulomb gauge for the infinitesimal variations of $A$. In this sense, SdW-horizontal variations generalize the concept of a (transverse) photon in the same manner.

Demanding that $\mathbb{h} \equiv \widehat{H}_{\mathbb{G}}(\mathbb{X}) = \mathbb{X} - (\mathbb{i}_{\mathbb{X}}\varpi_{\mathrm{SdW}})^{\sharp}$ is SdW-horizontal for all $\mathbb{X}$, leads to the following defining equation for $\varpi_{\mathrm{SdW}}$:

$$\begin{cases} \mathrm{D}^2 \varpi_{\mathrm{SdW}} = \mathrm{D}^i \mathbb{d}A_i & \text{in } R, \\ \mathrm{D}_s \varpi_{\mathrm{SdW}} = \mathbb{d}A_s & \text{at } \partial R. \end{cases} \tag{10}$$

In the Abelian case, the SdW connection is exact and flat.[23] In the non-Abelian case, the anholonomicity of $\varpi_{\mathrm{SdW}}$ satisfies the following [3, 5]:

$$\begin{cases} \mathrm{D}^2 \mathbb{F}_{\mathrm{SdW}} = g^{ij}[\mathbb{d}_H A_i \overset{\wedge}{,} \mathbb{d}_H A_j] & \text{in } R, \\ \mathrm{D}_s \mathbb{F}_{\mathrm{SdW}} = 0 & \text{at } \partial R. \end{cases} \tag{11}$$

---

[21] A weaker condition is indeed sufficient, see [3] and [5].

[22] The name comes from the fact that the kinetic energy of YM theory is given by $K = \mathbb{G}(\dot{\mathbb{A}}, \dot{\mathbb{A}})$ where $\dot{\mathbb{A}} := \int \dot{A}_i \frac{\delta}{\delta A_i}$. Hereafter the contraction $\mathrm{Tr}(\cdot\cdot)$ is normalized to coincide with the killing form over $\mathrm{Lie}(G)$ for $G = \mathrm{SU}(N)$.

[23] A connection $\varpi$ is exact iff $G$ is Abelian and $\varpi$ is flat.

Notice that the unrestricted nature of the gauge freedom at the boundary is crucial to obtain not only a boundary condition for the SdW horizontal modes, but most importantly a well-posed boundary value problem for $\varpi_{\text{SdW}}$ (and thus for $\mathbb{F}_{\text{SdW}}$, too).

In the following I will call elliptic boundary value problems of the same kind as (10) and (11), "SdW boundary value problems." Their properties are analyzed in detail in [5]. In this article, I will consider SdW boundary value problems to be uniquely invertible in $\Omega^0(\mathcal{A}, \text{Lie}(\mathcal{G}))$.[24]

**2.3  The phase space of YM theory with matter as a fibre bundle**  These constructions can be readily generalized to the full *off-shell* phase space

$$\Phi := \text{T}^*\mathcal{A} \times \Phi_{\text{matter}} \ni (A_i, E^i, \psi, \overline{\psi}). \tag{12}$$

Here, $E^i$ is the electric field and, for definiteness, I chose $\Phi_{\text{matter}} = \Psi \times \overline{\Psi}$ to be the space of Dirac spinors and their conjugates—which correspond to the matter's canonical momenta. The off-shell phase space is foliated by the action of gauge transformations, $A^g = g^{-1}Ag + g^{-1}\text{d}g$, $E^g = g^{-1}Eg$, $\psi^g = g^{-1}\psi$, and $\overline{\psi}^g = \overline{\psi}g$. Thus, the corresponding infinitesimal field-dependent gauge transformations $\xi \in \Gamma(\Phi, \Phi \times \text{Lie}(\mathcal{G}))$ define vertical vectors tangent to the gauge orbits in $\Phi$,

$$\xi^\sharp = \int \text{D}_i \xi \frac{\delta}{\delta A_i} + [E^i, \xi]\frac{\delta}{\delta E^i} - \xi\psi\frac{\delta}{\delta\psi} + \overline{\psi}\xi\frac{\delta}{\delta\overline{\psi}} \in \Gamma(\Phi, V) \subset \mathfrak{X}^1(\Phi). \tag{13}$$

Notice that I have redefined $V$ to be the vertical subspace of $\text{T}\Phi$ rather than $\text{T}\mathcal{A}$. Similarly, for the horizontal distribution $H \subset \text{T}\Phi$, $\text{T}\Phi = H \oplus V$.

A choice of an equivariant horizontal distribution can once again be identified with a choice of a connection on $\Phi$. Rather than considering general connections over $\Phi$, I will *exclusively* focus on connections defined on $\Phi$ through a *pull-back* from $\mathcal{A}$. That is, $\pi : \Phi \to \mathcal{A}$ the canonical projection, I will only consider functional connections of the form[25] $\tilde{\varpi} = \pi^*\varpi \in \Omega^1(\Phi, \text{Lie}(\mathcal{G}))$. This restriction will be important in the following. Henceforth, $\tilde{\varpi} \rightsquigarrow \varpi$.

Horizontal differentials can also be introduced on $\Phi$. Explicitly, they read

$$\begin{cases} \mathbb{d}_H A = \mathbb{d}A - \text{D}\varpi, \\ \mathbb{d}_H E = \mathbb{d}E - [E, \varpi], \end{cases} \quad \text{and} \quad \begin{cases} \mathbb{d}_H \psi = \mathbb{d}\psi + \varpi\psi, \\ \mathbb{d}_H \overline{\psi} = \mathbb{d}\overline{\psi} - \overline{\psi}\varpi. \end{cases} \tag{14}$$

Loosely speaking, horizontal differentials are "covariant" differentials in field space.[26] Indeed, as it is easy to show horizontal differentials transform always homogeneously under gauge transformations, even field-dependent ones:

$$\begin{cases} \mathbb{L}_{\xi^\sharp}\mathbb{d}_H A = [\xi, \mathbb{d}_H A], \\ \mathbb{L}_{\xi^\sharp}\mathbb{d}_H E = [\xi, \mathbb{d}_H E], \end{cases} \quad \text{and} \quad \begin{cases} \mathbb{L}_{\xi^\sharp}\mathbb{d}_H \psi = -\xi\mathbb{d}_H\psi, \\ \mathbb{L}_{\xi^\sharp}\mathbb{d}_H \overline{\psi} = (\mathbb{d}_H\overline{\psi})\xi. \end{cases} \tag{15}$$

Finally, from the above it is easy to prove the relationship between $\mathbb{d}_H^2$ and $\mathbb{F}$:

$$\begin{cases} \mathbb{d}_H^2 A = -\text{D}\mathbb{F}, \\ \mathbb{d}_H^2 E = -[E, \mathbb{F}], \end{cases} \quad \text{and} \quad \begin{cases} \mathbb{d}_H^2 \psi = \mathbb{F}\psi, \\ \mathbb{d}_H^2 \overline{\psi} = -\overline{\psi}\mathbb{F}. \end{cases} \tag{16}$$

---

[24]This is true at irreducible configurations, and also otherwise provided that $\text{Lie}(\mathcal{G})$ is replaced by $\text{Lie}(\mathcal{G}_*)$ modified; cf. footnote 16.

[25]For $\varpi$ a connection on $\mathcal{A}$, it is straightforward to check that $\tilde{\varpi} := \pi^*\varpi$ satisfies the two defining properties of a functional connection over $\Phi$, cf. (4).

[26]On forms that are horizontal and equivariant with respect to some representation $R$, $\mathbb{d}_H$ can be shown to to act as a covariant derivative $\mathbb{d}_H = \mathbb{d} - R(\varpi)$.

**2.4 Flat functional connections, gauge fixings, and dressings** There is a relationship between gauge fixings, flat functional connections, and dressings of the charged matter fields [3, Sect.9] (on dressings, see [46–53]). As explained at the end of paragraph 2.1, flat connections correspond to integrable horizontal distribution, i.e. to (a family of) global sections of $\mathcal{A} \to \mathcal{A}/\mathcal{G}$, i.e. to a gauge fixing. Moreover, flat connections are of the form $\varpi = \kappa^{-1}\mathbb{d}\kappa$ for some $\kappa = \kappa[A] \in \Omega^0(\mathcal{A}, \mathcal{G})$ which transforms by right translation under the action of $\mathcal{G}$, i.e. $R_g^*(h) = hg$. Pulling back $\kappa$ from $\mathcal{A}$ to $\Phi$, it is easy to prove that

$$\mathbb{d}\widehat{\psi} = \kappa\mathbb{d}_H\psi \qquad \text{for} \qquad \widehat{\psi} := \kappa\psi \tag{17}$$

and that $\widehat{\psi}$ is a composite, generally nonlocal,[27] gauge invariant field (similar equations hold for $\overline{\psi}$). E.g. in electromagnetism, the SdW connection is flat and $\widehat{\psi}$ reduces precisely to the Dirac dressing of the electron if $R = \mathbb{R}^3$ [5].

Therefore, choices of flat connections correspond to choices of dressings for the matter fields, and the horizontal differentials of the matter fields are closely related to the differentials of the associated dressed matter fields.

Its relationship to dressings equips the geometric object $\mathbb{d}_H$ with an intuitive physical interpretation.[28]

# 3 Symplectic geometry, the Gauss constraint, and flux superselection

**3.1 Off-shell symplectic geometry** Define the off-shell symplectic potential of YM theory and matter over $R$ to be given by

$$\theta := \int \sqrt{g}\,\mathrm{Tr}(E^i\mathbb{d}A_i) - \int \sqrt{g}\,\overline{\psi}\gamma^0\mathbb{d}\psi \in \Omega^1(\Phi), \tag{18}$$

where $\sqrt{g}$ is the square root of the spatial metric $g_{ij}$ on $R$, and $(\gamma^0, \gamma^i)$ are Dirac's $\gamma$-matrices.[29] Note that the first term in the equation above ($\theta_{\mathrm{YM}}$) is the tautological 1-form over $\mathrm{T}^*\mathcal{A}$—this identifies the geometrical meaning of the electric field $E^i$. Of course this formula for $\theta$ can be derived from the YM Lagrangian.

The above (polarization of the) symplectic potential is invariant under field-*in*dependent gauge transformations:

$$\mathbb{L}_{\xi^\sharp}\theta = 0 \qquad (\mathbb{d}\xi = 0), \tag{19}$$

but it fails to be invariant under field-dependent ones. Moreover, in the presence of boundaries $\theta$ fails to be horizontal even on-shell of the Gauss constraint ($\approx$):

$$\mathbb{i}_{\xi^\sharp}\theta \approx \oint \sqrt{h}\,\mathrm{Tr}(E_s\xi), \tag{20}$$

where $\sqrt{h}$ is the square root of the determinant of the induced metric on $\partial R$, and $\oint := \int_{\partial R} \mathrm{d}^{d-1}x$. This formula is the main reason why it has become common lore that gauge transformations that do not trivialize at the boundary must have a different status: their associated charge does not vanish. As proven in section 5.8, this is e.g. the interpretation embraced by the "edge

---

[27]Spatially local dressings exist in special circumstances where the gauge symmetry is "non-substantial" [52]. The prototypical example of this is a gauge symmetry introduced via a Stückelberg trick. For other examples see e.g. [3, Sect.s 7 & 8].

[28]See [3, Sect.9] for a generalization of the notion of dressing to the case of non-Abelian connections, which relates this construction to the Vilkovisky–DeWitt geometric effective action.

[29]I follow here Weinberg's conventions [54].

mode" approach of [32]. But there is an alternative to this conclusion which relies on the introduction of superselection sectors.

To understand what this means, appreciate how this choice will be imposed upon us, and understand how it can be implemented in the non-Abelian case without breaking the gauge symmetry at $\partial R$, it is essential to first devise an alternative to $\theta$ which is manifestly gauge-invariant *and* horizontal.

(For a sketch of the superselection construction in the Abelian case where most subtleties do not arise, see section 4.1.)

A manifestly gauge-invaraint and horizontal, i.e. *basic*, 1-form is obtained by taking the horizontal part of $\theta$. In formulas,

$$\theta^H := \theta(\widehat{H}(\cdot)) = \int \sqrt{g}\,\mathrm{Tr}(E^i \mathbb{d}_H A_i) - \int \sqrt{g}\,\overline{\psi}\gamma^0 \mathbb{d}_H \psi \in \Omega^1(\Phi) \tag{21}$$

is such that

$$\mathbb{L}_{\xi^\sharp}\theta^H = 0 \qquad \text{and} \qquad \mathring{\mathbb{i}}_{\xi^\sharp}\theta^H = 0, \tag{22}$$

even for field-dependent gauge transformations (cf. (15)).

The remaining part of the off-shell symplectic potential, its vertical part $\theta^V := \theta - \theta^H$, is on the other hand given by

$$\theta^V = \int \sqrt{g}\,\mathrm{Tr}(E^i D_i \varpi + \rho\varpi) = \int \sqrt{g}\,\mathrm{Tr}\big((-D_i E^i + \rho)\varpi\big) + \oint \sqrt{h}\,\mathrm{Tr}(E_s \varpi), \tag{23}$$

where I introduced the charge density $\rho := \sum_\alpha (\overline{\psi}\gamma^0 \tau_\alpha \psi)\tau_\alpha$ for $\{\tau_\alpha\}$ a Tr-orthonormal basis of Lie($G$).[30] On shell of the Gauss constraint $\theta^V$ becomes supported on the boundary (however, note that $\varpi$ is nonlocal within $R$). This is the part of $\theta$ that is responsible for it not being horizontal.

The off-shell symplectic form is defined by differentiating the off-shell symplectic potential, i.e. $\Omega := \mathbb{d}\theta$. Similarly, I define a horizontal presymplectic 2-form by differentiating the horizontal part of the symplectic potential, $\theta^H$:

$$\Omega^H := \mathbb{d}\theta^H. \tag{24}$$

Since $\theta^H$ is basic (22), and thus gauge invariant, it is not hard to realize that $\mathbb{d}_H \theta^H \equiv \mathbb{d}\theta^H$.[31] Moreover, using the Cartan calculus equation $[\![\mathbb{d}, \mathbb{L}]\!] = 0$, it is immediate to verify from (19) that $\Omega^H$ is also gauge invariant. In sum, $\Omega^H$ is basic and closed:

$$\mathring{\mathbb{i}}_{\xi^\sharp}\Omega^H = 0, \qquad \mathbb{L}_{\xi^\sharp}\Omega^H = 0, \qquad \text{and} \qquad \mathbb{d}\Omega^H = 0. \tag{25}$$

Let me observe that $\theta^H$ and $\Omega^H$ depend on the choice of horizontal distribution, i.e. on the choice of functional connection $\varpi$. Moreover, if the horizontal distribution is non-integrable i.e. $\mathbb{F} \neq 0$, one can show that $\Omega^H \neq \Omega(\widehat{H}(\cdot), \widehat{H}(\cdot))$, and only the former is a closed 2-form.[32]

A crucial remark: it is ultimately thanks to equation (19) that it is possible to define a basic and closed presymplectic structure on $\Phi$ in this geometric manner. In this regard, it is relevant to notice that equation (19) distinguishes YM theory from Chern-Simons and $BF$ theories, where no polarization exists in which the (off-shell) symplectic potential is gauge invariant— this is because in those theories the variables canonically conjugate to the gauge potential do not transform homogeneously under gauge transformations.[33]

---

[30]That is, $\rho$ is defined by $\mathrm{Tr}(\rho\xi) := \overline{\psi}\gamma^0 \xi\psi$ for all $\xi$. Indeed, $\rho$ (just as the electric field and flux) is most naturally understood as valued in the dual of the Lie algebra Lie($G$)*.

[31]Cf. footnote 26.

[32]Indeed, $\Omega(\widehat{H}(\cdot), \widehat{H}(\cdot)) = \Omega^H + \oint \sqrt{h}\,\mathrm{Tr}(f\,\mathbb{F})$ [5].

[33]See [41] for a derivation of the Wess-Zumino-Witten edge-mode theory of Chern-Simons from the failure of the latter to have a gauge-invariant symplectic structure.

**3.2 Radiative and Coulombic electric fields** Before delving into the reduction procedure, it is convenient to further analyze the dof entering $\theta^H$, and in particular its pure-YM contribution. In the previous sections I have introduced two decompositions: one for the tangent space $T\mathcal{A} = H \oplus V$, and a dual one for the 1-form $\theta = \theta^H + \theta^V \in T^*\mathcal{A}$. It is instructive to express the latter decomposition in coordinates, i.e. to define $E = E_{\text{rad}} + E_{\text{Coul}}$ so that

$$\theta_{\text{YM}}^H = \int \sqrt{g}\,\text{Tr}(E_{\text{rad}}^i \,\mathbb{d}A_i) \qquad \text{and} \qquad \theta_{\text{YM}}^V = \int \sqrt{g}\,\text{Tr}(E_{\text{Coul}}^i \,\mathbb{d}A_i). \tag{26}$$

Notice that, contrary to (21) and (23), in these formulas the burden of the horizontal and vertical projections is not carried by $\mathbb{d}_H A$ and $\varpi$, but instead by the functional properties of $E_{\text{rad}}$ and $E_{\text{Coul}}$.

Thus, from the verticality condition $\theta^V(\widehat{H}(\mathbb{X})) \equiv 0$ for all $\mathbb{X}$, one obtains the following $\varpi$-dependent conditions for $E_{\text{Coul}}$ (hereafter, $\varpi(\mathbb{X}) \equiv \mathbb{i}_{\mathbb{X}}\varpi$):

$$\int \sqrt{g}\,\text{Tr}\big(E_{\text{Coul}}^i(X_i - D_i\varpi(\mathbb{X}))\big) = 0 \qquad \forall \mathbb{X} = \int X_i \frac{\delta}{\delta A_i}. \tag{27}$$

Conversely, from the horizontality condition $\theta^H(\xi^\sharp) = \int \sqrt{g}\,\text{Tr}(E^i D_i \xi) \equiv 0$ for all $\xi$, one obtains through a simple integration by parts the following universal conditions for $E_{\text{rad}}$:

$$\begin{cases} D_i E_{\text{rad}}^i = 0 & \text{in } R, \\ s_i E_{\text{rad}}^i = 0 & \text{at } \partial R. \end{cases} \tag{28}$$

Therefore, $E_{\text{rad}}$ generalizes a transverse electric field to the non-Abelian and bounded case; this is why I call it "radiative." Independently of the choice of connection, the radiative electric field contains two independent dof, whereas the Coulombic electric field contains one.

Finally, note that the YM dof in $\theta$ organize as follows: the radiative electric field is paired with the horizontal modes of $A$ in $\theta^H$, whereas the Coulombic electric field is paired with $\varpi$ in $\theta^V$. The horizontal matter modes appearing in $\theta^H$ do not satisfy an independent condition of their own; they can be understood as "dressed matter fields" [2, 3, 5].

**3.3 An example: the SdW case** The reader will have noticed the analogy between (9) and (28) defining the SdW-horizontal variations and the radiative electric field, respectively. Indeed, the SdW choice of connection is the only one for which the velocity-momentum relation reads as a simple horizontal projection, i.e. $\mathbb{E}_{\text{rad}} = \widehat{H}_{\mathbb{G}}(\mathbb{\dot{A}})$ where $\mathbb{E}_{\text{rad}} = \int g_{ij} E_{\text{rad}}^i \frac{\delta}{\delta A_j}$ and $\mathbb{\dot{A}} = \int \dot{A}_i \frac{\delta}{\delta A_i}$.

This parallel between SdW-horizontal variations and radiative electric fields, has an analogue in the complementary sectors which comprise vertical (i.e. pure-gauge) variations and the Coulombic electric field. Indeed, equation (27) for $\varpi = \varpi_{\text{SdW}}$ dictates that the SdW-Coulombic electric field is of the form

$$E_{\text{Coul}}^i = g^{ij} D_j \varphi \qquad \text{(SdW)}, \tag{29}$$

for some $\text{Lie}(\mathcal{G})$-valued scalar $\varphi$ I will call the SdW Coulombic potential.

**3.4 The Gauss constraint and superselection sectors** Note that, in the absence of boundaries, the first of the equations (28) indicates that $E_{\text{rad}}$ does *not* take part in the Gauss constraint. This not only justifies the name "Coulombic" for $E_{\text{Coul}} = E - E_{\text{rad}}$, but most importantly prompts the following definition.

As I have argued earlier, in the presence of boundaries the Gauss constraint needs to be complemented by a boundary condition. The question is which boundary condition is the most

natural. This is where the second of the equations (28) plays a crucial role: demanding that $E_{\text{rad}}$ *completely* drops from the Gauss constraint—i.e. not only from its bulk term, but also from its boundary condition,—one is led to define the Gauss constraint in bounded regions as a boundary value problem *labelled* by the value of the electric flux, $f := s_i E^i_{|\partial R}$:

$$\mathsf{G}^f : \quad \begin{cases} \mathrm{D}_i E^i - \rho = 0 & \text{in } R, \\ s_i E^i - f = 0 & \text{at } \partial R. \end{cases} \tag{30}$$

Note that (28) for $E_{\text{rad}}$ is canonical and therefore so is the above extension of the Gauss constraint: i.e. neither equation depends on the choice of $\varpi$.

Introducing the radiative/Coulombic split of the electric field, $E = E_{\text{rad}} + E_{\text{Coul}}$, from (28) it follows that the burden of satisfying the Gauss constraint $\mathsf{G}^f$ falls completely on $E_{\text{Coul}}$:

$$\mathsf{G}^f : \quad \begin{cases} \mathrm{D}_i E^i_{\text{Coul}} - \rho = 0 & \text{in } R, \\ s_i E^i_{\text{Coul}} - f = 0 & \text{at } \partial R. \end{cases} \tag{31}$$

In Appendix A.2 I prove that the above boundary value problem uniquely fixes a $E_{\text{Coul}}$ as a function of $(A, \rho, f)$.[34] However, note that since the radiative/Coulombic *split* of $E$ depends on a choice of functional connection $\varpi$, so do the solutions of (31). Of course, all such solutions differ by a radiative contribution.

That $f$ is a datum completely independent from the values of $A$, $E_{\text{rad}}$, $\psi$ and $\overline{\psi}$, as well as from the geometry of $R$, is particularly clear for the SdW choice of connection, where $\mathsf{G}^f$ is yet another SdW boundary value problem which can always be inverted, no matter what the values of $\rho$ and $f$ are.[35] It is possible to show that the same holds true for any choice of connection by building on the SdW case, see Appendix A.2.

Configurations in $\Phi_{\mathsf{G}} := \bigcup_f \{\mathsf{G}^f = 0\}$ define the on-shell[36] subspace of $\Phi$. However, since $f$ is independent of any physical quantity contained in $R$, it is meaningful to further restrict attention from the set of on-shell configurations to a given superselection sector labelled by $f$. In order not to break (boundary) gauge-invariance down to gauge transformations that stabilize $f$, I introduce the *covariant superselection sector* (CSSS) $\Phi^{[f]}$:

$$\Phi^{[f]} := \bigcup_{f \in [f]} \{\phi \in \Phi | \mathsf{G}^f = 0\}, \tag{32}$$

where $[f] := \{f' \in \mathcal{C}^\infty(\partial R, \text{Lie}(\mathcal{G}))$ such that $\exists g \in \mathcal{G}$ for which $f' = g^{-1}_{|\partial R} f g_{|\partial R}\}$. Since gauge transformations act on the whole region $R$, this conjugacy class by definition does not include boundary gauge transformations that are not connected to the identity.[37]

The CSSS $\Phi^{[f]}$ is the arena in which the rest of this article will unfold.

# 4 Reduced symplectic structure in a CSSS

In this section I will construct the reduced symplectic structure in a CSSS. Before addressing the general case, I will quickly sketch the construction in the simpler Abelian case, highlighting the main difficulties characterizing its non-Abelian counterpart.

---

[34]Uniqueness of the solution holds only at irreducible configurations. Cf. footnotes 16 and 35.

[35]This statement holds true at irreducible configurations, which constitute a dense subset of configurations in non-Abelian YM theory. At reducible configurations, however, there are integral relations between these quantity in a number bounded by $\dim(G)$. In electromagnetism this is the integral Gauss law, $\int \sqrt{g}\rho = \oint \sqrt{h}f$. See also footnote 7 .

[36]In this article I ignore equations of motion. Thus "on-shell" means "on-shell of the Gauss constraint" only.

[37]For an elementary discussion, see [55].

**4.1 Overview of the Abelian case**  If $G$ is Abelian, the electric field is gauge invariant. Hence, a CSSS comprises one single flux, $[f] = \{f\}$, and thus reduces to the usual notion of superselection sector (SSS).

In the rest of this overview, equality within the SSS $\Phi^{\{f\}}$ will be denoted by "$\approx$."[38]

Being basic, $\Omega^H$ can be projected down to the reduced phase space. In the Abelian case and within a given SSS $\Phi^{\{f\}}$, the projection of $\Omega^H$ on the reduced SSS $\Phi^{\{f\}}/\mathcal{G}$ defines a 2-form which is not only closed but also non-degenerate. Therefore this construction equips the reduced SSS with a symplectic structure. But this symplectic structure has a drawback: it depends on the chosen notion of horizontality, i.e. on $\varpi$.

This dependence can be corrected by adding to $\Omega^H$ the term $\oint \sqrt{h}\mathrm{Tr}(f\mathbb{F})$. To see why this term is a viable addition to $\Omega^H$, observe that it is not only manifestly basic, but also exact and therefore closed. This follows from two facts. On the one hand the Abelian functional curvature is exact $\mathbb{F} = \mathrm{d}\varpi$ (7). On the other hand the Abelian superselection condition $[f] = \{f\}$ means that $\mathrm{d}f \approx 0$. Taken together, these two facts imply that in the given SSS, $\oint \sqrt{h}\mathrm{Tr}(f\mathbb{F}) \approx \mathrm{d}\oint \sqrt{h}\mathrm{Tr}(f\varpi)$. One can then check that not only $\Omega^H$ but also $\Omega^H + \oint \sqrt{h}\mathrm{Tr}(f\mathbb{F})$ defines, after projection, a symplectic structure on $\Phi^{\{f\}}/\mathcal{G}$.

Now, in the SSS $\Phi^{\{f\}}$, the 2-form $\Omega^H + \oint \sqrt{h}\mathrm{Tr}(f\mathbb{F})$ equals $\Omega$. Indeed, using (23) and the fact that in the Abelian SSS $\mathrm{d}f \approx 0$, one has

$$\Omega = \mathrm{d}\theta^H + \mathrm{d}\theta^V = \mathrm{d}\theta^H + \mathrm{d}\Big(-\int \sqrt{g}\,\mathsf{G}\varpi + \oint \sqrt{h}\,f\varpi\Big) \approx \Omega^H + \oint \sqrt{h}\mathrm{Tr}(f\mathbb{F}) \qquad \text{(Abelian)}, \quad (33)$$

where for brevity I denoted the bulk part of the Gauss constraint by $\mathsf{G} := \mathrm{D}_i E^i - \rho$. Therefore $\Omega$ itself is basic within the Abelian SSS $\Phi^{\{f\}}$. This can be checked explicitly and relies on the Gauss constraint $\mathsf{G}^f \approx 0$, the superselection condition $\mathrm{d}f \approx 0$, and the horizontality of the curvature $\mathbb{F} = \mathrm{d}\varpi$:

$$\begin{cases} \mathring{\mathbb{i}}_{\xi^\sharp}\Omega = -\int \sqrt{g}\,\xi\mathrm{d}\mathsf{G} + \oint \sqrt{h}\,\xi\mathrm{d}f \approx 0, \\[2mm] \mathbb{L}_{\xi^\sharp}\Omega = \mathrm{d}\mathbb{L}_{\xi^\sharp}\theta = \mathrm{d}\Big(-\int \sqrt{g}\,\mathsf{G}\mathrm{d}\xi + \oint \sqrt{h}\,f\mathrm{d}\xi\Big) \approx 0 \end{cases} \qquad \text{(Abelian)}. \qquad (34)$$

The first of these equations also shows that in a SSS the Hamiltonian charge with respect to $\Omega$ of any gauge transformation $\xi^\sharp$ (even field dependent ones!) vanishes[39] identically, even though $\mathring{\mathbb{i}}_{\xi^\sharp}\theta \not\approx 0$ and *also* $\mathrm{d}(\mathring{\mathbb{i}}_{\xi^\sharp}\theta) \not\approx 0$ if $\mathrm{d}\xi \neq 0$.

Since $\Omega$ is manifestly independent from $\varpi$, one concludes from the above equations that in the Abelian case it provides a *canonical* symplectic structure on the reduced SSS $\Phi^{\{f\}}/\mathcal{G}$, which is independent of the chosen notion of horizontality.

In the non-Abelian theory, matters are more complicated. There, not only $\mathrm{d}\varpi$ fails to be horizontal, $\mathbb{F}$ to be closed, and $[f]$ to be constituted by a single point (so that in a non-Abelian *Covariant*-SSS $\mathrm{d}f \not\approx 0$); but also $\Omega$ fails to be basic in a CSSS, and $\Omega^H$ fails to define a non-degenerate 2-form on the reduced CSSS. Therefore, any naive generalization of the above construction would fail in the non-Abelian theory.

The goal of the following discussion is to resolve these difficulties and provide a proper non-Abelian generalization. To achieve this goal, I will start by investigating the last of the differences listed above, i.e. why $\Omega^H$ fails to define a non-degenerate 2-form on the reduced phase space. Understanding this question will lead to the definition of flux rotations among other insights, and eventually to the sought construction of a symplectic structure on the reduced CSSS $\Phi^{[f]}/\mathcal{G}$ which is completely canonical.

---

[38]More precisely, $\approx$ stands for equality upon pullback to $\Phi^{\{f\}}$ understood as a sub-bundle of $\Phi$.

[39]Up to a field-space constant

**4.2   The projection of $\Omega^H$ in a CSSS**   Since the 2-form $\Omega^H$ is basic, it can be unambiguously projected down to the reduced phase space, and more specifically down to the reduced CSSS $\Phi^{[f]}/\mathcal{G}$.

The CSSS $\Phi^{[f]}$ is a submanifold of $\Phi$ foliated by the action of $\mathcal{G}$. Denote $\tilde{\pi} : \Phi^{[f]} \to \Phi^{[f]}/\mathcal{G}$ the projection on the space of gauge orbits and $\iota : \Phi^{[f]} \hookrightarrow \Phi$ the natural embedding. Note that $\iota$ acts as the identity map between the gauge orbits in $\Phi^{[f]}$ and $\Phi$. Note also that pulling back by $\iota^*$ means going on shell of the Gauss constraint (within a given CSSS).

Thus, the 2-form $\Omega^{\mathrm{red}}_\varpi \in \Omega^2(\Phi^{[f]}/\mathcal{G})$ stemming from the projection of $\iota^*\Omega^H$ is defined by the relation

$$\tilde{\pi}^*\Omega^{\mathrm{red}}_\varpi := \iota^*\Omega^H. \tag{35}$$

Once again, this definitioin is unambiguous because $\Omega^H$ and thus $\iota^*\Omega^H$ are basic.

Moreover, since $\mathbb{d}$ commutes with the pullback and since $\tilde{\pi}$ is surjective, one deduces from $\mathbb{d}\Omega^H = 0$ that also $\Omega^{\mathrm{red}}_\varpi$ is closed:

$$\mathbb{d}\Omega^{\mathrm{red}}_\varpi = 0. \tag{36}$$

Since $\Omega^{\mathrm{red}}_\varpi$ is closed, it defines a symplectic form on $\Phi^{[f]}/\mathcal{G}$ if and only if it is *non-degenerate*. In turn, being defined through the projection $\tilde{\pi} : \Phi^{[f]} \to \Phi^{[f]}/\mathcal{G}$, the 2-form $\Omega^{\mathrm{red}}_\varpi$ is non-degenerate if and only if the kernel of $\iota^*\Omega^H$ coincides with the space spanned by pure-gauge transformations, i.e. with $V^{[f]} = \mathrm{T}\big(\bigcup_{\phi \in \Phi^{[f]}} \mathcal{O}_\phi\big)$, where $\mathcal{O}_\phi$ is the gauge orbit of $\phi$. However, $\ker(\iota^*\Omega^H)$ does not generally coincide with $V^{[f]}$, unless $G$ is Abelian.

**4.3   The kernel of $\Omega^H$**   To see why $\ker(\iota^*\Omega^H)$ does not generally coinicide with $V^{[f]}$, and to compute it in the general case, one needs an explicit formula for $\Omega^H = \mathbb{d}\theta^H = \mathbb{d}_H\theta^H$:

$$\Omega^H = \int \sqrt{g}\,\mathrm{Tr}\big(\mathbb{d}_H E^i_{\mathrm{rad}} \wedge \mathbb{d}_H A_i\big) - \int \sqrt{g}\,\big(\mathbb{d}_H \overline{\psi}\gamma^0 \wedge \mathbb{d}_H \psi + \mathrm{Tr}(\rho\mathbb{F})\big) \in \Omega^2(\Phi). \tag{37}$$

Observe that only the radiative degrees of freedom and the "dressed" matter fields enter $\Omega^H$, whereas *no* component of the Coulombic electric field enters this formula.

Now, consider a vector $\mathbb{X} \in \mathrm{T}\Phi^{[f]}$, and denote $\eta = \varpi(\mathbb{X})$ its vertical part. Notice that, since $\varpi$ is defined by pullback from $\mathcal{A}$, $\eta$ is a function(al) of $\mathbb{X}(A)$ only. Then, computing $(\iota^*\Omega^H)(\mathbb{X})$ it is easy to see that $\mathbb{X} \in \ker(\iota^*\Omega^H)$ if and only if $\mathbb{X}(\bullet) = \eta^\sharp(\bullet)$ for $\bullet \in \{A, E_{\mathrm{rad}}, \psi, \overline{\psi}\}$, here seen as (coordinate) functions on $\Phi$. Since these conditions do not constrain the action of $\mathbb{X}$ on $E_{\mathrm{Coul}}$, this still leaves open the possibility that $\mathbb{X}$ is not purely vertical: i.e. it can still be that $\mathbb{X}(E_{\mathrm{Coul}}) \neq \eta^\sharp(E_{\mathrm{Coul}})$ and therefore $\mathbb{X} \neq \eta^\sharp$.

However, since $E_{\mathrm{Coul}}$ is fixed by the Gauss constraint (31), and $\mathbb{X}$ is assumed to preserve that constraint, the quantity $\mathbb{X}(E_{\mathrm{Coul}})$ can be determined by deriving equation (31) along $\mathbb{X}$. Denoting the action of $\mathbb{X}$ on $f$ by $\mathbb{X}(f) = -[f, \zeta'_\partial]$ for some field-dependent $\zeta'_\partial$ (recall that $\mathbb{X} \in \mathrm{T}\Phi^{[f]}$ and therefore preserves the CSSS), one can show that $\mathbb{X} \in \ker(\iota^*\Omega^H)$ if and only if $\mathbb{X} = \eta^\sharp + \mathbb{Y}_{\zeta_\partial}$, where $\mathbb{Y}_{\zeta_\partial}$ is defined by:

$$\mathbb{Y}_{\zeta_\partial}(\bullet) = 0 \quad \text{for } \bullet \in \{A, E_{\mathrm{rad}}, \psi, \overline{\psi}\}, \quad \text{and} \quad \begin{cases} \mathrm{D}_i \mathbb{Y}_{\zeta_\partial}(E^i_{\mathrm{Coul}}) = 0 & \text{in } R, \\ s_i \mathbb{Y}_{\zeta_\partial}(E^i_{\mathrm{Coul}}) = -[f, \zeta_\partial] & \text{at } \partial R, \\ \int \sqrt{g}\,\mathrm{Tr}\big(\mathbb{Y}_{\zeta_\partial}(E^i_{\mathrm{Coul}})\mathbb{d}_H A_i\big) = 0, \end{cases} \tag{38}$$

for $\zeta_\partial = \zeta'_\partial - \eta_{|\partial R}$. Notice that the last equation just states that $\mathbb{Y}_{\zeta_\partial}(E_{\mathrm{Coul}})$ is itself Coulombic. Therefore these equations have a unique solution for the very same reason that the Gauss constraint (31) does.

E.g. in the SdW case, that last equation states that $\mathbb{Y}_{\zeta_\partial}(E_{\text{Coul}}) = D\zeta$ for some Lie($\mathcal{G}$)-valued function $\zeta$. Using this fact, from the remaining equations one deduces that

$$\mathbb{Y}_{\zeta_\partial}^{(\text{SdW})} = \int \zeta \frac{\delta}{\delta\varphi} \qquad \text{for} \qquad \begin{cases} D^2\zeta = 0 & \text{in } R, \\ D_s\zeta = -[f, \zeta_\partial] & \text{at } \partial R \end{cases} \qquad (\text{SdW}). \qquad (39)$$

I will call *flux rotations* vectors of the form (38); I will denote the space spanned by these vectors $Y^{[f]} \subset T\Phi^{[f]}$. I have thus argued that the kernel of $\iota^*\Omega^H$ is composed of vertical vectors *and* flux rotations, i.e.

$$\ker(\iota^*\Omega^H) = V^{[f]} \oplus Y^{[f]}. \qquad (40)$$

Notice that since $\mathbb{Y}_{\zeta_\partial}(A) = 0$, flux rotations are *horizontal*, i.e. $\varpi(\mathbb{Y}_{\zeta_\partial}) = 0$ (once again, on $\Phi$, the connection $\varpi$ is defined by pullback from $\mathcal{A}$).

A more thorough proof of these statements can be found in Appendix A.3.

Henceforth, with a slight abuse of notation, I will also call "flux rotations" vector fields which arise as sections of $Y^{[f]} \subset T\Phi^{[f]}$; I will denote these vector fields $\mathbb{Y}_{\zeta_\partial}$ as well, and their space

$$\mathcal{Y}^{[f]} := \Gamma(\Phi, Y^{[f]}) \subset \mathfrak{X}^1(\Phi^{[f]}). \qquad (41)$$

Notice that this definition allows the parameter $\zeta_\partial$ in the vector field $\mathbb{Y}_{\zeta_\partial}$ to be field-dependent itself.

**4.4 Flux rotations are physical transformations** Flux rotations $\mathbb{Y}_{\zeta_\partial}$ affect the electric flux $f$ precisely as a gauge transformation $\xi$ with $\xi_{|\partial R} = -\zeta_\partial$ would. However, since they leave all other field components invariant, flux rotations are *not* gauge transformations. In fact, through the Gauss constraint (which is by definition imposed in a CSSS), they alter the bulk Coulombic field $E_{\text{Coul}}$ and thus physical observables such as the total YM energy in $R$. This establishes the physical nature of flux rotations.

Begin physical, flux rotations must survive the projection onto the reduced CSSS $\Phi^{[f]}/\mathcal{G}$. However, as vector fields not all flux rotations are projectable. Flux rotations which are projectable must be gauge invariant, i.e. $\mathbb{Y}_{\zeta_\partial}$ is projectable if and only if $[\![\mathbb{Y}_{\zeta_\partial}, \xi^\sharp]\!] = 0$ for all $\mathbb{d}\xi = 0$. It is easy to verify that this condition holds if and only if the parameter $\zeta_\partial$ transforms covariantly, i.e. if and only if

$$\mathbb{L}_{\xi^\sharp}\zeta_\partial = [\zeta_\partial, \xi_{|\partial R}]. \qquad (42)$$

I will denote the set of covariant flux rotations $\mathcal{Y}_{\text{cov}}^{[f]}$.

Let me emphasize that the definition of flux rotations depends on the choice of $\varpi$, i.e. $\mathbb{Y}_{\zeta_\partial} \equiv \mathbb{Y}_{\zeta_\partial}^{(\varpi)}$: any choice of $\varpi$ produces a distinct $\mathbb{Y}_{\zeta_\partial}^{(\varpi)}$ corresponding to a distinct physical transformation in the same phase space; each of these transformations affect slightly different components of the electric field while preserving the validity of the Gauss constraint. Choosing the SdW radiative/Coulombic split of the electric field, $E = E_{\text{rad}}^{(\text{SdW})} + D\varphi$ as providing a natural choice of coordinates over field space, one finds that whereas the SdW Coulombic potential of $\varphi$ depends only on the parameter $\zeta_\partial$, the change of the SdW-radiative part of $E$ is $\varpi$-dependent. That is, whereas $\mathbb{Y}_{\zeta_\partial}^{(\varpi)}(\varphi) = \mathbb{Y}_{\zeta_\partial}^{(\text{SdW})}(\varphi)$ as in (A3) for any choice of $\varpi$, one finds that $\mathbb{Y}_{\zeta_\partial}^{(\varpi)}(E_{\text{rad}}^{(\text{SdW})}) = 0$ if and only if $\varpi = \varpi_{\text{SdW}}$. Therefore, the transformations $\mathbb{Y}_{\zeta_\partial}^{(\varpi)}$ for different choices of $\varpi$ are all equally physical, but distinct from each other.

**4.5 The kernel of $\Omega_\varpi^{\text{red}}$** From (35), (40), and the previous discussion on flux rotations, one concludes that

$$\ker(\Omega_\varpi^{\text{red}}) = \tilde{\pi}_* Y^{[f]}. \qquad (43)$$

Therefore, $\Omega_\varpi^{\mathrm{red}}$ is non-degenerate if and only if $Y^{[f]}$ is trivial. Inspection of (38), shows that this is the case if $G$ is Abelian or if the flux is trivial (either because $f = 0$ or because $\partial R = \emptyset$). Therefore if $G$ is Abelian, or $f$ is trivial, the reduced space $(\Phi^{[f]}/\mathcal{G}, \Omega_\varpi^{\mathrm{red}})$ is symplectic. In these cases, the reduction procedure could be considered complete. However, unless $f$ is trivial, the resulting symplectic structure fails to be canonical: it depends on the *choice* of $\varpi$.

In the non-Abelian theory, if boundaries are present, $\Omega_\varpi^{\mathrm{red}}$ even fails to provide a symplectic structure for the reduce: $\Omega_\varpi^{\mathrm{red}}$ is degenerate with flux rotations constituting its nontrivial kernel.

### 4.6 Completing $\Omega_\varpi^{\mathbf{red}}$

In the non-Abelian case, the issue with $\Omega_\varpi^{\mathrm{red}}$ is that, although there are different $f$'s in $[f]$ each potentially associated with a different Coulombic field, the candidate symplectic 2-form $\Omega_\varpi^{\mathrm{red}}$ cannot tell them apart from each other. This leaves flux rotations as degenerate directions of $\Omega_\varpi^{\mathrm{red}}$. To correct this issue, I propose to complete $\Omega_\varpi^{\mathrm{red}}$ by adding to it a symplectic form on $[f]$, the space of fluxes belonging to a given CSSS.

The remarkable aspect is that this completion, despite *not* being provided by the off-shell symplectic structure $\Omega$, can still be chosen canonically. This canonical choice is provided by the Kirillov–Konstant–Souriau (KKS) construction of the homogeneous symplectic structure over a (co)adjoint orbit (see e.g. [56]). Although this symplectic structure fails to be $\varpi$-horizontal, I will show this issue can be easily rectified.

Regarding the fact that the KKS symplectic structure is canonical on *co*adjoint orbits, note that electric fields—and thus fluxes—are dual to the gauge potential $A$ and therefore are best understood as valued in the dual $\mathrm{Lie}(G)^*$ (as a vector space). The identification is performed via the Killing form, e.g. $f \leftrightarrow f^* = \mathrm{Tr}(f \cdot)$. Although I will not use this notation in the following, this observation further justifies the naturalness of the use of the KKS construction.

### 4.7 Review of the KKS symplectic structure on $[f]$

To provide an explicit formula for the KKS symplectic form on $[f]$, let me first choose a reference flux $f_o \in [f]$ and thus over-parametrize $[f]$ by group-valued variables $u \in \mathcal{G}_{|\partial R}$:

$$f = u f_o u^{-1}. \tag{44}$$

From this formula it follows that

$$[f] \cong \mathcal{G}_{|\partial R}/\mathcal{G}_{|\partial R}^o \tag{45}$$

as a right-quotient, where $\mathcal{G}_{|\partial R}^o \subset \mathcal{G}_{|\partial R}$ is the subset of $\mathcal{G}_{|\partial R}$ which stabilizes $f_o$. In other words, right translations of $u$ by a stabilizer transformations $g_o \in \mathcal{G}_{|\partial R}^o$, $u \mapsto u g_o$, end up stabilizing the reference $f_o$ and therefore have no effect on $f$. Therefore, these transformations are a redundancy in the description of $[f]$ in terms of the group-valued $u \in \mathcal{G}_{|\partial R}$: the goal is to eventually quotient them away.

To be clear, the notation $\mathcal{G}_{|\partial R}$ stands for $\mathcal{G}_{|\partial R} = \{u \in C^\infty(\partial R, G) \text{ such that } \exists g \in \mathcal{G} \text{ for which } u = g_{|\partial R}\}$, that is the group $\mathcal{G}_{|\partial R}$ coincides with the subset of boundary gauge transformations $C^\infty(\partial R, G)$ which are connected to the identity. In other words $[f]$ is the *connected* part of the adjoint orbit of $f$ by the adjoint action of $\mathcal{G}^\partial = C^\infty(\partial R, G)$.[40]

Let me stress that $f_o \in \mathrm{Lie}(\mathcal{G}_{|\partial R})$ is a mere reference: all flux dof are completely—and redundantly—encoded in the group elements $u \in \mathcal{G}_{|\partial R}$. Hence, $\mathbb{d}f_o \equiv 0$ *always*.

To explicitly construct the (otherwise canonically given) KKS symplectic structure on $[f]$, I will first endow $\mathcal{G}_{|\partial R}$ with a presymplectic 2-form which I will then project down to a symplectic 2-form on $\mathcal{G}_{|\partial R}^o/\mathcal{G}_{|\partial R} \cong [f]$. With this goal in mind, it is useful to consider $\mathcal{G}_{|\partial R}$ as a field-space for the group value fields $u$ which is foliated by the right action of the stabilizer transformations

---

[40]Cf. footnote 37.

$g_o \in \mathcal{G}^o_{|\partial R}$, and to define the projection $\pi_o$ onto the associated space of orbits $\mathcal{G}_{|\partial R}/\mathcal{G}^o_{|\partial R} \cong [f]$:

$$\pi_o : \mathcal{G}_{|\partial R} \to \mathcal{G}_{|\partial R}/\mathcal{G}^o_{|\partial R} \cong [f], \qquad u \mapsto f := u f_o u^{-1}. \tag{46}$$

Thus, on $\mathcal{G}_{|\partial R}$, define the presymplectic potential $\vartheta^{[f]} := \oint \sqrt{h}\,\mathrm{Tr}(f_o u^{-1}\mathbb{d}u)$ and the corresponding presymplectic form

$$\omega^{[f]} := \mathbb{d}\vartheta^{[f]} = -\oint \sqrt{h}\,\mathrm{Tr}\big(\tfrac{1}{2}f_o[u^{-1}\mathbb{d}u \overset{\wedge}{,} u^{-1}\mathbb{d}u]\big). \tag{47}$$

This 2-form is basic with respect to the stabilizer transformations of the reference flux $f_o$, i.e. it is basic in $(\mathcal{G}_{|\partial R}, \pi_o)$.[41] Therefore, $\omega^{[f]}$ can be projected down to $\mathcal{G}_{|\partial R}/\mathcal{G}^o_{|\partial R}$ along $\pi_o$. This projection defines the 2-form $\omega^{[f]}_{\mathrm{KKS}} \in \Omega^2(\mathcal{G}_{|\partial R}/\mathcal{G}^o_{|\partial R})$ through the relation

$$\pi_o^* \omega^{[f]}_{\mathrm{KKS}} := \omega^{[f]}. \tag{48}$$

The 2-form $\omega^{[f]}_{\mathrm{KKS}}$ is non degenerate, and therefore symplectic. Moreover, although the construction of the bundle $(\mathcal{G}_{|\partial R}, \pi_o)$ depends on the choice of reference $f_o$, $\omega^{[f]}_{\mathrm{KKS}}$ is homogeneous over $[f] \cong \mathcal{G}_{|\partial R}/\mathcal{G}^o_{|\partial R}$, and thus independent of the choice of the reference $f_o$. The 2-form $\omega^{[f]}_{\mathrm{KKS}}$ is precisely the canonical KKS symplectic form on $[f]$.[42]

If $G$ is Abelian, the electric field is gauge invariant and $[f] = \{f_o\}$ reduces to one (functional) point that cannot support a 2-form. Consistently, the presymplectic 2-form $\omega^{[f]}$ vanishes in this case, and therefore so does $\omega^{[f]}_{\mathrm{KKS}}$. Interestingly, however, the presymplectic potential $\vartheta^{[f]}$ *fails* to vanish, even in the Abelian case—a fact that will play a role later.

Thinking of $[f]$ as embedded in the CSSS $\Phi^{[f]}$, one can pull-back $\omega^{[f]}_{\mathrm{KKS}}$ along this embedding from $[f]$ to $\Phi^{[f]}$ thus giving a 2-form that I will denote by the same symbol.

But seen as 2-form on $\Phi^{[f]}$, $\omega^{[f]}_{\mathrm{KKS}}$ fails to basic with respect to the action of *gauge* transformations on $\Phi^{[f]}$. This menas that, as it is, $\omega^{[f]}_{\mathrm{KKS}}$ cannot be projected down to the reduced CSSS $\Phi^{[f]}/\mathcal{G}$ and therefore cannot be used to complete $\Omega^{\mathrm{red}}_{\varpi}$ to a symplectic form. This issue can be solved by defining a gauge-horizontal version of $\omega^{[f]}_{\mathrm{KKS}}$.

**4.8 A gauge-horizontal KKS 2-form** In view of the over-parametrization of $[f] \cong \mathcal{G}_{|\partial R}/\mathcal{G}^o_{|\partial R}$ by $u \in \mathcal{G}_{|\partial R}$, I will *temporarily* extend the field space $\Phi^{[f]}$ to $\Phi^{[f]}_{\mathrm{ext}}$ by replacing the flux dof $f \in [f_o]$ with the group-valued dof $u \in \mathcal{G}_{|\partial R}$. This extension parallels the construction of the previous paragraph, from which I will borrow the notation.

Thus, consider a fibre bundle which has $\Phi^{[f]}$ as a base manifold and $\mathcal{G}^o_{|\partial R}$ as a fibre:

$$\pi_o : \Phi^{[f]}_{\mathrm{ext}} \to \Phi^{[f]} = \Phi^{[f]}_{\mathrm{ext}}/\mathcal{G}^o_{|\partial R}, \quad (A, E_{\mathrm{rad}}, \psi, \overline{\psi}, u) \mapsto (A, E_{\mathrm{rad}}, \psi, \overline{\psi}, f = u f_o u^{-1}). \tag{49}$$

The fibre-generating symmetries on $(\Phi^{[f]}_{\mathrm{ext}}, \pi_o)$ are given by the stabilizer transformations of $f_o$, which act on $u$ from the right while leaving all other fields invariant: $(A, E_{\mathrm{rad}}, \psi, \overline{\psi}, u) \mapsto (A, E_{\mathrm{rad}}, \psi, \overline{\psi}, u g_o)$. Therefore, an infinitesimal stabilizer transformation $\sigma_o \in \mathrm{Lie}(\mathcal{G}^o_{|\partial R})$ defines a vector field on $\Phi^{[f]}_{\mathrm{ext}}$. In analogy with the map $\cdot^\sharp$, introduce

$$\cdot^\S : \mathrm{Lie}(\mathcal{G}^o_{|\partial R}) \to \mathfrak{X}^1(\Phi^{[f]}_{\mathrm{ext}}), \qquad \sigma_o \mapsto \sigma_o^\S \tag{50}$$

---

[41]On the contrary, the presymplectic potential $\vartheta^{[f]}$ *fails* to be basic in $(\mathcal{G}_{|\partial R}, \pi_o)$.

[42]E.g. in the finite dimensional example $\mathcal{G}_{|\partial R} \rightsquigarrow SU(2)$, the (co)adjoint orbit of a non-vanishing element of the (dual of the) Lie algebra $\mathrm{Lie}(SU(2))$ is a 2-sphere, whereas the corresponding KKS symplectic form is, up to a scale, the area element of the round 2-sphere.

so that (here, as above, $\bullet \in \{A, E_{\mathrm{rad}}, \psi, \overline{\psi}\}$)

$$\sigma_o^{\S}(\bullet) := 0 \qquad \text{and} \qquad \sigma_o^{\S}(u) := u\sigma_o. \tag{51}$$

It is straightforward to extend the action of gauge symmetries and flux rotations from $\Phi^{[f]}$ to $\Phi_{\mathrm{ext}}^{[f]}$. Indeed, it is enough to prescribe their action on the $u$'s so that the ensuing action on $f$ is the known one. For this, notice that gauge transformation and flux rotations act identically on $f$ (up to a sign), that is $\mathbb{L}_{\xi^{\sharp}} f = [f, \xi_{|\partial R}]$ and $\mathbb{L}_{\mathbb{Y}_{\zeta_{\partial}}} f = -[f, \zeta_{\partial}]$ respectively.[43] From these, it is natural to prescribe that both gauge transformations and flux rotations act on $u$ from the left:

$$\mathbb{L}_{\xi^{\sharp}} u = -\xi_{|\partial} u \qquad \text{and} \qquad \mathbb{L}_{\mathbb{Y}_{\zeta_{\partial}}} u = \zeta_{\partial} u. \tag{52}$$

Hence, $\Phi_{\mathrm{ext}}^{[f]}$ is also foliated by gauge transformations.

Combining the action of gauge transformations $\mathcal{G}$ and stabilizer transformations $\mathcal{G}_{|\partial R}^o$, one obtains an action of the group $\mathcal{G}_{\mathrm{ext}} := \mathcal{G} \times \mathcal{G}_{|\partial R}^o$ on $\Phi_{\mathrm{ext}}^{[f]}$ and a projection $\Pi$ to the space of the $\mathcal{G}_{\mathrm{ext}}$-orbits in $\Phi_{\mathrm{ext}}^{[f]}$—which is nothing else than the reduced CSSS. In formulas:

$$(\mathcal{G} \times \mathcal{G}_{|\partial R}^o) \times \Phi_{\mathrm{ext}}^{[f]} \to \Phi_{\mathrm{ext}}^{[f]}$$
$$\big((g, g_o), (\bullet, u)\big) \mapsto (\bullet, u)^{(g, g_o)} = (\bullet^g, g_{|\partial R}^{-1} u g_o) \tag{53}$$

and

$$\Pi : \Phi_{\mathrm{ext}}^{[f]} \to \Phi_{\mathrm{ext}}^{[f]} / (\mathcal{G} \times \mathcal{G}_{|\partial R}^o) = \Phi^{[f]} / \mathcal{G}, \tag{54}$$

where $\bullet = (A, E_{\mathrm{rad}}, \psi, \overline{\psi})$ with the usual $A^g = g^{-1} A g + g^{-1} \mathrm{d}g$, $E_{\mathrm{rad}}^g = g^{-1} E_{\mathrm{rad}} g$, $\psi^g = g^{-1}\psi$ and $\overline{\psi}^g = \overline{\psi} g$.

The space of $\Pi$-vertical vectors in $\mathrm{T}\Phi_{\mathrm{ext}}^{[f]}$—defined as the space vectors tangent to the orbits of $\mathcal{G}_{\mathrm{ext}}$—is thus given by:

$$V_{\mathrm{ext}} = \mathrm{Span}\big\{(\xi^{\sharp}, \sigma_o^{\S})\big\}. \tag{55}$$

Using the functional connection $\varpi$ over $(\Phi^{[f]}, \tilde{\pi})$, one defines through a pull-back by $\pi_o^*$ a functional connection over $(\Phi_{\mathrm{ext}}^{[f]}, \Pi)$—which I will still denote $\varpi$. Because of the gauge-transformation property of $u$ (52), one has

$$\mathbb{d}_H u = \mathbb{d}u + \varpi_{|\partial R} u. \tag{56}$$

With these tools and notations, one can finally define on $\Phi_{\mathrm{ext}}^{[f]}$ the horizontal version of the KKS potential $\vartheta^{[f]}$, that is:

$$\vartheta_{\mathrm{ext}}^{H,[f]} := \oint \sqrt{h}\,\mathrm{Tr}(f_o u^{-1} \mathbb{d}_H u) \in \Omega^1(\Phi_{\mathrm{ext}}^{[f]}). \tag{57}$$

This 1-form is basic with respect to the action of gauge transformations.[44] Therefore, defining $\omega_{\mathrm{ext}}^{H,[f]} := \mathbb{d}\vartheta_{\mathrm{ext}}^{H,[f]}$ one obtains a gauge-basic and closed 2-form on $\Phi_{\mathrm{ext}}^{[f]}$, i.e.

$$\mathbb{d}\omega_{\mathrm{ext}}^{H,[f]} = 0, \qquad \mathring{\mathbb{i}}_{\xi^{\sharp}} \omega_{\mathrm{ext}}^{H,[f]} = 0, \qquad \text{and} \qquad \mathbb{L}_{\xi^{\sharp}} \omega_{\mathrm{ext}}^{H,[f]} = 0. \tag{58}$$

Explicitly, from $\omega_{\mathrm{ext}}^{H,[f]} = \mathbb{d}\vartheta_{\mathrm{ext}}^{H,[f]} = \mathbb{d}_H \vartheta_{\mathrm{ext}}^{H,[f]}$ (which holds because $\vartheta_{\mathrm{ext}}^{H,[f]}$ is basic) and $\mathbb{d}_H^2 u = \mathbb{F}u$,

$$\omega_{\mathrm{ext}}^{H,[f]} = \oint \sqrt{h}\,\mathrm{Tr}\Big(-\tfrac{1}{2} f_o [u^{-1} \mathbb{d}_H u \overset{\wedge}{,} u^{-1} \mathbb{d}_H u] + f_o u^{-1} \mathbb{F}u\Big) \in \Omega^2(\Phi_{\mathrm{ext}}^{[f]}). \tag{59}$$

---

[43]Recall, gauge transformations and flux rotations have (very) distinct actions on the other fields in $\Phi^{[f]}$.

[44]However, $\vartheta_{\mathrm{ext}}^{H,[f]}$ *fails* to be basic with respect to the *whole* structure group $\mathcal{G}_{\mathrm{ext}}$, because it fails to be basic with respect to the action of the stabilizer $\mathcal{G}_{|\partial R}^o$. Cf. footnote 41.

Moreover, as it was the case for $\omega^{[f]}$, the 2-form $\omega_{\text{ext}}^{H,[f]}$ is also basic with respect to the flux-reference stabilizer transformations, and therefore it is basic in $(\Phi_{\text{ext}}^{[f]}, \Pi)$:[45]

$$\mathring{\mathbb{i}}_{(\xi^\sharp, \sigma_o^\S)} \omega_{\text{ext}}^{H,[f]} = 0 \qquad \text{and} \qquad \mathbb{L}_{(\xi^\sharp, \sigma_o^\S)} \omega_{\text{ext}}^{H,[f]} = 0. \tag{60}$$

Because of this property, $\omega_{\text{ext}}^{H,[f]}$ can be projected not only down to $\Phi^{[f]}$—where it gives a gauge-horizontal version of the KKS symplectic structure $\omega_{\text{KKS}}^{[f]}$—but also down to the gauge-reduced CSSS $\Phi_{\text{ext}}^{[f]}/(\mathcal{G} \times \mathcal{G}_{|\partial R}^o) = \Phi^{[f]}/\mathcal{G}$.

Before using this fact to finally introduce the sought completion of $\Omega_\varpi^{\text{red}}$ which will turn the reduced CSSS $\Phi^{[f]}/\mathcal{G}$ into a symplectic space, let me conclude this section with an observation.

**4.9  The completion of $\Omega_\varpi^{\text{red}}$**  To define the sought symplectic completion of $\Omega_\varpi^{\text{red}}$, first define in $(\Phi_{\text{ext}}^{[f]}, \Pi)$ the presymplectic completion of $\Omega^H$ by $\omega^{H,[f]}$ as

$$\Omega_{\text{ext}}^{H,[f]} := \pi_o^* \iota^* \Omega^H + \omega_{\text{ext}}^{H,[f]} \in \Omega^2(\Phi_{\text{ext}}^{[f]}). \tag{61}$$

In coordinates, this reads:

$$\Omega_{\text{ext}}^{H,[f]} = \int \sqrt{g}\, \text{Tr}\big(\mathbb{d}_H E_{\text{rad}}^i \curlywedge \mathbb{d}_H A_i\big) - \int \sqrt{g}\, \big(\mathbb{d}_H \overline{\psi} \gamma^0 \curlywedge \mathbb{d}_H \psi + \text{Tr}(\rho\, \mathbb{F})\big)$$
$$+ \oint \sqrt{h}\, \text{Tr}\big(-\tfrac{1}{2} f_o [u^{-1} \mathbb{d}_H u \curlywedge u^{-1} \mathbb{d}_H u] + f_o u^{-1} \mathbb{F} u\big). \tag{62}$$

This form is closed, basic, and its kernel can be shown to comprise vertical vectors only (now, the KKS contribution takes care of the flux rotations in the kernel of $\pi_o^* \iota^* \Omega^H$; see Appendix A.4 for a proof):

$$\ker(\Omega_{\text{ext}}^{H,[f]}) = V_{\text{ext}}. \tag{63}$$

Being basic, this 2-form, can be projected down to the reduced CSSS $\Phi^{[f]}/\mathcal{G}$ to give the 2-form $\Omega^{\text{red},[f]} \in \Omega^2(\Phi^{[f]}/\mathcal{G})$, defined through the relation

$$\Pi^* \Omega^{\text{red},[f]} := \Omega_{\text{ext}}^{H,[f]}. \tag{64}$$

Finally, thanks to the above-mentioned properties of $\Omega^{H,[f]}$, the 2-form $\Omega^{\text{red},[f]}$ is also closed and non-degenerate, and thus

$$\text{the reduced CSSS } (\Phi^{[f]}/\mathcal{G}, \Omega^{\text{red},[f]}) \text{ is symplectic.} \tag{65}$$

(As explained in footnote 16, I have been neglecting *reducible* configurations—cf. appendix A.2. Here, let me only mention that in electromagnetism, where all configurations are reducible, the above equation needs to be corrected by excluding from $V^{[f]}$ the 1-dimensional space of spatially constant "gauge transformations." This fact is relevant because these transformations—related to the *total* electric charge—are then promoted to physical transformations in the reduced field space. The situation is much more complicated, and less clear-cut, in non-Abelian theories, cf. the forthcoming v3 of [5].)

---

[45]Since $\varpi$ is pulled-back from $\Phi^{[f]}$, one has $\mathring{\mathbb{i}}_{\sigma_o^\S} \varpi = 0 = \mathbb{L}_{\sigma_o^\S} \varpi$.

**4.10  Independence from $\varpi$**  Although it is not manifest from e.g. (62), the presymplectic 2-form $\Omega_{\text{ext}}^{H,[f]}$ is independent of $\varpi$, and therefore so is the symplectic form $\Omega^{\text{red},[f]}$ on the reduced CSSS $\Phi^{[f]}/\mathcal{G}$. The reduced symplectic structure $\Omega^{\text{red},[f]}$ on $\Phi^{[f]}/\mathcal{G}$ is indeed completely canonical.

To show this—and highlight the role of the KKS symplectic structure—it is convenient to perform the reduction by $\Pi$ of $(\Phi_{\text{ext}}^{[f]}, \Omega_{\text{ext}}^{H,[f]})$ in two steps: first to $(\Phi^{[f]}, \Omega^{H,[f]})$ by projecting out the stabilizer transformations, and then to $(\Phi^{[f]}, \Omega^{\text{red},[f]})$ by projecting out gauge transformations.

Thus, let me backtrack to the definition of $\vartheta^{H,[f]}$ (57). Using $\text{d}_H u = \text{d}u + \varpi_{|\partial R} u$ and the fact that within a CSSS $\iota^*\theta^V = \oint \sqrt{h}\,\text{Tr}(f\varpi)$ (23), this 1-form can be written as

$$\vartheta^{H,[f]} = \vartheta^{[f]} + \pi_o^*\iota^*\theta^V \in \Omega^1(\Phi_{\text{ext}}^{[f]}). \tag{66}$$

Differentiating, one finds

$$\omega^{H,[f]} = \text{d}\vartheta^{H,[f]} = \omega^{[f]} + \pi_o^*\iota^*\text{d}\theta^V \in \Omega^2(\Phi_{\text{ext}}^{[f]}) \tag{67}$$

where $\omega^{[f]}$ is precisely the KKS presymplectic form (47). Note that the rightmost term in this equation explains why $\omega^{H,[f]}$ does not identically vanish in the Abelian case, even though $\omega^{[f]}$ does: in this case it is easy to check that $\omega^{H,[f]} = \pi_o^*\iota^*\text{d}\theta^V = \pi_o^*\iota^* \oint \sqrt{h}\,f_o\mathbb{F}$, in agreement with the overview of paragraph 4.1.

Although (in the non-Abelian theory) only the *sum* of $\omega^{[f]}$ and $\pi_o^*\iota^*\text{d}\theta^V$ is basic with respect to gauge transformations, each term is individually basic with respect to stabilizer transformations of the reference flux $f_o$. Therefore, each term can be independently projected down to $\Phi^{[f]}$ along $\pi_o$. In particular, $\omega^{[f]}$ projects to the KKS 2-form $\omega_{\text{KKS}}^{[f]} \in \Omega^2(\Phi^{[f]})$ (48).

From equations (48) and (67), one readily sees that in the definition of the presymplectic 2-form $\Omega_{\text{ext}}^{H,[f]}$ over $(\Phi_{\text{ext}}^{[f]}, \Pi)$ (61), the horizontal and vertical parts of the off-shell symplectic structure combine as in $\Omega = \Omega^H + \text{d}\theta^V$ (see (23)), thus yielding

$$\Omega_{\text{ext}}^{H,[f]} = \pi_o^*\iota^*\Omega + \omega^{[f]} \in \Omega^2(\Phi_{\text{ext}}^{[f]}),. \tag{68}$$

Then, subsequent reductions along $\Phi_{\text{ext}}^{[f]} \xrightarrow{\pi_o} \Phi^{[f]} \xrightarrow{\tilde{\pi}} \Phi^{[f]}/\mathcal{G}$ lead first to a gauge-basic 2-form on $\Phi^{[f]} = \pi_o(\Phi_{\text{ext}}^{[f]})$

$$\Omega^{H,[f]} := \iota^*\Omega + \omega_{\text{KKS}}^{[f]} \in \Omega^2(\Phi^{[f]}), \tag{69}$$

and finally to the fully reduced symplectic 2-form $\Omega^{\text{red},[f]}$ on $\Phi^{[f]}/\mathcal{G} = \tilde{\pi}(\Phi^{[f]})$.

Because not only $\omega^{[f]}$ and $\omega_{\text{KKS}}^{[f]}$, but also $\Omega$ and the maps $\iota$ and $\pi_o$ are independent of $\varpi$, so must be the expressions (68) and (69), and therefore the reduced symplectic space:

$$(\Phi^{[f]}/\mathcal{G}, \Omega^{\text{red},[f]}) \text{ is independent from the choice of } \varpi. \tag{70}$$

**4.11  The reduced symplectic structure $(\Phi^{[f]}/\mathcal{G}, \Omega^{\text{red},[f]})$ is canonical**  An important aspect of equations (68) and (69) which express $\Omega_{\text{ext}}^{H,[f]}$ and $\Omega^{H,[f]}$ in terms of $\Omega$—as opposed to equation (61) which expresses $\Omega_{\text{ext}}^{H,[f]}$ in terms of $\Omega^H$,—is that only the *sum* of the 2-forms appearing on their right-hand side defines a 2-form which is basic with respect to gauge transformations.

The only exceptions to this statement arise when $\omega_{\text{KKS}}^{[f]}$ vanishes, that is when the flux is trivial (either because $f = 0$ or $\partial R = \emptyset$) or when $G$ is Abelian. This explains why the Abelian case is so much simpler.

In the Abelian case, comparison with the result of paragraph 4.5 shows that there are multiple symplectic structures available on the reduced superselection sector $\Phi^{\{f_o\}}/\mathcal{G}$: there is

the symplectic structure $\Omega^{\mathrm{red},\{f_o\}}$, but also the family of structures $\Omega^{\mathrm{red}}_{\varpi}$ for each $\varpi$. However, as the notation suggests, only $\Omega^{\mathrm{red},\{f_o\}}$ is independent of $\varpi$—cf. paragraph 4.1.

In the non-Abelian case none of the 2-forms $\Omega^{\mathrm{red}}_{\varpi}$ is non-degenerate due to the nontrivial nature of flux-rotations; hence, the only available *symplectic* form over the reduced CSSS $\Phi^{[f]}/\mathcal{G}$ is $\Omega^{\mathrm{red},[f]}$, which also happens to be *independent* of $\varpi$. Therefore the completion of $\Omega^{\mathrm{red}}_{\varpi}$ through the addition of a (gauge-horizontal) KKS symplectic structure on the space of fluxes, not only cures the kernel of $\iota^*\Omega^H$ but also its dependence on $\varpi$.

Indeed, the reduced symplectic structure $\Omega^{\mathrm{red},[f]}$ is fully canonical: its construction does not depend on any external choice or input, even the KKS form is canonically given on $[f]$. In sum, once the focus is set on *covariant* superselection sectors, the form of $\Omega^{\mathrm{red},[f]}$ is enforced upon us by the resulting geometry of field space and by the existence of flux rotations.

**4.12  Flux rotations symmetries**  What is the fate of the projectable flux transformations $\mathcal{Y}^{[f]}_{\mathrm{cov}}$ in relation to the canonical symplectic structure $\Omega^{\mathrm{red},[f]}$?

Consider a flux rotation $\mathbb{Y}_{\zeta_\partial} \in \mathcal{Y}^{[f]}_{\mathrm{cov}}$ viewed, as described above, as a horizontal vector field on $\Phi^{[f]}_{\mathrm{ext}}$. Recall that a flux rotation $\mathbb{Y}_{\zeta_\partial}$ is projectable down to $\Phi^{[f]}_{\mathrm{ext}}/\mathcal{G}_{\mathrm{ext}} = \Phi^{[f]}/\mathcal{G}$ if and only if it is "covariant" i.e. if only if the label $\zeta_\partial$ has a field-dependence that makes it change covariantly along the gauge directions: $\mathbb{L}_{\xi^\sharp}\zeta_\partial = [\zeta_\partial, \xi_{|\partial R}]$. Denote its projection $\tilde{\mathbb{Y}}_{\zeta_\partial} := \Pi_* \mathbb{Y}_{\zeta_\partial} \in \mathfrak{X}^1(\Phi^{[f]}/\mathcal{G})$.

Now, for a covariant parameter $\zeta_\partial$, define

$$H_{\zeta_\partial} := -\oint \sqrt{h}\, \mathrm{Tr}(f_o u^{-1} \zeta_\partial u) \in \Omega^0(\Phi^{[f]}/\mathcal{G}) \tag{71}$$

and note that this quantity is invariant both under the flux-reference stabilizer transformations, and under gauge transformations. Therefore, $H_{\zeta_\partial} = -\oint \sqrt{h}\, \mathrm{Tr}(f\zeta_\partial)$ can be understood as defining a function on the reduced CSSS $\Phi^{[f]}/\mathcal{G}$, or more precisely $H_{\zeta_\partial} = \Pi^* \tilde{H}_{\zeta_\partial}$ for a $\tilde{H}_{\zeta_\partial} \in \Omega^0(\Phi^{[f]}/\mathcal{G})$. Moreover, from its gauge invariance, one deduces that

$$\mathbb{d}H_{\zeta_\partial} = \mathbb{d}_H H_{\zeta_\partial}. \tag{72}$$

One can then compute the contraction

$$\Omega^{H,[f]}_{\mathrm{ext}}(\mathbb{Y}_{\zeta_\partial}) = \omega^{H,[f]}_{\mathrm{ext}}(\mathbb{Y}_{\zeta_\partial}) = \oint \sqrt{h}\, \mathrm{Tr}\big(f_o[u^{-1}\zeta_\partial u, u^{-1}\mathbb{d}_H u]\big), \tag{73}$$

and thus to verify that

$$\Omega^{H,[f]}_{\mathrm{ext}}(\mathbb{Y}_{\zeta_\partial}) = -\mathbb{d}H_{\zeta_\partial} + H_{\mathbb{d}_H \zeta_\partial}, \tag{74}$$

where $\mathbb{d}_H \zeta_\partial = \mathbb{d}\zeta_\partial - [\zeta_\partial, \varpi_{|\partial R}]$. As all other terms, $H_{\mathbb{d}_H \zeta_\partial}$ is also basic both with respect to flux-reference stabilizer transformations and gauge transformations. It is therefore projectable to a $\tilde{h}_{\zeta_\partial} \in \Omega^1(\Phi^{[f]}/\mathcal{G})$ according to $\Pi^* \tilde{h}_{\zeta_\partial} := H_{\mathbb{d}_H \zeta_\partial}$.

Using the defining relation of $\Omega^{\mathrm{red},[f]}$ and the surjectivity of $\Pi$, one finds

$$\Omega^{\mathrm{red},[f]}(\tilde{\mathbb{Y}}_{\zeta_\partial}) = -\mathbb{d}\tilde{H}_{\zeta_\partial} + \tilde{h}_{\zeta_\partial}. \tag{75}$$

Therefore, on the reduced CSSS $\Phi^{[f]}/\mathcal{G}$ the flux rotation $\tilde{\mathbb{Y}}_{\zeta_\partial}$ is a Hamiltonian vector field of charge $\tilde{H}_{\zeta_\partial}$ if and only if $\tilde{h}_{\zeta_\partial} = 0$, i.e. if and only if $H_{\mathbb{d}_H \zeta_\partial} = 0$.

However, for $H_{\mathbb{d}_H \zeta_\partial} = 0$ to vanish (without $H_{\zeta_\partial}$ to vanish as well) one needs

$$\mathbb{d}_H \zeta_\partial = 0. \tag{76}$$

In turn, this condition can be satisfied nontrivally throughout $\Phi_{\text{ext}}^{[f]}$—i.e. without incurring in overly-restrictive integrability conditions—only if $\varpi$ is flat, i.e. only if $\mathbb{F} = 0$.

Then, if $\mathbb{F} = 0$ and if in addition the parameters $\zeta_\partial$'s do not change under flux-rotations $\mathbb{Y}_{\zeta'_\partial}(\zeta_\partial) = 0$ (this is the case if $\zeta_\partial$ is independent of $E_{\text{Coul}}$), the charges $\tilde{H}_{\zeta_\partial}$ satisfy a Lie($G_{|\partial R}$) Poisson algebra

$$\{\tilde{H}_{\zeta_\partial}, \tilde{H}_{\zeta'_\partial}\} := \Omega^{\text{red},[f]}(\tilde{\mathbb{Y}}_{\zeta_\partial}, \tilde{\mathbb{Y}}_{\zeta'_\partial}) = \tilde{H}_{[\zeta_\partial, \zeta'_\partial]}, \tag{77}$$

constituting a representation of the following Lie algebra of field-space vector fields:[46]

$$[\![\tilde{\mathbb{Y}}_{\zeta_\partial}, \tilde{\mathbb{Y}}_{\zeta'_\partial}]\!] = \tilde{\mathbb{Y}}_{[\zeta_\partial, \zeta'_\partial]}. \tag{78}$$

Notice that the Hamiltonian nature of flux rotations relies on the flatness of $\varpi$, even though $\Omega^{\text{red},[f]}$ is $\varpi$-independent, because the very *definition* of flux rotations relies on a choice of $\varpi$ to provide the radiative/Coulombic split of the electric field: only $E_{\text{Coul}}$ is affected by the action of $\mathbb{Y}_{\zeta_\partial}$—see paragraph 4.4. In particular, if the topology of $\mathcal{A}$ is such that no flat $\varpi$ exists on it,[47] it is then impossible to define a radiative/Coulombic split that corresponds to flux rotations which are Hamiltonian.

To summarize, covariant flux rotations define Hamiltonian vector fields, i.e. kinematical symmetries, on $\Phi^{[f]}/\mathcal{G}$ only if they are based on a $\varpi$ which is flat. In this case, their Hamiltonian generator is given by a (gauge-invariant) smearing of the electric flux, $\tilde{H}_{\zeta_\partial}$. Moreover, if the parameters $\zeta_\partial$'s are chosen independent of $E_{\text{Coul}}$, these charges satisfy a Lie($G_{|\partial R}$) Poisson algebra, hallmark of the noncommutativity of the electric fluxes $f$.

Finally, note that since flux rotations alter the Coulombic part of the electric field and therefore the energy content of the field configuration, it seems unlikely that flux rotations can be promoted to dynamical symmetries too.

## 5 Beyond CSSS: edge modes

In the first part of this article I built a canonical symplectic structure on reduced CSSS. In the reminder of this article I will discuss how to define a reduced symplectic structure in a context that goes beyond the CSSS framework. This will involve the inclusion of new "edge mode" dof symplectically conjugate to the electric flux. I will argue that the Donnelly–Freidel prescription for the inclusion of edge modes [32] is the most natural one. Despite this fact I will show that this prescription is equivalent to breaking gauge symmetry at the boundary. Based on this observation I will argue at the end that only the CSSS approach provides a canonical framework which is free of ambiguities.

**5.1 Beyond CSSS** In the construction of $\Omega^{\text{red},[f]}$, it was crucial to restrict attention to a (covariant) flux superselection sector. However, the notion of superselection has at times been criticized and put under scrutiny [28–31]. This criticism relies on the observation that the superselection of a dof $\phi$ is often the artificial consequence of one's inadvertent neglect of another dof $\overline{\phi}$ which is naturally "conjugate" to $\phi$ and thus serves as a "reference frame" for it: were $\overline{\phi}$ included in the description of the system, $\phi$ would not be superselected.

As briefly summarized in Sect. 1.6, a careful analysis of the gluing of YM dof across adjacent regions shows that in the present context the neglected dof responsible for the superselection of the flux $f$ are the gauge invariant, radiative, dof supported in the complement of $R$ within the Cauchy surface $\Sigma$ [5] (see Sect. 6). Therefore, the emergence of superselection in the present context is perfectly in line with the understanding of superselection outlined above.

---

[46]This equation holds under the same condition which determine the validity of (77).

[47]This is the same as saying that $\mathcal{A}$ admits no global section [57,58].

This said, it is nonetheless of interest to investigate the consequences of *not* restricting to a flux superselection sector.

Then, $\Phi^{[f]}$ is replaced with the larger space of all on-shell field configurations

$$\Phi_{\mathsf{G}} := \bigcup_{f \in \mathcal{F}} \{\phi \in \Phi | \mathsf{G}^f = 0\}, \tag{79}$$

where $\mathcal{F} := \{f\}$ is the total space of fluxes. On the reduced on-shell space, $\Phi_{\mathsf{G}}/\mathcal{G}$, the 2-form $\Omega_{\varpi}^{\mathrm{red}}$ (35) can only have a larger kernel than before, with the result that this kernel is now nontrivial even in Abelian theories. This is because in $\Phi_{\mathsf{G}}/\mathcal{G}$ one can not only "rotate" fluxes in a give conjugacy class but also change their conjugacy class, and now both transformations go undetected by $\Omega_{\varpi}^{\mathrm{red}}$.

In a CSSS, the issue of the kernel of $\Omega_{\varpi}^{\mathrm{red}}$ was solved by recognizing the presence of a canonically-given symplectic structure on the space of covariantly superselected fluxes $[f] \cong \mathcal{G}_{|\partial R}/\mathcal{G}_{|\partial R}^o$. However, in general no canonical symplectic structure exists for the total space of fluxes $\mathcal{F} \cong \mathcal{C}^\infty(\partial R, \mathrm{Lie}(G))$.[48] In a finite dimensional analogue, whereas a (co)adjoint orbit in (the dual of) a Lie algebra admits a canonical symplectic structure, the Lie algebra itself does not: e.g. the Lie algebras $(\mathbb{R}, +)$ and $\mathrm{Lie}(\mathrm{SU}(2))$ even fail to be even-dimensional.

Therefore, the only option to solve the problem of the kernel of $\Omega_{\varpi}^{\mathrm{red}}$ in the context of non-superselected on-shell fields, is to enlarge, or extend, the phase space by including new, *additional*, dof canonically conjugate to the fluxes $f \in \mathcal{F}$. The question is: is there a most natural way to perform this extension?

**5.2 Two possible phase space extensions**  As I have already noted in section 4.6, fluxes are best understood as objects valued in the *dual* of the Lie algebra, $\mathcal{F} \cong \mathcal{C}^\infty(\partial R, \mathrm{Lie}(G)^*)$, where the dualization is made through the map $f \mapsto f^* = \mathrm{Tr}(f \cdot)$. For brevity I will use the (slightly misleading) notation $\mathrm{Lie}(\mathcal{G}^\partial)^* := \mathcal{C}^\infty(\partial R, \mathrm{Lie}(G)^*)$.

Enlarging the phase space by adding new dof canonically conjugate to the flux means "doubling" the space $\mathcal{F}$ to obtain a new symplectic space $(\mathcal{D}^{\mathcal{F}}, \omega)$ on which gauge transformation have a Hamiltonian action. There are two natural ways of achieving this.

These two ways are based on the "doubled" spaces $(\mathcal{D}_0^{\mathcal{F}}, \omega_0 = \mathbb{d}\vartheta_0)$ and $(\mathcal{D}_{\mathrm{DF}}^{\mathcal{F}}, \omega_{\mathrm{DF}} = \mathbb{d}\vartheta_{\mathrm{DF}})$ respectively defined by:

$$\mathcal{D}_0^{\mathcal{F}} := \mathrm{Lie}(\mathcal{G}^\partial) \times \mathrm{Lie}(\mathcal{G}^\partial)^* \ni (\alpha, f^*) \quad \text{and} \quad \vartheta_0 := \oint \sqrt{h}\,\mathrm{Tr}(f \mathbb{d}\alpha), \tag{80}$$

and

$$\mathcal{D}_{\mathrm{DF}}^{\mathcal{F}} := \mathcal{G}^\partial \times \mathrm{Lie}(\mathcal{G}^\partial)^* \ni (k, f^*) \quad \text{and} \quad \vartheta_{\mathrm{DF}} := \oint \sqrt{h}\,\mathrm{Tr}(f \mathbb{d}k k^{-1}). \tag{81}$$

Notice that $\mathcal{D}_0^{\mathcal{F}} \cong \mathcal{C}^\infty(\partial R, \mathrm{T}^*\mathrm{Lie}(G))$ and $\mathcal{D}_{\mathrm{DF}}^{\mathcal{F}} \cong \mathcal{C}^\infty(\partial R, \mathrm{T}^*G)$, both featuring $\mathrm{Lie}(\mathcal{G}^\partial)^*$ as momentum space. In particular, with the latter identification, $(k, f^*)_{x \in \partial R}$ are coordinates on $\mathrm{T}^*G$ obtained through the right-invariant trivialization of the bundle, and $\vartheta_{\mathrm{DF}}$ originates in the tautological 1-form on $\mathrm{T}^*G$. The space $(\mathcal{D}_{\mathrm{DF}}^{\mathcal{F}}, \omega_{\mathrm{DF}})$ is the *edge-mode* phase-space proposed by Donnelly and Freidel [32].[49] It will become clear that this is the most natural choice between the two.

The inclusion of the degrees of freedom $\alpha$ or $k$ leads to the following extensions of the on-shell phase space (here, $\bullet \in \{0, \mathrm{DF}\}$):

$$\pi_{\mathcal{F},\bullet} : \Phi_{\mathsf{G}}^{\mathcal{F},\bullet} := \to \Phi_{\mathsf{G}}, \qquad \Phi_{\mathsf{G}}^{\mathcal{F},\bullet} := \begin{cases} \Phi_{\mathsf{G}} \times \mathrm{Lie}(\mathcal{G}^\partial) & \text{for } \bullet = 0, \\ \Phi_{\mathsf{G}} \times \mathcal{G}^\partial & \text{for } \bullet = \mathrm{DF}, \end{cases} \tag{82}$$

---

[48]This group is distinct from $\mathcal{G}_{|\partial R}$ since the latter does not contain "large boundary gauge transformations." Cf. footnote 37.

[49]The space $(\mathcal{D}_0^{\mathcal{F}}, \omega_0)$ has appeared in [59, Sect.2.7, Rmk.15]. See also [41] for more details.

for projections $\pi_{\mathcal{F},\bullet}$ which send $(\alpha, f) \mapsto f$ and $(k, f) \mapsto f$ respectively.

The demand of gauge invariance of $\vartheta_\bullet$, i.e. $\mathbb{L}_{\xi^\sharp}\vartheta_\bullet = 0$ for $\mathbb{d}\xi = 0$, forces us to demand that gauge transformations act on $\alpha$ and $k$ (now seen as coordinates in $\Phi_G^{\mathcal{F},\bullet}$) by the adjoint representation and (inverse) left translations respectively:

$$\alpha^g = g_{|\partial R}^{-1} \alpha g_{|\partial R} \qquad \text{and} \qquad k^g = g_{|\partial R}^{-1} k. \tag{83}$$

Thus, the extended spaces $\Phi_G^{\mathcal{F},\bullet}$ are naturally foliated by gauge orbits. I will call the tangent space to the gauge orbits in $\Phi_G^{\mathcal{F},\bullet}$ the *vertical subspace of* $\Phi_G^{\mathcal{F},\bullet}$ and I will denote it $V_{\mathcal{F},\bullet}$.

Pulling back $\varpi$ from $\Phi_G$ to $\Phi_G^{\mathcal{F},\bullet}$ through the canonical projection $\pi_{\mathcal{F},\bullet}$ (but omitting the pullback in the following formulas), one can introduce horizontal derivatives on $\Phi_G^{\mathcal{F},\bullet}$:

$$\mathbb{d}_H f = \mathbb{d}f + [f, \varpi_{|\partial R}] \qquad \text{and} \qquad \begin{cases} \mathbb{d}_H \alpha = \mathbb{d}\alpha + [\alpha, \varpi_{|\partial R}] & \text{for } \bullet = 0, \\ \mathbb{d}_H k = \mathbb{d}k + \varpi_{|\partial R} k & \text{for } \bullet = \text{DF}. \end{cases} \tag{84}$$

Hence, the horizontal modifications of the canonical symplectic forms on $\Phi_G^{\mathcal{F},\bullet}$ are

$$\vartheta_0^H := \oint \sqrt{h} \, \text{Tr}\big(f \, \mathbb{d}_H \alpha\big) \in \Omega^1(\Phi_G^{\mathcal{F},0}), \tag{85a}$$

and

$$\vartheta_{\text{DF}}^H := \oint \sqrt{h} \, \text{Tr}\big(f \, \mathbb{d}_H k k^{-1}\big) \in \Omega^1(\Phi_G^{\mathcal{F},\text{DF}}). \tag{85b}$$

It is immediate to check that these 1-forms are basic with respect to the action of gauge transformations, and therefore so are the following 2-forms (cf. the derivation of (25))

$$\omega_0^H := \mathbb{d}\vartheta_0^H = \oint \sqrt{h} \, \text{Tr}\big(\mathbb{d}_H f \, \curlywedge \, \mathbb{d}_H \alpha + [f, \alpha]\mathbb{F}\big) \in \Omega^2(\Phi_G^{\mathcal{F},0}), \tag{86a}$$

and

$$\omega_{\text{DF}}^H := \mathbb{d}\vartheta_{\text{DF}}^H = \oint \sqrt{h} \, \text{Tr}\big(\mathbb{d}_H f \, \curlywedge \, \mathbb{d}_H k k^{-1} + \tfrac{1}{2} f [\mathbb{d}_H k k^{-1} \, \stackrel{\curlywedge}{,} \, \mathbb{d}_H k k^{-1}] + f \mathbb{F}\big) \in \Omega^2(\Phi_G^{\mathcal{F},\text{DF}}). \tag{86b}$$

Using the projections $\pi_{\mathcal{F},\bullet}$, one can thus define the following presymplectic 2-forms over the extended on-shell phase spaces $\Phi_G^{\mathcal{F},\bullet}$ (here, $\iota : \Phi_G \hookrightarrow \Phi$):

$$\Omega_\bullet^{H,\mathcal{F}} := \pi_{\mathcal{F},\bullet}^* \iota^* \Omega^H + \omega_\bullet^H \in \Omega^2(\Phi_G^{\mathcal{F}}). \tag{87}$$

These presymplectic 2-forms are basic and closed. Moreover, their respective kernels are given by the vertical subspaces of $\Phi_G^{\mathcal{F},\bullet}$:

$$\ker(\Omega_\bullet^{H,\mathcal{F}}) = V_{\mathcal{F},\bullet}. \tag{88}$$

Hence, the presymplectic 2-forms $\Omega_\bullet^{H,\mathcal{F}}$ induce a symplectic structure on the reduced spaces $\Phi_G^{\mathcal{F},\bullet}/\mathcal{G}$. Introducing the projections $\tilde{\pi}_{\mathcal{F},\bullet} : \Phi_G^{\mathcal{F},\bullet} \to \Phi_G^{\mathcal{F},\bullet}/\mathcal{G}$, the reduced symplectic structures $\Omega_\bullet^{\text{red},\mathcal{F}} \in \Omega^2(\Phi_G^{\mathcal{F},\bullet}/\mathcal{G})$ are defined by the relations

$$\tilde{\pi}_{\mathcal{F},\bullet}^* \Omega_\bullet^{\text{red},\mathcal{F}} := \Omega_\bullet^{H,\mathcal{F}}. \tag{89}$$

Whereas $\Omega_0^{\mathrm{red},\mathcal{F}}$ is $\varpi$-dependent, the DF symplectic structure is $\varpi$-*in*dependent. Indeed, equations (84–87) and (23) yield (omitting the pullbacks[50]):

$$\Omega_{\mathrm{DF}}^{H,\mathcal{F}} = \mathbb{d}(\theta^H + \vartheta_{\mathrm{DF}}^H) = \mathbb{d}(\theta^H + \theta^V + \vartheta_{\mathrm{DF}}) = \Omega + \omega_{\mathrm{DF}}, \tag{90}$$

where the right-most term in this equation is manifestly independent of the choice of $\varpi$.

Given the relationship of (flat) functional connections and gauge fixings this seems to mean that the DF symplectic structure is fully gauge-invariant. However, despite these appearances, I shall argue in section 5.8 that the gauge invariance of the DF symplectic structure is an illusion.

**5.3 Lie bialgebras and quantum doubles** Both symplectic structures $\Omega_{\mathcal{F},\bullet}^H$ find their origin in the theory of Lie-bialgebras and quantum doubles [56], albeit being somewhat trivial examples thereof. In particular, $\omega_0$ reflects the canonical symplectic structure carried by the double Lie-bialgebra $\mathfrak{d}_0 = \mathrm{Lie}(G) \oplus \mathrm{Lie}(G)^*$ built from the Lie bialgebra $\mathfrak{g}_0 = (\mathrm{Lie}(G), [\cdot,\cdot], \gamma = 0)$ with trivial cobracket $\gamma$. Similarly, $\omega_{\mathrm{DF}}$ reflects the canonical symplectic structure carried by the Heisenberg double $\mathfrak{D}_+ = \exp \mathfrak{d}_0 \cong \mathrm{T}^*G$.

I mention this because, although the cases treated here are the most trivial examples of Lie-bialgebras and quantum doubles, it turns out that upon discretization other far less trivial "double" structures can naturally arise [60] or become available [61,62]. These structures can be understood as deformations of these most trivial cases to new phase spaces where gauge acts through a quantum-group symmetry. Typically, these more general structure involve some sort of "exponentiated flux" (in a way similar to how $\mathfrak{D}_+ = \exp \mathfrak{d}_0$). I refer to the cited articles for further references on this topic and its relevance for quantum gravity, self-dual formulations of YM theory, and the theory of topological phases of matter.

**5.4 Dof in $(\Phi_{\mathrm{ext}}^{[f]}, \Omega_{\mathrm{ext}}^{H,[f]})$ vs. $(\Phi_{\mathsf{G}}^{\mathcal{F},\mathbf{DF}}, \Omega_{\mathrm{DF}}^{H,\mathcal{F}})$** The reader will have surely noticed the parallel between the formulas that characterize the field-space extension à la DF $\Phi_{\mathsf{G}}^{\mathcal{F},\mathrm{DF}}$, and those that describe the presymplectic structure in the extended description of the CSSS $\Phi_{\mathrm{ext}}^{[f]}$. Indeed, the two are *formally* mapped onto each other by $u \rightsquigarrow k$.[51]

But crucially, whereas the $u$'s are just an (over)-parametrization of the already existing flux dof in the CSSS $[f]$, in the DF framework not only the electric fluxes live in the larger space $f \in \mathcal{F}$ but also the edge modes $k$'s embody new, independent dof. Mathematically this difference is encoded in the relation $f = uf_o u^{-1}$ and the ensuing flux-stabilizer symmetry $u \mapsto ug_o$ that reduces the variables $u \in \mathcal{G}_{|\partial R}$ to variables in $\mathcal{G}_{|\partial R}/\mathcal{G}_{|\partial R}^o \cong [f]$. There is no such symmetry acting on the edge modes.

**5.5 Flux rotations in the Donnelly-Freidel extension** The natural extension of flux rotations to the Donnelly–Freidel extended phase space requires that the edge modes $k$ also transform under this kinematical symmetry. In $\Phi_{\mathsf{G}}^{\mathcal{F},\mathrm{DF}}$, I thus define[52]

$$\mathbb{Y}_{\zeta_\partial}(f) = -[f,\zeta_\partial], \qquad \mathbb{Y}_{\zeta_\partial}(k) = \zeta_\partial k \qquad \text{and} \qquad \mathbb{Y}_{\zeta_\partial}(\bullet) = 0 \text{ otherwise}, \tag{91}$$

i.e. for $\bullet \in \{A, E_{\mathrm{rad}}, \psi, \overline{\psi}\}$. As above, for flux rotations to be projectable onto the reduced phase space, $\zeta_\partial$ must be field dependent in a way that makes it transform covariantly under gauge transformations: $\mathbb{L}_{\xi^\sharp}\zeta_\partial = [\zeta_\partial, \xi_{|\partial R}]$. I call the corresponding flux rotations, covariant flux rotations.

---

[50]The action of the missing pullbacks is independent of $\varpi$.

[51]E.g. $\vartheta^{[f]} = \oint \sqrt{h}\,\mathrm{Tr}(f_o u^{-1}\mathbb{d}u) = \oint \sqrt{h},\mathrm{Tr}(f\,\mathbb{d}uu^{-1}) \rightsquigarrow \oint \sqrt{h}\,\mathrm{Tr}(f\,\mathbb{d}kk^{-1}) = \vartheta_{\mathrm{DF}}$.

[52]Note that the vector fields $\mathbb{Y}_{\zeta_\partial}$ are here redefined to be sections of $\mathrm{T}\Phi_{\mathsf{G}}^{\mathcal{F},\bullet}$ rather than $\mathrm{T}\Phi^{[f]}$.

Notice that the very definition of flux rotations *depends* on a choice of $\varpi$: it requires splitting the electric field into radiative and Coulombic components, so that $\mathbb{Y}_{\zeta_\partial}$ can act on the latter and not on the former. This is why, although $\Omega^H_{\mathcal{F},\text{DF}}$ does not depend on $\varpi$, the following flow equation does:

$$\Omega^{H,\mathcal{F}}_{\text{DF}}(\mathbb{Y}_{\zeta_\partial}) = -\mathbb{d}_H H^{\text{DF}}_{\zeta_\partial} + H^{\text{DF}}_{\mathbb{d}_H \zeta_\partial} \qquad \text{where} \qquad H^{\text{DF}}_{\zeta_\partial} = \oint \sqrt{h}\,\text{Tr}(f\,\zeta_\partial). \tag{92}$$

From this and in complete analogy with the reasoning made within a single CSSS, one deduces that covariant flux rotations are Hamiltonian in the reduced symplectic space $(\Phi^{\mathcal{F},\text{DF}}_G/\mathcal{G}, \Omega^{H,\mathcal{F}}_{\text{DF}})$ if and only if $\varpi$ is flat.

In the Abelian case, $f$ is left invariant by flux rotations, which have the sole effect of translating $k$: i.e. Abelian flux rotations have absolutely no effect on the Gauss constraint and the Coulombic electric field. Indeed, in the CSSS framework, Abelian flux rotations are completely trivial. Here their action is nontrivial only because of the extension of the phase space by edge modes. In this sense—contrary to what happens in a CSSS—in DF the physical significance of Abelian flux rotations ultimately relies on the physical interpretation one attaches to the edge mode $k$; see below.

**5.6 "Boundary symmetries" of the Donnelly-Freidel extension**  Having introduced the new edge-mode dof $k$, the possibility arises of producing a new symmetry that translates the $k'$'s while leaving all other fields untouched. This idea yields what DF called "surface (or boundary) symmetries" [32].

Define the vector field $\mathbb{Z}_{\eta_\partial} \in \mathfrak{X}^1(\Phi^{\mathcal{F},\text{DF}}_G)$ by the following action on the coordinate functions of $\Phi^{\mathcal{F},\text{DF}}_G$:

$$\mathbb{Z}_{\eta_\partial}(k) = k\eta_\partial \qquad \text{and} \qquad \mathbb{Z}_{\eta_\partial}(\bullet) = 0 \text{ otherwise}, \tag{93}$$

for $\eta_\partial$ a possibly field-dependent parameter valued in $\text{Lie}(\mathcal{G}^\partial)$. Since these transformations and gauge transformations act on the opposite side of $k$, the vector field $\mathbb{Z}_{\eta_\partial}$ is projectable to the reduced phase space—i.e. $[\![\mathbb{Z}_{\eta_\partial}, \xi^\sharp]\!] = 0$ ($\mathbb{d}\xi = 0$)—if and only if $\mathbb{d}\eta_\partial = 0$. I will henceforth assume this condition to hold. The ensuing $\mathbb{Z}_{\eta_\partial}$ are DF's boundary symmetries.

Boundary symmetries are Hamiltonian [32] (hence the name "symmetries"). Indeed,

$$\Omega^H_{\mathcal{F},\text{DF}}(\mathbb{Z}_{\eta_\partial}) = -\mathbb{d}Q^{\text{DF}}_{\eta_\partial} \qquad \text{where} \qquad Q^{\text{DF}}_{\eta_\partial} = \oint \sqrt{h}\,\text{Tr}\big(\eta_\partial \text{Ad}^{-1}_k f\big). \tag{94}$$

Notice that $Q^{\text{DF}}_{\eta_\partial}$ is gauge invariant and hence the pullback by $\tilde{\pi}_{\mathcal{F},\text{DF}}$ of a function $\tilde{Q}^{\text{DF}}_{\eta_\partial}$ defined on the reduced phase space: $Q^{\text{DF}}_{\eta_\partial} = \tilde{\pi}^*_{\mathcal{F},\text{DF}}\tilde{Q}^{\text{DF}}_{\eta_\partial}$.

In the Abelian case, these transformations have the same action as flux rotations—however, this is a coincidence due to the fact that Abelian flux rotations are in a sense trivial (see above). In general, they represent a pure translation of the edge modes $k$ by a parameter which is "malleable" over $\partial R$ but "rigid" throughout phase space. Their physical meaning fully relies on the physical interpretation one attaches to the edge mode $k$.

Geometrically DF's boundary symmetries encode an ambiguity in the identification of the "origin" of the extended phase space $\mathcal{D}^{\mathcal{F}}_{\text{DF}} = \mathcal{G}^\partial \times \text{Lie}(\mathcal{G}^\partial)^*$ with that of $\mathcal{C}^\infty(\partial R, \text{T}^*G)$. In fact, whereas $\text{T}^*G$ seen as a group possesses an identity, i.e. a preferred origin, as a manifold it is completely homogeneous and thus lacks a preferred origin. Therefore the manifold isomorphism $\text{T}^*G \cong G \times \text{Lie}(G)^*$ is natural but not canonical, i.e. depends on the choice of an origin.[53] In other words, the origin of these symmetry can be ultimately traced back to the

---

[53]Combinations of flux rotations and "boundary symmetry" can also change the orientation of the tangent plane at the origin, without shifting the origin.

fact that the projection from $T^*G$ to $\mathrm{Lie}(G)^*$ fails to be canonical—even if the cotangent bundle $T^*G$ is trivial.

This suggests that the "boundary symmetries" might encode a fundamental ambiguity in the definition of the DF extended phase space and symplectic structure. This is what I will discuss next.

**5.7  DF edge modes as open Wilson lines**  Comparison to the lattice suggest an interpretation of the edge mode $k$'s as open Wilson lines which land transversally onto the boundary. This interpretation explains not only the fact that $k$ is conjugate to the flux, but also its gauge transformation property.

Since variations of $A$ *within $R$* do not affect $k$, for this interpretation to be valid the whole Wilson line needs to lie in the *complementary* region $\overline{R} = \Sigma \setminus \mathring{R}$—which might seem puzzling if $\partial R$ is an asymptotic boundary.

Thus, for any $x \in \partial R$, one can interpret $k$ as being given by the path-ordered exponential of $A$ along some (arbitrary) choice of paths $\{\gamma_x\}_{x \in \partial R}$:

$$k(x) = \overleftarrow{\mathbb{P}\exp} \int_{\gamma_x} A \qquad \text{where} \qquad \gamma_x : [0,1] \to \overline{R},\ \gamma_x(1) = x. \tag{95}$$

From this perspective, the "boundary symmetries" of the edge modes are nothing else than changes in the choice of gauge at $\gamma_x(0)$ (or possibly changes in the choice of the ensemble of paths $\{\gamma_x\}$). According to this construction, the Wilson lines $k(x)$ have the interpretation of "gauge reference frames" with respect to the *exterior* of $R$; or, if $\gamma_x(0)$ is taken arbitrarily close to $\gamma_x(1) = x$, it seems reasonable to conclude that the edge modes $k$ are nothing else than the result of breaking of the original gauge symmetry at $\partial R$.

This viewpoint is confirmed by the following observation: the DF extension and symplectic structures can be obtained by reducing the space of on-shell configurations equipped with the symplectic structure $\Omega$ with respect to the action of the group $\mathring{\mathcal{G}}$ of gauge transformations that are trivial at the boundary.[54] I will call $\mathring{\mathcal{G}}$ the group of *bulk* gauge transformations.

This observation means that DF *does* break the gauge symmetry at the boundary, only to give the impression it does not by resorting to a Stückelberg trick at $\partial R$. Even more importantly, it also means that DF is based on an implicit choice of a (boundary) gauge fixing and that the effects of changing this gauge fixing are completely degenerate with the effects of changing the field configuration up to bulk-only gauge transformations. Let me sketch a proof.

**5.8  DF edge modes from breaking of boundary gauge invariance**  To define a symplectic form on $\Phi_G/\mathring{\mathcal{G}}$, it is necessary to first parametrize this space effectively. Instead of working with $\Phi_G$ foliated by orbits of $\mathring{\mathcal{G}}$, I will work with a larger space $\underline{\Phi}_G \times \mathcal{K}$ foliated by orbits of an enlarged gauge group $\mathcal{G} \times \mathring{\mathcal{G}}$:

$$\pi_{\mathcal{K}} : \underline{\Phi}_G \times \mathcal{K} \to \Phi_G, \quad (\underline{A}, \underline{E}, \dots, k) \mapsto (A = k^{-1}\underline{A}k + k^{-1}\mathrm{d}k, E = k^{-1}\underline{E}k, \dots), \tag{96}$$

where $k \in \mathcal{K} \cong \mathcal{G}$ is a new $\mathcal{G}$-valued field, and where the two gauge symmetries act as:

$$\mathcal{G} : \begin{cases} \underline{A}^g = g^{-1}\underline{A}g + g^{-1}\mathrm{d}g, \\ \underline{E}^g = g^{-1}\underline{E}g, \\ \dots \\ k^g = g^{-1}k, \end{cases} \qquad \text{and} \qquad \mathring{\mathcal{G}} : \begin{cases} \underline{A}^{\mathring{g}} = \underline{A}, \\ \underline{E}^{\mathring{g}} = \underline{E}, \\ \dots \\ k^{\mathring{g}} = k\mathring{g}; \end{cases} \tag{97}$$

---

[54]I have heard or read this argument before, but I was unable to track a publication filling in the details.

here the "…" stand for the obvious generalizations in presence of matter fields. Note that the $\mathcal{G}$-symmetry is meant to reabsorb the new dof $k$ (à la Stückelberg), whereas the the $\mathring{\mathcal{G}}$ symmetry is the original gauge symmetry which now conveniently acts on the new $k$ fields only.

Denote $\iota : \Phi_{\mathrm{G}} \hookrightarrow \Phi$ as before. Then, the on-shell symplectic structure $\iota^*\Omega$—for the full $\Omega = \int \sqrt{g}\,\mathrm{Tr}(\mathbb{d}E \wedge \mathbb{d}A) + \ldots$—is basic with respect to the action of *bulk* gauge transformations $\mathring{\mathcal{G}}$. Its pullback by $\pi_{\mathcal{K}}$ yields the following 2-form on the extended space $\underline{\Phi}_{\mathrm{G}} \times \mathcal{K}$:

$$\pi_{\mathcal{K}}^* \iota^* \Omega = \int \sqrt{g}\,\mathrm{Tr}(\mathbb{d}\underline{E} \wedge \mathbb{d}\underline{A}) + \cdots + \mathbb{d}\oint \sqrt{h}\,\mathrm{Tr}(f\,\mathbb{d}k k^{-1}). \tag{98}$$

This 2-form is basic with respect to the action of the enlarged gauge group $\mathcal{G} \times \mathring{\mathcal{G}}$. This is clear from the following two facts: on the one hand $\pi_{\mathcal{K}}^*(A) = k^{-1}\underline{A}k + k^{-1}\mathrm{d}k$ etc. are manifestly $\mathcal{G}$-invariant expressions, and on the other hand (on-shell of the Gauss constraint) $k$ appears only at the boundary where the action of $\mathring{\mathcal{G}}$ is trivial. Since $\mathring{\mathcal{G}}$ acts trivially on the expression $\pi_{\mathcal{K}}^* \iota^* \Omega$, I will denote by the same symbol the 2-form obtained by projecting $\pi_{\mathcal{K}}^* \iota^* \Omega$ down to $(\underline{\Phi}_{\mathrm{G}} \times \mathcal{K})/\mathring{\mathcal{G}}$. Note that for now I am quotienting out the *bulk* gauge symmetries only.

From these expressions and the action of the $\mathcal{G}$-gauge symmetry it is clear that (cf. also (90))[55]

$$\left((\underline{\Phi}_{\mathrm{G}} \times \mathcal{K})/\mathring{\mathcal{G}}, \pi_{\mathcal{K}}^* \iota^* \Omega\right) \cong \left(\Phi_{\mathrm{G}}^{\mathcal{F},\mathrm{DF}}, \Omega_{\mathrm{DF}}^{H,\mathcal{F}}\right). \tag{99}$$

Therefore, the DF edge modes are nothing else than the would-be-gauge dof unfrozen by the action of quotienting $\Phi_{\mathrm{G}}$ *only* by bulk gauge transformations, rather than by the full group of gauge transformations. In other words, the DF edge modes are the would-be-gauge dof unfrozen through the action of explicitly breaking gauge invariance at the boundary.

Changes in the would-be-boundary-gauge of $A$ correspond to right translations of $k$ by elements of $\mathcal{G}_{|\partial R}$—similarly to the DF boundary symmetries. This suggests that the DF boundary symmetries corresponds to changes in the choice of gauge at the boundary in a context where one is demanding configurations to be equivalent only up to *bulk* gauge transformations.

Let me be more precise about this point, because it involves an important subtlety. To understand what the relationship is among (*i*) right translations of $k$, (*ii*) DF boundary symmetries, and (*iii*) (would-be-)gauge transformations supported at the boundary, I will revert to discussing the action of the full group of gauge transformations $\mathcal{G}$ rather than just $\mathring{\mathcal{G}}$. In other words, I will consider the effect of extending the action of $\mathring{\mathcal{G}}$ to $\mathcal{G}$ in order to understand how would-be-gauge transformations manifest in the gauge-breaking interpretation of the DF phase space presented above.

Thus, consider the bundle $\Phi_{\mathrm{G}} \to \Phi_{\mathrm{G}}/\mathcal{G}$ and refer to figure 1 for a graphical representation of what follows. The choice of a gauge fixing in $\Phi_{\mathrm{G}} \to \Phi_{\mathrm{G}}/\mathcal{G}$ is usually understood as the choice of global section $\sigma : \Phi_{\mathrm{G}}/\mathcal{G} \to \Phi_{\mathrm{G}}$ (left panel in fig. 1). Through vertical translations of the section $\sigma \to \sigma^g := R_g \circ \sigma$ by field-*in*dependent $g \in \mathcal{G}$ ($\mathbb{d}g g^{-1} = 0$), one can define from $\sigma$ an equivariant horizontal foliation $H_\sigma \subset \mathrm{T}\Phi_{\mathrm{G}}$ composed of leaves each "parallel" to $\sigma$ (center panel in fig. 1). Since the choice of a section and of an equivariant horizontal foliation are in 1-to-1 correspondence, I will henceforth identify the choice of a gauge fixing with the entire equivariant horizontal *foliation*, and not with the single section. The choice of a single section in a gauge fixing corresponds to choosing a leaf in the gauge fixing foliation. Infinitesimal changes of horizontal leaf within the horizontal foliation are nothing else than

---

[55]A subtlety: this identification is correct only up to global issues. In the DF phase space one is free to consider $k \in \mathcal{G}^\partial$, whereas in the gauge-breaking setting discussed here one finds that $k \in \mathcal{G}_{|\partial R}$ which corresponds to the connected component of $\mathcal{G}^\partial$ which is connected to the identity. It is not clear to me if this distinction is of any relevance, i.e. whether taking $k \in \mathcal{G}^\partial$ in DF means simply dealing with multiple copies of the same space/theory—one per connected component of $\mathcal{G}^\partial$,—which cannot communicate with each other.

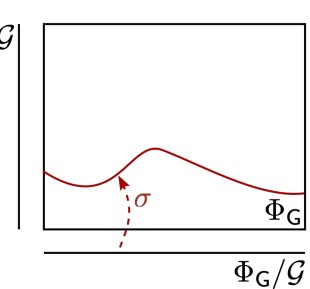
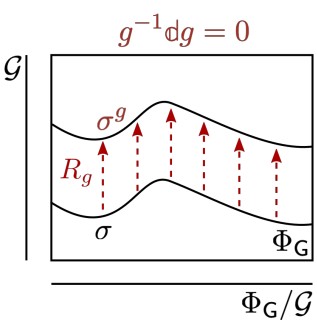
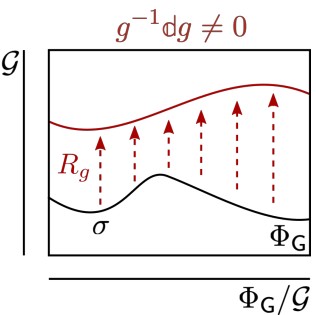

Figure 1: The (fiducial) infinite dimensional field space bundle $\Phi_G \to \Phi_G/\mathcal{G}$. (*Left*) The choice of a gauge fixing section $\sigma : \Phi_G/\mathcal{G} \to \Phi_G$. (*Center*) The generation of the equivariant horizontal foliation associated to $\sigma$ by means of field-*in*dependent gauge transformations. (*Right*) A change in gauge fixing through the action of a field-dependent gauge transformation. The DF boundaries symmetries correspond to changes of leaf in the gauge fixing foliation depicted in the central panel.

the field-*in*dependent boundary symmetries of DF $\mathbb{Z}_{\eta_\partial}$, $\mathbb{d}\eta_\partial = 0$. Boundary symmetries change the leaf in a gauge fixing foliation, not the gauge fixing itself.

Here is an example. In electromagnetism (on a manifold without boundaries), a choice of section is provided by the condition $\nabla^i A_i = 0$. Acting on this condition by a field-*in*dependent gauge transformation $A \mapsto A + \mathrm{d}\lambda$ one produces a family of conditions of the type $\nabla^i A_i = \Lambda$ for $\Lambda = \Delta\lambda$ a field-*in*dependent function over $R$. Then, according to the language introduced above, each function $\lambda$ corresponds to a different leaf in the horizontal foliation corresponding to the Coulomb gauge-fixing of $A$. To change the gauge fixing (foliation), say from Coulomb to axial gauge, one needs instead to act on the condition $\nabla^i A_i$ with a field-*dependent* gauge transformation, which thus changes functional form (right panel of fig. 1). Only field-independent gauge transformations, i.e. *leaf changes*, are Hamiltonian in DF (c.f. the condition $\mathbb{d}\eta_\partial = 0$) and their boundary values correspond to the DF boundary symmetries.

Now, note that the tangent space $H_\sigma$ to a gauge fixing foliation defines a unique connection $\varpi = \varpi_\sigma$ through $H_\sigma = \ker\varpi_\sigma$. Note that $\varpi_\sigma$ encodes all the leaves associated to the gauge fixings at once, and cannot tell them apart. This means that there is no analogue of the DF boundary symmetry in a formulation based on $\varpi$. More explicitly, the Frobenius integrability of $H_\sigma$ means that $\varpi_\sigma$ is flat, i.e. $\varpi_\sigma = h^{-1}\mathbb{d}h$ for some group valued field-space function $h$; in this formulation, leaf changes are field-*in*dependent left translations of $h$, i.e. $h \mapsto gh$ with $\mathbb{d}gg^{-1} = 0$, which do not affect $\varpi_\sigma$ at all. In this regard, see [3, Sect.9] on the relation between flat connections, gauge fixings and dressings (cf. also [53]).

Conversely, as noted above, "true" changes of gauge fixing, i.e. changes of the gauge fixing foliation, can be obtained by acting on $\sigma$ by a field-*dependent* translation. These transformations however are *not* Hamiltonian symmetries of the DF framework. Indeed, it is straightforward to check that the DF symplectic structure is *not* invariant under such transformations.[56] Therefore, the DF symplectic structure on the DF extended phase space depends on an implicit choice of a (boundary) gauge-fixing as much as the reduced symplectic structure on a CSSS depends on an explicit choice of a connection $\varpi$.

(By an "*implicit* choice of a gauge fixing," I mean that in the DF formalism there is no place for specifying which gauge fixing one is using in expressing e.g. $A$ up to boundary gauge transformations $\mathring{\mathcal{G}}$ in terms of the $\mathcal{G}$-gauge classes $(\underline{A}, k) \sim (\underline{A}^g, g^{-1}k)$. In this sense, the boundary gauge is *broken* rather than fixed.)

---

[56]This relies on the fact that if $\mathbb{d}gg^{-1} \neq 0$ then $\mathbb{d}kk^{-1} \neq \mathbb{d}(kg)(kg)^{-1}$.

In sum, although the DF symplectic structure is $\varpi$ independent, it would be erroneous to conclude that it is also gauge-fixing independent—even though its dependence on a gauge fixing is implicit. In the DF framework, a new Hamiltonian boundary symmetry arises which is not available in the CSSS framework and corresponds to changes of a leaf within a gauge-fixing foliation. Importantly, changes in the choice of gauge fixing in the DF framework *fail* to be Hamiltonian symmetry of the corresponding reduced phase space.

In the light of this, it is curious to note that the extension of the phase space by the alternative Lie algebra-valued edge modes $\alpha$—although dependent on $\varpi$—does not suffer of the same type of ambiguity afflicting the DF extension. But since a $\varpi$ dependence is also akin to a gauge-fixing dependence, ultimately both extensions suffer of very similar issues.

**5.9  A physical interpretation for the edge modes?**  A brief remark. The gauge-breaking point of view developed above does *not* encompass possible "emergent" models of physical edge dof similar to those which underpin e.g. the quantum Hall effect in the effective Chern–Simons description,[57] nor physical models of the edge modes in terms of a physically coupled boundary system.

In the latter case, the boundary system is physical, i.e. corresponds to an actual "object" at the boundary of the region. In other words, the boundary is a physical interface. From this perspective group- or Lie algebra-valued edge modes seem a natural but far from unique choice of dof to be coupled at this interface.[58]  Still, in certain cases, DF-like variables *do* emerge naturally. For example, one could couple the gauge system in question (say Abelian) with a superconductor material (at its boundary): if the superconductor is well-described through spontaneous symmetry breaking,[59] then the superconductor's phase provides a physical model for the (Abelian) edge mode, so that changes in $k$ correspond to changes in the state of the superconductor *relative* to the gauge fields [65] (see also the discussion of the Higgs connection in [3, Sect.7]).

Interpreting edge modes as models of a physical system living at the interface $\partial R$ means that edge modes in general do not "disappear" upon gluing of two complementary regions along $\partial R$,[60] a fact that might have consequences for the interpretation of the "entangling (or fusion) product" procedure of gluing introduced in [32].

# 6  Gluing, briefly

Finally, a few words about gluing based on joint work with Gomes [5]. To talk about gluing without introducing further complications of topological origin, consider a setting where the gauge system lives over a topologically trivial Cauchy surface $\Sigma \cong \mathbb{R}^D$ or $\mathbb{S}^D$ viewed as the "gluing" of two complementary regions $\Sigma = R^+ \cup R^-$ across the interface $S = \pm \partial R^\pm$, with $R^\pm \cong \mathbb{D}^D$ and $S \cong \mathbb{S}^{D-1}$.

It has been argued that the introduction of edge modes as gauge reference frames is a necessary step in the reconstruction of global gauge-invariant dof from regional ones.

It has also been argued that edge modes are necessary because without them the union of the regional dof does not encompass all the global dof, i.e. there are more physical dof in $\Sigma$

---

[57]In these effective models, both the bulk and boundary dof emerge as collective modes from the same set of underlying, or "fundamental," dof—i.e. electrons and Maxwell fields. Also, recall that YM theories have a very different symplectic structure compared to Chern–Simons theories. See the remark after (25).

[58]Group valued edge modes are possibly the most "natural" insofar the gauge group acts freely on them. Also, see [63] for an example, among others, of a different kind of boundary dof.

[59]See e.g. [64].

[60]In Chern–Simons their disappearance relies on the chirality of the edge theory.

than in the disjoint union of $R^{\pm}$.[61]

The second statement is correct, the first one is not. Contrary to what happens in a non-gauge system, the reduced symplectic structures on the CSSS's associated with $R^{\pm}$ and $\Sigma$ *fail* to be additive under the gluing $(R^+, R^-) \to \Sigma = R^+ \cup R^-$, i.e. $\Omega_{\Sigma}^{\mathrm{red}} \neq \Omega_{R^+}^{\mathrm{red},[f]} + \Omega_{R^-}^{\mathrm{red},[-f]}$, even after unfreezing the flux dof $f$.[62] The missing dof in $\Omega_{\Sigma}^{\mathrm{red}}$ are the dof conjugate to the electric flux through the interface $S = \pm \partial R^{\pm}$.

Despite this fact, it turns out that the missing term in $\Omega_{\Sigma}^{\mathrm{red}}$ *can* be reconstructed from a combination of the dof present in the regional symplectic structures $\Omega_{R^{\pm}}^{\mathrm{red},[\pm f]}$.

This result might be surprising and is discussed in great detail in [5]. The crucial point is that the "missing" dof which spoil additivity are encoded in the *mismatch* of the regional radiative/horizontal dof at the interface $\partial R$. This quantity is $\Sigma$-nonlocal and, clearly, can be computed from the knowledge of the radiative/horizontal modes in *both* regions, without being encoded in either region alone. I hold the appearance of these mismatches to be a neat example of the relational interpretation of gauge theories [1, 66–69].[63]

# 7 Conclusions

In this article I have studied the construction of a symplectic structure on the reduced phase space of YM theory in the presence of boundaries. I have done so both within covariant superselection sectors for the electric flux, and in a larger context where new "edge-mode" dof conjugate to the fluxes are included in an extended phase space.

The construction within covariant superselection sectors leads to a result that is completely canonical. In the non-Abelian case, this is achieved after one realizes that a canonical completion of the "radiative" symplectic structure $(\iota^* \Omega^H)$ exists which equips the Coulombic electric field with its own symplectic structure. This completion is based on the Kirillov–Konstant–Souriau construction and leads to non-commutative electric fluxes whose boundary smearings generate—if $\varpi$ is flat—physical transformations of the underlying system (flux rotations).

The second construction relies on the inclusion à la Donnelly and Freidel (DF) of new edge dof to the phase space [32]. Although this construction seems at first sight canonical, i.e. independent of any external choice, I argued that the result is nonetheless dependent on an (implicit) choice of gauge at the boundary. This is related to the fact that DF edge-modes can be constructed as would-be-gauge boundary dof originating in the incomplete reduction of the phase space by *bulk* gauge transformations only.

Importantly, the implicit gauge-fixing dependence present in the edge mode description fails to be a Hamiltonian symmetry of the resulting reduced phase space. (The DF boundary symmetries have a related, but more limited, interpretation.) In other words, there is no residual "meta-symmetry" on the reduced phase space which allows one to "physically" implement changes in the choice of the gauge fixing—a choice which therefore remains imprinted in the formalism. In sum, despite being the most natural choice in a context not restricted by a choice

---

[61]This corresponds to the non-factorizability of the Hilbert space of lattice gauge theory upon subdivision of the lattice. Also, a side note: in [32] it was also made clear that edge modes are an over parametrization of the global dof, hence the authors' "entangling product" (or "fusion product")—that is a symplectic reduction procedure that introduces both a new constraint identifying the fluxes on the two sides of the interface and a quotienting procedure to mod-out the conjugate degree of freedom, i.e. "half" of the edge modes.

[62]Notice that in summing the regional symplectic structures, the two KKS contributions for the fluxes cancel each other. Indeed, for the gluing to be meaningful, the fluxes $f^{\pm} = u_{\pm}^{-1} f_o^{\pm} u_{\pm}$ must be equal and opposite to each other, $f^+ = -f^-$, and therefore—choosing $f_o^+ = -f_o^-$ as references—one has $[u_+] = [u_-]$. If the fluxes did not match, it would mean that a charged system were present at the interface.

[63]A note of warning: although I support their general perspective, I personally disagree with some of the more specific statements regarding what is observable and what is not which are put forward in e.g. [69]. This discussion, which would take us too far astray here, is related to the content of footnote 14.

of (covariant) superselection sector, the DF framework includes new would-be-gauge dof by actually *breaking* gauge invariance at the boundary.

Compare this to what happens in a covariant superselection sector where (*i*) no new dof need to be included and (*ii*) the resulting symplectic form is completely canonical, i.e. independent from any external choice. Once again, the only surprising feature arising within a (non-Abelian) covariant superselection sector is the need to complete the symplectic structure for the covariantly superselected fluxes—but then the completion provided by the Kirillov–Konstant–Souriau construction is fully canonical.

Edge modes are also not necessary for "gluing" the YM dof supported on adjacent regions, a fact thoroughly discussed in [5]. There it is shown that a formulation of gluing that does not rely on edge modes has the advantage of revealing the characteristically nonlocal and relational features of the YM dof.

Finally, a word on the superselection of electric fluxes—which imply the superselection of charges [21].[64] This concept has been put under scrutiny and criticized in the past [28] (cf. [31]). In this regard, note that in the present context the superselection of the fluxes is a consequence of restricting one's analysis to a specific region $R$ by deliberately "tracing over" its complement $\overline{R}$ in a Cauchy surface $\Sigma = R \cup \overline{R}$.[65] Thus, the flux superselection is a consequence of this tracing, and *not* a property of the entire universe—a distinction that might get muddled when considering idealized asymptotic boundaries. Furthermore, the fact that covariant superselection sectors admit a canonical symplectic structure, whereas the most natural way to go beyond flux superselection inherently breaks the gauge symmetry at the boundary, provides—in my view—a strong argument in favour of the viability of the notion of flux superselection as attached to finite regions.

## Acknowledgements

I am indebted with Henrique Gomes for the work done together on this topic over the years, and for his thoughtful comments on various aspects of this work. I would also like to thank Hal Haggard for a valuable conversation and many fruitful questions, as well as Jordan François for his feedback. Finally, my gratitude goes to Claire and her Little Fields Farm for the many joyful moments that helped me stay sane during 2020.

# A Appendix

## A.1 Relation to Marsden–Weinstein–Meyer symplectic reduction

In this appendix I briefly sketch the relationship between the symplectic reduction procedure developed in this article and the theory of moments maps developed by Marsden and Weinstein, and Meyer (MWM) [18, 19]. As mentioned in the introduction, drawing a rigorous relationship would however require not only a careful separation of bulk gauge transformation from boundary ones, but also of their moment maps—whose detailed analysis I leave to future work (see also [20]).

**1.1 MWM symplectic reduction in a nutshell** Adapting the notation from the main text, consider a (finite dimensional) symplectic space $(\Phi, \Omega)$, on which a (finite) Lie group $\mathcal{G}$ acts

---

[64]See also: [22–24] as well as [25]. Moreover, for recent results on a residual gauge-fixing dependence of QED in the presence of (asymptotic) boundaries and flux superselection, see [26] (and also [27]). At present it is unclear how these recent results square with the classical treatment presented here.

[65]This language is borrowed from the literature on entanglement entropy, where superselection sectors do play a role [70, 71].

smoothly,

$$R: \quad \begin{aligned} \Phi \times \mathcal{G} &\rightarrow \Phi, \\ (\phi, g) &\mapsto \phi^g := R_g(\phi). \end{aligned} \tag{100}$$

The action $R$ is said Hamiltonian if an equivariant moment map $J$ exists, i.e. if a map $J : \Phi \rightarrow \mathrm{Lie}(\mathcal{G})^*$ exists such that for all $\xi \in \mathrm{Lie}(\mathcal{G})$,

$$\mathbb{L}_{\xi^{\sharp}} J = \mathrm{ad}^*_{\xi} J \qquad \text{and} \qquad \Omega(\xi^{\sharp}) = -\mathrm{d}\langle J, \xi \rangle, \tag{101}$$

where $\xi^{\sharp} \in \Gamma(T\Phi)$ is the infinitesimal flow generated by $\xi$. If $\Phi$ is closed and compact of dimension $2n$, the prescription $\int_{\Phi} \Omega^n J = 0$ fixes $J$ uniquely.

Then, according to the results of MWM, if $0 \in \mathrm{Lie}(\mathcal{G})^*$ is a regular value of $J$, the space

$$\Phi//\mathcal{G} := J^{-1}(0)/\mathcal{G} \tag{102}$$

is canonically equipped with a symplectic structure $\Omega^{\mathrm{red}}$ inherited from $(\Phi, \Omega)$ through the equation

$$\tilde{\pi}^* \Omega^{\mathrm{red}} = \iota^* \Omega, \tag{103}$$

where $\tilde{\pi} : J^{-1}(0) \rightarrow J^{-1}(0)/\mathcal{G}$ and $\iota : J^{-1}(0) \hookrightarrow \Phi$ are a canonical projection and injection, respectively.

This result can be generalized[66] to non-vanishing values of $J$. A convenient way to think of this generalization is via the "shifting trick" [16, Sect.26] (and [17]). Consider first the coadjoint orbit $[j] \subset \mathrm{Lie}(\mathcal{G})^*$ equipped with its canonical KKS symplectic structure $\omega_{\mathrm{KKS}}^{[j]}$. On $([j], \omega_{\mathrm{KKS}}^{[j]})$, the coadjoint action of $\mathcal{G}$ is Hamiltonian with respect to the (identity) moment map $I : [j] \hookrightarrow \mathrm{Lie}(\mathcal{G})^*$. Thus, the enlarged symplectic space

$$(\tilde{\Phi}, \tilde{\Omega}) := (\Phi \times [j], \Omega + \omega_{\mathrm{KKS}}^{[j]}) \tag{104}$$

carries a Hamiltonian action of $\mathcal{G}$ with moment map

$$\tilde{J} = J - I, \tag{105}$$

which allows one to define the symplectic reduction of $\Phi$ at $J \neq 0$ in terms of the reduction of the enlarged space $\tilde{\Phi}$ at $\tilde{J} = 0$:

$$\Phi//_{[j]}\mathcal{G} := \tilde{J}^{-1}(0)/\mathcal{G}. \tag{106}$$

Since the action of $\mathcal{G}$ on $[j]$ is transitive, none of its dof survive the quotient and the only role of the enlargement of $(\Phi, \Omega)$ by $([j], \omega_{\mathrm{KKS}}^{[j]})$ has been that of shifting the value of the moment map $J$ at which the reduction is performed.[67]

### 1.2 Relation to YM theory without boundaries

Ignoring complications due to an infinite dimensional context and the presence of reducible configurations, the MWM symplectic reduction can be directly applied to the YM phase space when boundaries are absent. Then, on a Cauchy surface $\Sigma$, $\partial\Sigma = \emptyset$, one has for pure YM theory: $\Phi = \mathrm{T}^*\mathcal{A}$, $\mathcal{A} = \Omega^1(\Sigma, \mathrm{Lie}(G)) \ni A$, $\Omega = \int_{\Sigma} \sqrt{g}\,\mathrm{Tr}(\mathrm{d}E^i \wedge \mathrm{d}A_i)$, $\mathcal{G} = C^{\infty}(\Sigma, G)$ and $R_g(A) = A^g = g^{-1}Ag + g^{-1}\mathrm{d}g$.

---

[66]The original work referenced above already presented the result in this more general case. See also [72] for further generalizations.

[67]It is not hard to see that the reduced space $\Phi//_{[j]}\mathcal{G}$ is also equal to $J^{-1}([j])/\mathcal{G}$, which is also isomorphic to $J^{-1}(j_o)/\mathcal{G}_{j_o}$ for some $j_o \in [j]$ with stabilizer $\mathcal{G}_{j_o}$. The space $J^{-1}(j_o)/\mathcal{G}_{j_o}$ is the one referred to in the original work [18].

From these formulas it is easy to verify that the relevant moment map $J : \Phi \to \mathrm{Lie}(\mathcal{G})^*$ for the gauge transformations $A \mapsto A^g$ is given by the Gauss constraint[68] $\mathsf{G} = \mathrm{D}_i E^i$

$$\langle J, \bullet \rangle = \int_\Sigma \sqrt{g}\, \mathrm{Tr}(\mathsf{G}\bullet) \qquad (\partial\Sigma = \emptyset), \tag{107}$$

and that the MWM symplectic reduction on $J^{-1}(0)/\mathcal{G}$ coincides with the one detailed in the main text. This is most directly seen by comparing equation (103) with (35) and noticing that *in the absence of boundaries*: (*i*) $\iota$ and $\tilde{\pi}$ have the same meaning in both equations; (*ii*) $\iota^*\Omega^H = \iota^*\Omega$, since $\Omega - \Omega^H = \mathrm{d}\theta^V = -\mathrm{d}\int \sqrt{g}\, \mathrm{Tr}(\mathsf{G}\varpi)$ which identically vanishes when pulled back by $\iota$. (Incidentally, this also shows that in the absence of boundaries there is no residual dependence on $\varpi$ left on the lhs of (35).)

Notice that reduction at $J \neq 0$ has no physical bearing in this case:[69] all physical configurations of a gauge theory must satisfy the Gauss constraint.

**1.3 Boundaries and superselection sectors** Restricting the above construction to a smooth and bounded $R \subset \Sigma$, $\partial R \neq \emptyset$, one recognizes that the correct moment map for the action of $\mathcal{G}$ is not directly given by the Gauss constraint, but rather by its "integration by parts:"

$$\langle J, \bullet \rangle = \int_R \sqrt{g}\, \mathrm{Tr}(E^i \mathrm{D}_i \bullet) = \int_R \sqrt{g}\, \mathrm{Tr}(\mathsf{G}\bullet) + \oint_{\partial R} \sqrt{g}\, \mathrm{Tr}(E_s \bullet). \tag{108}$$

On the one hand, this equation makes clear that, in the presence of boundaries, one cannot be satisfied by performing the MWM symplectic reduction solely at $J = 0$, for this would demand not only the vanishing of the Gauss constraint in the interior of $R$, but also the vanishing of the electric flux through $\partial R$—a condition which is physically overly restrictive.

On the other hand, performing the MWM through the shifting trick at a generic orbit $[j]$ is also not physically satisfactory, for choosing a generic orbit $[j]$ would introduce violations of the Gauss constraint in the interior of $R$.

In fact, the ideal solution sits somewhere in between these two extremes: heuristically, one would like to focus precisely on those "distributional" orbits $[j]$ such that $j \in [j]$ takes nontrivial values at the boundary $\partial R$ only. In other words, one would like to fix the above bulk integral to zero, while letting the flux $E_s$ to belong to the (coadjoint) orbit of a *certain* nonvanishing electric flux $f \neq 0$. Rather symbolically, this could be written as $[j] = [f]$.

By a careful separation of the radiative and Coulombic components of the electric field, as done in the main body of this article, one can in principle isolate the bulk and boundary components of $J$ and thus provide—through the application of the "shifting trick" to the boundary component only—an alternative, albeit ultimately equivalent, explanation of equations (1) or (69), and hence of the symplectic reduction by flux superselection sectors advocated for in this article.

The reader wishing to make the connection with the MWM paradigm of symplectic reduction more explicit can find in [20] a presentation of many of the results of this paper which follows much more closely the MWM paradigm (but contains no *explicit* reference to the shifting trick). That article provides an elementary (and much less general) summary of the results of this article and of [5].

---

[68]If $\Phi$ was enlarged to contain charged matter fields, then $J$ would be equal to the full Gauss constraint featuring their charge density $\rho$.

[69]Unless one is willing to consider the presence of non-dynamical charges—i.e. charged particles that source Gauss without being themselves included in the phase space of the theory.

## A.2 Uniqueness of $E_{\mathrm{Coul}}$

The goal of this appendix is to prove the uniqueness of $E_{\mathrm{Coul}}$ as a solution to the Gauss constraint $\mathsf{G}^f$ (31) at irreducible configurations of $A$.

**Definition A.1** (Reducibility parameters). *$\chi \in \mathrm{Lie}(\mathcal{G})$ is said a* reducibility parameter *for $A$ if and only if it is such that $\mathrm{D}\chi := \mathrm{d}\chi + [A, \chi] = 0$.*

Reducibility parameters are to YM configurations what Killing vector fields are to metrics in General Relativity: *global* symmetries. The set of reducibility parameters of a configuration $A$ forms a vector space (and in fact a Lie algebra) whose dimension is necessarily *finite* and bounded by $\dim(G)$. This dimension is maxed out by vacuum configurations with $F[A] = 0$.

**Definition A.2** (Irreducible configurations of $A$). *A configuration of the gauge potential $A \in \mathcal{A}$ is said* irreducible *if and only if its only reducibility parameter is the vanishing one, $\chi = 0$.*

In Abelian theories all configurations are reducible (consider $\chi = const$). In non-Abelian theories, on the other hand, irreducible configurations constitute a dense set in $\mathcal{A}$. See footnote 16.

**Definition A.3** (SdW boundary value problem—cf. section 2.2). *Given a region $R$, the following elliptic boundary value problem for the $\mathrm{Lie}(G)$-valued function $\xi$*

$$\begin{cases} \mathrm{D}^2 \xi = \alpha & \text{in } R, \\ \mathrm{D}_s \xi = \beta & \text{at } \partial R, \end{cases}$$

*is called a* Singer–DeWitt (SdW) boundary value problem *with bulk source $\alpha$ and boundary condition $\beta$ (both valued in $\mathrm{Lie}(G)$).*

**Lemma A.1** (Kernel of the SdW boundary value problem). *The kernel of the SdW boundary value problem at $A \in \mathcal{A}$ is given by the irreducibility parameters of $A$.*

*Proof.* First notice that, by definition, a $\mathrm{Lie}(\mathcal{G})$-valued variable $\xi$ is in the kernel of the SdW boundary value problem if and only if

$$\begin{cases} \mathrm{D}^2 \xi = 0 & \text{in } R, \\ \mathrm{D}_s \xi = 0 & \text{at } \partial R. \end{cases}$$

Clearly any $\xi$ which is a reducibility parameter of $A$ satisfies this condition. To see why the converse is also true notice that from this equation one deduces

$$0 = -\int \sqrt{g} \,\mathrm{Tr}(\xi \mathrm{D}^2 \xi) + \oint \sqrt{h} \,\mathrm{Tr}(\xi \mathrm{D}_s \xi) = \int \sqrt{g} \, g^{ij} \mathrm{Tr}(\mathrm{D}_i \xi \mathrm{D}_j \xi) = \mathbb{G}(\xi^\sharp, \xi^\sharp),$$

which vanishes if and only if $\mathrm{D}\xi = 0$, i.e. if and only if $\xi$ is a reducibility parameter of the configuration $A$. $\qquad\square$

**Proposition A.1** (Uniqueness of $E_{\mathrm{Coul}}$). *Suppose that $A \in \mathcal{A}$ is an irreducible. Then, for any choice of functional connection $\varpi$ and electric flux $f$, the Gauss constraint $\mathsf{G}_f = 0$ has one and only one solution $E_{Coul} = E_{Coul}(A, \rho, f)$.*

*Proof.* The proof of this statement proceeds in two steps. In the first step I prove the existence and uniqueness of the solution to the Gauss constraint for the SdW choice of connection, i.e. $\varpi = \varpi_{\mathrm{SdW}}$. In the second step, I show that this result can be used to prove existence and uniqueness for any other choice of connection.

*Part 1.* For the SdW choice of connection $E^i_{\text{Coul}} = g^{ij}D_j\varphi$, the Gauss constraint becomes a SdW boundary value problem:

$$\begin{cases} D^2\varphi = \rho & \text{in } R, \\ D_s\varphi = f & \text{at } \partial R \end{cases} \qquad \text{(SdW)}.$$

From the invertibility of the SdW boundary value problem at irreducible configurations (Lemma A.1), we deduce existence and uniqueness of $\varphi$.

*Part 2.* Consider now an arbitrary connection $\varpi' = \varpi_{\text{SdW}} + \nu$ where $\nu$ is a horizontal and covariant 1-form in $\Omega^1(\mathcal{A}, \text{Lie}(\mathcal{G}))$, i.e. for any field-dependent $\xi$, $\mathbb{i}_{\xi^\sharp}\nu = 0$ and $\mathbb{L}_{\xi^\sharp}\nu = [\nu, \xi]$ (see (4)).

Denoting with a prime (e.g. $E'_{\text{Coul}}$) the quantities constructed from $\varpi'$ rather than $\varpi_{\text{SdW}}$, solving the Gauss constraint for $E'_{\text{Coul}}$ means solving the following system of equations:

$$\begin{cases} D_i(E'_{\text{Coul}})^i = \rho & \text{in } R, \\ s_i(E'_{\text{Coul}})^i = f & \text{at } \partial R, \\ \int \sqrt{g}\,\text{Tr}((E'_{\text{Coul}})^i \mathbb{d}_{H'}A_i) = 0, \end{cases} \qquad \text{(A1)}$$

where the last equation is a rewriting of (27).

Now, decompose $E = E'_{\text{Coul}}$ into its SdW-radiative ($\varepsilon_{\text{rad}}$) and SdW-Coulombic ($D\gamma$) componets, that is $(E'_{\text{Coul}})^i = \varepsilon^i_{\text{rad}} + g^{ij}D_j\gamma$. Also, observe that $\mathbb{d}_{H'}A = \mathbb{d}_\perp A - D\nu \equiv \mathbb{d}_\perp A - D\nu(\widehat{H}_{\text{SdW}}(\cdot))$, where the last equality follows from $\mathbb{i}_{\xi^\sharp}\nu = 0$. Inserting these formulas in the system of equations above, and using (28) for $\varepsilon_{\text{rad}}$, one obtains:

$$\begin{cases} D^2\gamma = \rho & \text{in } R, \\ D_s\gamma = f & \text{at } \partial R, \\ \int \sqrt{g}\,\text{Tr}(\varepsilon^i_{\text{rad}}\mathbb{d}_\perp A_i) = \int \sqrt{g}\,\text{Tr}(D^i\gamma D_i \nu(\widehat{H}_{\text{SdW}}(\cdot))). \end{cases} \qquad \text{(A2)}$$

From Part 1, $\gamma$ is uniquely determined by $A$, $\rho$ and $f$. To finish the uniqueness proof for $E'_{\text{Coul}}$, the component $\varepsilon^i_{\text{rad}}$ must also be shown unique from the last equation of (A2).

From the existence and uniqueness of $\gamma$, the right hand side of that equation yields a well-determined SdW-horizontal one-form $\alpha_\perp := \int \sqrt{g}\,\text{Tr}(D^i\gamma D_i \nu(\widehat{H}_{\text{SdW}}(\cdot)) \in T^*\Phi$.

Since $\varepsilon_{\text{rad}}$ is by construction radiative (i.e. satisfies (28)), one sees that

$$\int \sqrt{g}\,\text{Tr}(\varepsilon^i_{\text{rad}}\mathbb{d}_\perp A_i) \equiv \int \sqrt{g}\,\text{Tr}(\varepsilon^i_{\text{rad}}\mathbb{d}A_i)$$

and therefore from the last of (A2) it follows that $\varepsilon^i_{\text{rad}}$ is nothing but the "component" description of the 1-form $\alpha_\perp$. Since $\alpha_\perp$ is uniquely defined, so must be $\varepsilon^i_{\text{rad}}$.

Thus, having uniquely determined $\gamma$ and $\varepsilon_{\text{rad}}$ in terms of $(A, \rho, f)$, we have uniquely determined $(E'_{\text{Coul}})^i = \varepsilon^i_{\text{rad}} + g^{ij}D_j\gamma$ as well. This concludes the proof. $\qquad \square$

Note that, from (A1) and (A2), $\gamma = \varphi$ and therefore the difference between the Coulombic modes associated to two different functional connections is always radiative, i.e. $E'_{\text{Coul}} - E_{\text{Coul}} = \varepsilon_{\text{rad}}$.

## A.3 The kernel of $\iota^*\omega^H$: proof of (40)

The goal of this appendix is to prove (40) which states that

$$\ker(\iota^*\Omega^H) = V^{[f]} \oplus Y^{[f]}.$$

(In this appendix, as in the main body of this article, I neglect *reducible* configurations; cf. appendix A.2, footnote 16, and the comment at the end of section 4.9.)

**Lemma A.2.** *Given a choice of functional connection $\varpi$, let $\phi$ be an on-shell configuration $\phi \in \{G^f = 0\}$ and $\mathbb{X} \in \iota_*(\mathrm{T}_\phi \Phi^{[f]})$ a variation tangent to a CSSS (i.e. $\mathbb{X}$ this preserves the validity of the Gauss constraint but only the equivalence class of the flux $f$). Denote $\eta := \varpi(\mathbb{X})$ and $\delta_\mathbb{X} f := \mathbb{X}(f)$. Suppose that $\mathbb{X}$ acts on all field components[70] except $E_{Coul}$ as the gauge transformation $\eta$ would: that is $\mathbb{X}(\bullet) = \eta^\sharp(\bullet)$ for $\bullet \in \{A, E_{rad}, \psi, \overline{\psi}\}$. Then, $\mathbb{X}$ is uniquely determined by $\eta$ and its own action on $f$, according to the formula $\mathbb{X} = \eta^\sharp + \mathbb{Y}$ where*

   *(i)* $\mathbb{Y}$ *is a functional of* $(\delta_\mathbb{X} f + \mathrm{ad}_{\eta|\partial R} f)$, *i.e.* $\mathbb{Y} := \mathbb{Y}[\,\delta_\mathbb{X} f + \mathrm{ad}_{\eta|\partial R} f\,]$;

   *(ii)* $\mathbb{Y} = 0$ *if and only if* $\delta_\mathbb{X} f = [f, \eta_{|\partial R}]$;

   *(iii)* $\mathbb{Y}$ *is tangent to* $\Phi^{[f]} \subset \Phi$;

   *(iv)* $\mathbb{Y}$ *is $\varpi$-horizontal, i.e.* $\varpi(\mathbb{Y}) = 0$;

   *(v) Finally, if* $\varpi = \varpi_{SdW}$, *then*

$$\mathbb{Y}^{(SdW)} = \int \zeta \frac{\delta}{\delta\varphi} \quad where \quad \begin{cases} \mathrm{D}^2 \zeta = 0 & in\ R, \\ \mathrm{D}_s \zeta = \delta_\mathbb{X} f - [f, \eta_{|\partial R}] & at\ \partial R \end{cases} \qquad (SdW). \qquad (A3)$$

*Proof.* Note that $\mathbb{X}$ is uniquely determined if so is its action on on the remaining coordinate on $\Phi$, i.e. $E_{\mathrm{Coul}}$. Therefore, the uniqueness of $\mathbb{X}$ as a functional of $\eta$ and $\delta_\mathbb{X} f$ is a corollary of proposition A.1 which states the uniqueness of the solution of the Gauss constraint $E_{\mathrm{Coul}}$. Indeed, if $(A, \rho, f)$ determine $E_{\mathrm{Coul}}$ uniquely then these quantities and their first order variations, that is $(\mathbb{X}(A), \mathbb{X}(\rho), \mathbb{X}(f))$, uniquely determine the first order variation of $E_{\mathrm{Coul}}$.

Therefore, let me prove that $\mathbb{X}(E_{\mathrm{Coul}})$ is uniquely determined in terms of $\eta$ and $\delta_\mathbb{X} f$. For this, consider first the variation of[71] (A1) along an *arbitrary* direction $\mathbb{X}$:

$$\begin{cases} \mathrm{D}_i \mathbb{X}(E^i_{\mathrm{Coul}}) = \mathbb{X}(\rho) - [\mathbb{X}(A_i), E^i_{\mathrm{Coul}}] & in\ R, \\ s_i \mathbb{X}(E^i_{\mathrm{Coul}}) = \mathbb{X}(f) & at\ \partial R, \\ \mathbb{L}_\mathbb{X} \int \sqrt{g}\, \mathrm{Tr}(E^i_{\mathrm{Coul}} \mathbb{d}_H A_i) = 0. \end{cases}$$

The last equation states that variations along $\mathbb{X}$ do not alter the fact that $E_{\mathrm{Coul}}$ satisfies its defining functional property, that is $\int \sqrt{g}\, \mathrm{Tr}(E^i_{\mathrm{Coul}} \mathbb{d}_H A_i) = 0$ (see the proof of (A.1)). Now, specializing to a configuration $\phi \in \Phi_G$ that satisfies the Gauss constraint and $\mathbb{X}$ that satisfies the hypothesis of the proposition, the above simplifies to

$$\begin{cases} \mathrm{D}_i \mathbb{X}(E^i_{\mathrm{Coul}}) = [\rho, \eta] - [\mathrm{D}_i \eta, E^i_{\mathrm{Coul}}] = \mathrm{D}_i [E^i_{\mathrm{Coul}}, \eta] & in\ R, \\ s_i \mathbb{X}(E^i_{\mathrm{Coul}}) = \delta_\mathbb{X} f & at\ \partial R, \\ \int \sqrt{g}\, \mathrm{Tr}(\mathbb{X}(E^i_{\mathrm{Coul}}) \mathbb{d}_H A_i) + \int \sqrt{g} E^i_{\mathrm{Coul}}[\mathbb{d}_H A_i, \eta]) = 0, \end{cases}$$

where I used that $\varpi$ is defined as a pullback from $\mathcal{A}$ and that $\mathbb{L}_{\eta^\sharp} \mathbb{d}_H A = [\mathbb{d}_H A, \eta]$. Introducing

$$\delta_\mathbb{Y} E^i_{\mathrm{Coul}} := \mathbb{X}(E^i_{\mathrm{Coul}}) - [E^i_{\mathrm{Coul}}, \eta]$$

the above system of equations can be rewritten as

$$\begin{cases} \mathrm{D}_i \delta_\mathbb{Y} E^i_{\mathrm{Coul}} = 0 & in\ R \\ s_i \delta_\mathbb{Y} E^i_{\mathrm{Coul}} = \delta_\mathbb{X} f - [f, \eta_{|\partial R}] & at\ \partial R \\ \int \sqrt{g}\, \mathrm{Tr}(\delta_\mathbb{Y} E^i_{\mathrm{Coul}} \mathbb{d}_H A_i) = 0. \end{cases}$$

---

[70]Seen as (coordinate) *functions* on field space, so that e.g. $\mathbb{X}(A) = X_A$ for $\mathbb{X} = \int X_A \frac{\delta}{\delta A} + \dots$.

[71]Here we suppress the prime, $E'_{\mathrm{Coul}} \rightsquigarrow E_{\mathrm{Coul}}$.

To conclude, refer to proposition A.1 (cf. (A1)) to deduce that $\delta_{\mathbb{Y}} E^i_{\text{Coul}}$ is uniquely determined by $\delta_{\mathbb{X}} f$ and $\eta_{|\partial R}$. Thus, whereas $\mathbb{X}$ acts on $(A, E_{\text{rad}}, \psi, \overline{\psi})$ as a pure gauge transformation $\eta$ (by hypothesis), its action on $E_{\text{Coul}}$ is fully determined by the boundary value of $\eta$ and the action of $\mathbb{X}$ on the electric flux $f$. Therefore, one can write $\mathbb{X} = \eta^{\sharp} + \mathbb{Y}$ where the vector $\mathbb{Y}$ is defined by: $\mathbb{Y}(\bullet) = 0$ for $\bullet \in \{A, E_{\text{rad}}, \psi, \overline{\psi}\}$ and $\mathbb{Y}(E_{\text{Coul}}) = \delta_{\mathbb{Y}} E_{\text{Coul}}$ as determined above.

The vector $\mathbb{Y}$ is tangent to $\Phi^{[f]} \subset \Phi$ because both $\mathbb{X}$ and $\eta^{\sharp}$ are.[72] Moreover, $\mathbb{Y}$ is horizontal because the connection $\varpi \in \Omega^1(\Phi, \text{Lie}(\mathcal{G}))$ is defined as a pullback to $\Phi$ of a connection on $\mathcal{A}$: indeed, from this it follows that $\varpi(\mathbb{Y}) = \varpi(\mathbb{X}) - \varpi(\xi^{\sharp}) = \varpi((\mathbb{X})_A) - \xi = 0$. Finally, $\mathbb{Y}$ vanishes if and only if $\delta_{\mathbb{X}} f = [f, \xi_{|\partial R}]$, i.e. $\delta_{\mathbb{X}} f = [f, \varpi(\mathbb{X})_{|\partial R}]$, that is if and only if $f$ also transforms by the *same* gauge transformation as every other field.

Since $\mathbb{X}$ stays within the tangent of the covariant superselection sector, its action on $f$ must also be of the form $\delta_{\mathbb{X}} f = [f, \zeta'_{\partial}]$, for some $\zeta'_{\partial} \in \text{Lie}(\mathcal{G}_{|\partial R})$, where $\mathcal{G}_{|\partial R} := \{u_{\partial} \in C^{\infty}(\partial R, G) | \exists g \in \mathcal{G} \text{ such that } u_{\partial} = g_{|\partial R}\}$. But importantly, in general $\zeta'_{\partial} \neq \varpi(\mathbb{X})_{|\partial R} \equiv \eta_{|\partial R}$; e.g. $\zeta'_{\partial}$ can be non-zero even if $\varpi(\mathbb{X}) \equiv \eta = 0$. Thus, the result of the previous paragraph can be rephrased as stating that $\mathbb{Y}$ vanishes if and only if $\zeta'_{\partial} = \eta_{|\partial R}$.

Let me now specialize to $\varpi = \varpi_{\text{SdW}}$. In this case $E^i_{\text{Coul}} = D^i \varphi$ and the above system of equations becomes:

$$\begin{cases} D_i \delta_{\mathbb{Y}} E^i_{\text{Coul}} = 0 & \text{in } R, \\ s_i \delta_{\mathbb{Y}} E^i_{\text{Coul}} = \delta_{\mathbb{X}} f - [f, \eta_{|\partial R}] & \text{at } \partial R, \\ \int \sqrt{g} \, \text{Tr}(\delta_{\mathbb{Y}} E^i_{\text{Coul}} \mathbb{d}_{\perp} A_i) = 0 \end{cases} \quad \text{(SdW)}.$$

From the last equation, and the properties of the SdW split, it follows that $\delta_{\mathbb{Y}} E^i_{\text{Coul}} = D^i \zeta$ is a pure gradient. Plugging this relationship back into the first two equations one obtains a SdW boundary value problem for $\zeta$. Now, a vector $\mathbb{Y}$ annihilating $(A, E_{\text{rad}}, \psi, \overline{\psi})$, but not annihilating $E_{\text{Coul}}$, is proportional in the SdW basis to $\frac{\delta}{\delta \varphi}$. From $D^i \zeta = \mathbb{Y}(E^i_{\text{Coul}}) = \mathbb{Y}(D^i \varphi) = D^i \mathbb{Y}(\varphi)$, one finally deduces that $\mathbb{Y} = \int \zeta \frac{\delta}{\delta \varphi}$ as in (A3). $\qquad \square$

The main outcome of this lemma is the characterization of the vectors $\mathbb{Y}$ which we shall refer to as "flux rotations:"

**Definition A.4** (Flux rotations). *Given a functional connection $\varpi$ and a covariant superselection sector $\Phi^{[f]}$, call* flux rotations *$\mathbb{Y}_{\zeta_{\partial}} \in Y^{[f]} \subset T\Phi^{[f]}$ vectors $\mathbb{Y}_{\zeta_{\partial}} \in T_{\phi}\Phi^{[f]}$ defined by the following action on the coordinate functions $\{A, E_{rad}, \psi, \overline{\psi}, f\}$ on $\Phi^{[f]}$:*

$$\mathbb{Y}_{\zeta_{\partial}}(\bullet) = 0 \quad \text{for } \bullet \in \{A, E_{rad}, \psi, \overline{\psi}\}, \quad \text{and} \quad \begin{cases} D_i \mathbb{Y}_{\zeta_{\partial}}(E^i_{Coul}) = 0 & \text{in } R, \\ s_i \mathbb{Y}_{\zeta_{\partial}}(E^i_{Coul}) = -[f, \zeta_{\partial}] & \text{at } \partial R, \\ \int \sqrt{g} \, \text{Tr}(\mathbb{Y}_{\zeta_{\partial}}(E^i_{Coul}) \mathbb{d}_H A_i) = 0. \end{cases}$$

*With an slight abuse of language and notation, call also* flux rotations *vector fields over $\Phi^{[f]}$ which are sections of $Y^{[f]}$:*

$$\mathbb{Y}_{\zeta_{\partial}} \in \mathcal{Y}^{[f]} := \Gamma(\Phi^{[f]}, Y^{[f]}) \subset \mathfrak{X}^1(\Phi^{[f]}).$$

Note that the last of the equations defining $\mathbb{Y}_{\zeta_{\partial}}$ states that $\mathbb{Y}_{\zeta_{\partial}}(E_{\text{Coul}})$ is itself Coulombic. Therefore the defining equation for flux rotations has a unique solution for the very same reason that the Gauss constraint does—see Proposition A.1.

Note also that, as vector fields, flux rotation admit parameters $\zeta_{\partial}$ which are themseleves field-*dependent* parameters valued in $\text{Lie}(\mathcal{G}_{\partial R})$, i.e. $\zeta_{\partial} \in \Gamma(\Phi^{[f]}, \Phi^{[f]} \times \text{Lie}(\mathcal{G}_{\partial R}))$.

---

[72]Note, however, that nowhere in the proof of the lemma $\delta_{\mathbb{X}} f$ was required to be of the form $[f, \zeta'_{\partial}]$. Indeed, the lemma would work in precisely the same way for the more general vectors tangent to $\Phi_G$ rather than $\Phi^{[f]}$.

Let me now collect two important properties enjoyed by flux rotations in the following proposition (whose proof is trivial at the light of Lemma A.2 and Definition A.4):

**Proposition A.2.** *Flux rotations* $\mathbb{Y} \in \mathcal{Y}^{[f]} \subset \mathfrak{X}^1(\Phi^{[f]})$ *satisfy the following properties:*

*(i) they are horizontal,* $\varpi(\mathbb{Y}_{\zeta_\partial}) = 0$, *and*

*(ii) if G is Abelian, flux rotations are trivial,* $\mathcal{Y}^{[f]} = \{0\}$.

Now, thanks to the above lemma and definition, it is possible to finally characterize the degeneracy properties of $\Omega^H$ in a covariant superselection sector. As expected gauge transformations (i.e. vertical vectors) are in the kernel of $\iota^*\Omega^H$. But so are *flux rotations*, which indeed constitute the most interesting part of this kernel:

**Proposition A.3** (The kernel of $\iota^*\Omega^H$). *Given a choice of functional connection $\varpi$, in the covariant superselection sector $\Phi^{[f]}$ one has*

$$\ker(\iota^*\Omega^H) = V^{[f]} \oplus Y^{[f]},$$

*where $V^{[f]} = \bigcup_{\phi \in \Phi^{[f]}} \mathrm{T}\mathcal{O}_\phi$ is the space of vertical vector fields in $\mathrm{T}\Phi^{[f]}$, and $Y^{[f]} \subset \mathrm{T}\Phi^{[f]}$ is the space of flux rotations over $\Phi^{[f]}$.*

*Proof.* A vector $\mathbb{X} \in \mathrm{T}_\phi \Phi^{[f]}$ is in the kernel of $\iota^*\Omega^H$ if and only if (iff) $\iota^*\Omega^H(\mathbb{X}) = 0$, i.e. iff $\iota^*(\Omega^H(\iota_*\mathbb{X})) = 0$, i.e. iff (see (37))

$$\iota^* \int \sqrt{g}\, \mathrm{Tr}\big(-h_A \mathbb{d}_H E_{\mathrm{rad}} + h_{\mathrm{rad}} \mathbb{d}_H A\big) + \iota^* \int \sqrt{g}\, \big(\mathbb{d}_H \overline{\psi}\gamma^0 h_\psi - h_{\overline{\psi}}\gamma^0 \mathbb{d}_H \psi\big) = 0,$$

where we set $h_\bullet := \mathring{\mathbb{i}}_{\iota_*\mathbb{X}} \mathbb{d}_H \bullet \equiv \mathring{\mathbb{i}}_{\widehat{H}(\iota_*X)} \mathbb{d} \bullet \equiv (\widehat{H}(\iota_*\mathbb{X}))_\bullet$ for $\bullet \in \{A, E_{\mathrm{rad}}, \psi, \overline{\psi}\}$.

Since $\iota^* \mathbb{d}_H A$, $\iota^* \mathbb{d}_H E_{\mathrm{rad}}$, $\iota^* \mathbb{d}_H \psi$, $\iota^* \mathbb{d}_H \overline{\psi}$ are all independent from each other, this expression vanishes identically iff $h_\bullet = 0$ for all $\bullet$ as above. Therefore the only horizontal component of $\iota_*\mathbb{X}$ that can survive is $h_{\mathrm{Coul}} := \mathring{\mathbb{i}}_{\iota_*\mathbb{X}} \mathbb{d}_H E_{\mathrm{Coul}}$.

In view of the horizontal/vertical decomposition of $\iota_*\mathbb{X}$, the statement that $h_\bullet = 0$ is equivalent to demanding $(\iota_*\mathbb{X})_\bullet = (\xi^\sharp)_\bullet$ or equivalently that $(\iota_*\mathbb{X})(\bullet) = \xi^\sharp(\bullet)$, for $\bullet$ as above and $\xi := \varpi(\iota_*\mathbb{X}) = \varpi((\iota_*\mathbb{X})_A)$—the last equality follows from the fact that the functional connection $\varpi \in \Omega^1(\Phi, \mathrm{Lie}(\mathcal{G}))$ is defined by pullback of a functional connection on $\mathcal{A}$.

This means that $\iota_*\mathbb{X}$ satisfies the conditions under which Lemma A.2 holds. Thus, from Lemma A.2, Definition A.4, and the arguments above, it follows that if $\mathbb{X} \in \ker(\iota^*\Omega^H)$ then $\iota_*\mathbb{X} = \eta^\sharp + \mathbb{Y}_{\zeta_\partial}$ for $\zeta_\partial = \eta_{|\partial R} - \zeta'_\partial$ in the notation of the lemma. Hence, $\ker(\iota^*\Omega^H) \subset V^{[f]} \oplus Y^{[f]}$

To conclude the proof, it is enough to observe that any vector in $V^{[f]}$ or in $Y^{[f]}$ is in the kernel of $\Omega^H$: the first case is obvious, the second follows from the fact that flux rotations $\mathbb{Y}_{\zeta_\partial} \in Y^{[f]}$ only act on the Coulombic component of the electric field which is not featured in $\Omega^H$ (cf. (37)). $\qquad\square$

**Corollary A.3.1.** *Suppose the hypotheses of proposition A.3 hold. If moreover G is Abelian or f is trivial (either because $f = 0$ or because $\partial R \neq \emptyset$), then $\ker(\iota^*\Omega^H) = V$.*

*Proof.* If $G$ is Abelian or $f$ is trivial, then it is immediate to see (e.g. from lemma A.2) that $\mathbb{Y} \equiv 0$. Hence $Y^{[f]} = \{0\}$ is trivial and $\ker(\iota^*\Omega^H) = V$. $\qquad\square$

## A.4  The kernel of $\Omega_{\text{ext}}^{H,[f]}$: proof of (63)

The goal of this appendix is to prove (63) which states that

$$\ker(\Omega_{\text{ext}}^{H,[f]}) = V_{\text{ext}}.$$

**Proposition A.4** (The kernel of $\Omega_{\text{ext}}^{H,[f]}$). *In the covariant superselection sector $[f]$, one has*

$$\ker(\Omega^{H,[f]}) = V_{ext},$$

*where $V_{ext} = \text{Span}\{(\xi^\sharp, \sigma_o^\S)\} \subset \text{T}\Phi_{ext}^{[f]}$ is the space of vertical vector fields in $(\Phi_{ext}^{[f]}, \Pi)$; i.e. $V_{ext} = \text{Lie}(\mathcal{G})^\sharp \oplus \text{Lie}(\mathcal{G}_{|\partial R}^o)^\S$ is the direct sum of pure gauge transformations and flux-reference stabilizer transformations. (Cf. section 4.8.)*

*Proof.* Recall (61) and (68):

$$\Omega_{\text{ext}}^{H,[f]} = \pi_o^* \iota^* \Omega^H + \omega_{\text{ext}}^{H,[f]} = \pi_o^* \iota^* \Omega + \omega^{[f]}.$$

One can easily check that $V^{\text{ext}} \subset \ker(\Omega^{H,[f]})$ because by construction both $\pi_o^* \iota^* \Omega^H$ and $\omega_{\text{ext}}^{H,[f]}$ are gauge-horizontal, and both $\pi_o^* \iota^* \Omega$ and $\omega^{[f]}$ are flux-reference-stabilizer horizontal.

One is left to prove that $\ker(\Omega^{H,[f]}) \subset V_{\text{ext}}$. This can be done by adapting the argument put forward in the proof to Proposition A.3. Using the same notation as there: $\mathbb{X} \in \ker(\Omega^{H,[f]})$ iff

$$0 = \Omega_{\text{ext}}^{H,[f]}(\mathbb{X}) = \iota^* \int \sqrt{g} \, \text{Tr}\big(- h_A \mathbb{d}_H E_{\text{rad}} + h_{\text{rad}} \mathbb{d}_H A\big) + \iota^* \int \sqrt{g} \, \big(\mathbb{d}_H \overline{\psi} \gamma^0 h_\psi - h_{\overline{\psi}} \gamma^0 \mathbb{d}_H \psi\big)$$

$$+ \oint \sqrt{h} \, \text{Tr}\big([h_u, f_o] u^{-1} \mathbb{d}_H u + f_o u^{-1} \mathbb{F}(\iota_* \mathbb{X}) u\big), \tag{A4}$$

where $h_u := u^{-1} \mathbb{i}_{\mathbb{X}} \mathbb{d}_H u \equiv u^{-1} \mathbb{i}_{\widehat{H}(\mathbb{X})} \mathbb{d} u \equiv u^{-1} \widehat{H}(\mathbb{X})_u$. Notice that $\mathbb{d}_H f = \text{Ad}_u[u^{-1} \mathbb{d}_H u, f_o]$.

Therefore, if $\mathbb{X} \in \ker(\Omega^{H,[f]})$, the above expression must vanish when contracted with *any* vector $\mathbb{X}' \in \text{T}_\phi \Phi_{\text{ext}}^{[f]}$.

First, consider the vector $\mathbb{X}^1$ defined by $\mathbb{X}^1(E_{\text{rad}}) = X_{\text{rad}}^1$ and $\mathbb{X}^1(\bullet) = 0$ for $\bullet \in \{A, \psi, \overline{\psi}, u\}$. Notice that, since $E_{\text{rad}}$ does not participate to the Gauss constraint, this vector is indeed tangent to $\Phi_{\text{ext}}^{[f]}$. Moreover, since $(\mathbb{X}^1)_A = 0$ this vector is also horizontal. Therefore, $0 = \Omega_{\text{ext}}^{H,[f]}(\mathbb{X}, \mathbb{X}^1) = \iota^* \int \sqrt{g} \, \text{Tr}(-h_A X_{\text{rad}}^1)$. From the arbitrariness of $X_{\text{rad}}^1$, one concludes that $h_A$ vanishes.

Since $\iota_* \mathbb{X}(A) = h_A = 0$, $\mathbb{F}(\iota_* \mathbb{X}) = 0$ too. Hence, equation (A4) reduces to

$$0 = \Omega_{\text{ext}}^{H,[f]}(\mathbb{X}) = \iota^* \int \sqrt{g} \, \text{Tr}\big(h_{\text{rad}} \mathbb{d}_H A\big) + \iota^* \int \sqrt{g} \, \big(\mathbb{d}_H \overline{\psi} \gamma^0 h_\psi - h_{\overline{\psi}} \gamma^0 \mathbb{d}_H \psi\big)$$

$$+ \oint \sqrt{h} \, \text{Tr}\big([h_u, f_o] u^{-1} \mathbb{d}_H u\big). \tag{A5}$$

From the independence of $\mathbb{d}_H A$, $\mathbb{d}_H \psi$ and $\mathbb{d}_H \overline{\psi}$ one similarly concludes that $h_{\text{rad}}$, $h_\psi$ and $h_{\overline{\psi}}$ also vanish.

Finally, for the last term in (A5) to vanish identically, one must demand that $[h_u, f_o] = 0$ i.e. that $h_u \in \text{Lie}(\mathcal{G}_{|\partial R}^o)$.

Therefore one concludes that if $\mathbb{X} \in \ker(\Omega^{H,[f]})$, then $\widehat{H}(\mathbb{X}) \in \text{Lie}(\mathcal{G}_{|\partial R}^o)^\S$ i.e. that $\mathbb{X}$ is either vertical or a flux-stabilizer transformations. In formulas, $\mathbb{X} \in V_{\text{ext}}$. $\qquad\square$

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
