# Peer review of "Symplectic reduction of Yang-Mills theory with boundaries: from superselection sectors to edge modes, and back"

_SciPost Physics, doi:SciPost Phys. 10, 125 (2021)_

## Round 1 · Referee Report · Nicholas Teh (Referee 1) · 2021-4-22

Strengths

1.This paper presents a genuinely novel point of view on how to identify quasilocal degrees of freedom for Yang-Mills gauge theory, thus completing a project initiated by Gomes and Riello in a series of interesting investigations into the field space geometry of gauge theory in the presence of spatial boundaries. This identification is done by the introduction of covariant superselection sectors and an additional KKS (canonical) symplectic form in each sector.

  1. The presentation is clear throughout: the calculations are reasonably explicit so the reader can check them, and when they are not, multiple references are given. The geometrical foundations of the theory are clearly explained.

Weaknesses

  1. The greatest weakness of the paper is that, despite calling the procedure "symplectic reduction", it is unclear how this procedure relates to the usual symplectic geometry constructions of reduction that involve a moment map. One simple example that it might be helpful for the author to briefly comment on for the reader is how this form of symplectic reduction relates to the construction given in Atiyah and Bott's "The Moment Map and Equivariant Cohomology" (which results in a finite-dimensional moduli space) -- this would also help establish a link between standard equivariant constructions and the equivariant symplectic geometry introduced by the author.

  2. The geometrical formalism used in this paper (sans field space connection) is generally introduced in the context of the covariant phase space approach: it would be nice if the author could briefly explain how the spatial constructions in this paper relate to a fully covariant approach.

  3. In criticizing the approach of Donnelly and Freidel, I would recommend that the author update his criticism to deal with the more developed point of view on edge modes offered in https://arxiv.org/pdf/2006.12527.pdf

Report

I would recommend publication in this journal subject to minor corrections.

Requested changes

I would ask that the author briefly comment on points (1), (2) and (3) of the Weaknesses section in this paper.

---

## Round 1 · Referee Report · Anonymous (Referee 2) · 2021-5-8

Report

Strengths: This is very clearly written, and well worked out, account of symplectic reduction for YM theories, in the context of boundaries. It extends (?and completes?) an impressive line of work by the author and others over some five years. It discusses well, and convincingly, the relation to other approaches and constructions, especially by Friedel and co authors (beginning, so far as I know, with Donnelly and Friedel in 2016). This is the topic of Section 5.

Weaknesses: 1: About content: I think the only weakness is that I would like the Section 5 discussion of the relation to Friedel and co authors to say some more details about the relation to their more recent papers, which are cited as [27] to [29] on p. 23 but not discussed In the present draft, the only real point of reference is Donnelly and Friedel in 2016 (the author’s [26]).

2: About language. There are slight lapses of English throughout; but one can always reconstruct the meaning. Examples at the beginning and end are as follows.

Sec 1.1, paragraph 3, line 1: say: perspective, what is responsible for this ETC Sec 1.1, paragraph 4, line 3: say: start with the second task Sec 1.2, paragraph 2, line 2: say: concentrate one’s efforts Sec 1.2, last paragraph (p.3), line 1: say: The fixing of a (covariant) superselection sector ETC Sec 1.3, paragraph 3, last line: say: denoting by s^i

Sec 1.3, last paragraph (page 4), lines 3 to 5: about the introductory intuitive statement of superselection. (A): line 3 to 4: you say: quantum states cannot be stirred …any physical operation: This is too loose (if true!). I would just say the obvious, like: 0 matrix elements between the sectors, all observables block diagonalized. And (B): line 5: you say: theory factorizes … sectors labelled ETC Again too loose. So I suggest instead, you should say: and the theory’s states are (decompose into) statistical mixtures of states in sectors labelled ETC.

And at end: Sec 5.8, last paragraph (page 31), line 1: replace: At the light of this: by: In the light of this Sec 5.9, paragraph 2(page 31), line 4: replace: but by far non-unique: with EITHER: but very non-unique: OR: but far from unique. Section 6, last line, p. 32: should say: mismatches to be a neat example

The author may find it best to get an academic native speaker of English to go through and fix these minor matters of language.

Also: Souriau has an A in his name; so not: Sourieu!

I would certainly recommend publication, subject to minor revisions

---

## Round 2 · Author Response

ANONYMOUS REFEREE

I would like to thank the referee for their report, and for catching typos big and small. Regarding the discussion on the more recent works of Freidel et al, I left it out on purpose for a few reasons I am now going to spell out.

First, the literature on edge modes is quite vast at this point and different authors sometimes hold slightly different perspectives on them. Since the goal of this work was not to review all the different viewpoints, I simply picked the one that seems to have become the standard reference.

Second, in the more recent work of Freidel himself and collaborators, I haven’t noticed any substantial difference in philosophy or mathematical setup—but, if the referee think I am mistaken, I’d be more than happy to rectify my understanding and comment on any more specific point.

Third, often the edge mode literature is not very precise in relation to what their prescriptions entail with respect to symplectic reduction: they simply do not ask the question in these terms (this is in my opinion an important gap that this work aims to fill). For this reason, in writing this work, I already had to extrapolate from Freidel and Donnelly. This extrapolation was not particularly involved, but there is nonetheless a risk of misinterpretation every time such a step is taken and I am therefore wary of generalizing my comments too much—even if I have still the impression most works are quite consistent with each other in this regard (see points 1 and 2).

Fourth and last, the more recent papers of Freidel et al (and in fact of most other authors) focus on gravity and not YM theory, which is the only theory studied in my own work. This point is relevant because the interplay of diffeomorphisms and boundaries/corners is much more intricate than that of “standard” gauge symmetries. E.g. there is no standard moment map for diffeos transverse to the Cauchy surface or, in a quasilocal setting, for those transverse to the corners. Therefore, even though most of the work done by Freidel et al de facto sets aside these possibilities by focusing on diffeomorphisms which are either tangent to the boundary or vanishing there (at zeroth order), I would still prefer not to put forward generic statements on symplectic reduction in gravitational theories that I cannot back with solid mathematical arguments.

This said, I do think that some of those works, e.g. the recent one by Donnelly, Freidel and Speranza, is most likely going to be relevant for the definition of superselection sectors in GR (even though they haven’t framed it in such terms).

I hope this is enough to justify my choice of leaving out comments on more recent works on edge modes, but if the referee thinks otherwise and reckons that there are specific points worth commenting in the article, I’d be happy to include a discussion about those.

PROF. TEH

I would like to thank prof. Teh for his review and for going one step further and disclosing his identity. I will answer to his criticisms here below.

  1. Indeed, prof. Teh is absolutely right on this point and I want to thank him for catching this omission: leaving out a discussion of the relationship with the Marsden-Weinstein-Meyer (MWM) symplectic reduction was a blatant oversight on my part. The connection between my reduction prescription and the one of MWM can indeed be drawn, and is now sketched at the end of section 1.5 and in the appendix A.1. Unfortunately, being precise and explicit about this connection would require a consistent expansion of my article, which seems to me not warranted at this point. However, I believe that the material now included in the appendix is enough to make the connection more than plausible. Also, in my most recent preprint (arXiv:2104.10182) many of the topics presented here are summarized in a more elementary (but less general) way that follows the MWM paradigm more closely.

  2. Making explicit contact with the covariant phase space approach would indeed be nice. Unfortunately, this requires a careful analysis of the equations of motion in finite regions, which is not completely straightforward and would thus require a new set of notations and techniques in order to discuss these matters at a satisfactory level of detail comparable to the rest of the paper. Personally, I do not have full control of the details yet, the problem being (once again!) how to deal with the Gauss constraint appropriately.

I can however share a few thoughts which I have put forward in a recent preprint of mine which is more discursive and less encompassing in nature (arXiv:2104.10182). There, I argue that the correct way of thinking of the symplectic data over a finite subregion R of a Cauchy surface is not in terms of the evolution along a “spacetime cylinder” C = R x (-1,1), but rather in the causal domain (or diamond’’) D(R). Ignoring the subtleties coming from the elliptical Gauss constraint, from a (hyperbolic, Lorentz invariant) PDE perspective this statement is rather obvious: R supports precisely the dof sufficient to reconstruct the dynamics in D(R), not in C. This perspective also helps us make sense of the superselection of the flux f, which is now attached to thebelt’’ $B = \partial R$ of the diamond D(R) in a completely spacetime invariant way, since the flux can be written as the pullback to $B$ of the (dualized) field strength: $f = \iota_B^* (\star F)$. As I have already said, however, making these qualitative statements more rigorous requires a thorough study of the Maxwell-Yang-Mills equations in D(R), that I do not fully control and feel would go beyond the present scope of the paper.

This is why I decided that dealing with the kinematical phase space $\Phi = T^*\Omega^1(R, Lie(G))$ was good enough a compromise. This said, the same kinematical (or off-shell) phase space used in this article can be found in a fully covariant manner by focusing on the degree zero part of the BV-BFV boundary structure (here ``boundary'' is understood in relation to the spacetime bulk). I have added a comment regarding this point in footnote 15.

  1. Virtually the same criticism regarding my references to Donnelly and Freidel was raised by the other referee. I refer to that answer for my thoughts on it.

---

## Round 2 · List of Changes

LIST OF CHANGES

1) Minor grammatical changes throughout.

2) Improvements in section 1.3 following the suggestions of the anonymous referee.

3) End of sect. 1.5 (paragraph below eq 1): I have added a comment on the relationship between the reduction procedure adopted in this paper and the “canonical” one by Marsden-Weinstein. More details can be found in a new 2-page long appendix (A.1). This is in answer to prof. Teh's first comment.

4a) I have slightly amended section 1.6 (in particular the first 3 paragraphs) in order to better contextualize the physical origin of superselection in the light of standard criticisms present in the quantum foundation literature. This criticism was mentioned in the previous version but left unaddressed. It is relevant in regard to the transition from the superselection framework to that involving edge modes.

4b) The same goes for the first two paragraphs of section 5.1.

5) Footnote 15 has been modified to justify the choice of kinematical phase space (point 2 from prof. Teh's review).

6) Added footnotes 49 and 63.

7) Corrected reference 34 of the current numbering (it mistakenly linked to another paper by one of the authors which was not relevant for the present context).

8) Added a few references: [16–20] on Marsden-Weinstein-Meyer reduction and related topics, [30] on superselection and reference frames, [59] on an alternative, linear, edge mode phase space, [68] on relational interpretations of gauge theories, [72]on Marsden-Weinstein-Meyer reduction.

---

## Editorial Decision

published